# GROUNDING AI EXPLANATIONS IN EXPERIENCE: A REFLECTIVE COGNITIVE ARCHITECTURE FOR CLINICAL DECISION SUPPORT

## ABSTRACT

Effective disease prediction in modern healthcare demands the twin goals of high accuracy and transparent, clinically meaningful explanations. Existing machine learning and large language model (LLM) based approaches often struggle to balance these goals. Many models yield accurate but unclear statistical outputs, while others generate fluent but statistically unsupported narratives, often undermining both the validity of the explanation and the predictive accuracy itself. This shortcoming comes from a shallow interaction with the data, preventing the development of a deep, detailed understanding similar to a human expert's. We argue that high accuracy and high-quality explanations are not separate objectives but are mutually reinforcing outcomes of a model that develops a deep, direct understanding of the data. To achieve this, we propose the Reflective Cognitive Architecture (RCA), a novel framework that coordinates multiple LLMs to learn from direct experience. RCA features an iterative rule refinement mechanism that improves its logic from prediction errors and a distribution-aware rules check mechanism that bases its reasoning in the dataset's global statistics. By using predictive accuracy as a signal to drive deeper comprehension, RCA builds a strong internal model of the data. We evaluated RCA on one private and two public datasets against 22 baselines. The results demonstrate that RCA not only achieves state-of-the-art accuracy and robustness with a relative improvement of up to 40% over the baseline but, more importantly, leverages this deep understanding to excel in generating explanations that are clear, logical, evidence-based, and balanced, highlighting its potential for creating genuinely trustworthy clinical decision support systems. The code is available at `https://anonymous.4open.science/r/anonym107`. Access to the CRT dataset requires an application and approval process. If approved, we will anonymize and open-source the data.

## 1 INTRODUCTION

Disease prediction is a foundation of modern healthcare, providing a crucial opportunity for timely interventions that can slow disease progression, improve patient outcomes, and reduce medical costs (Nahian et al., 2022). In clinical practice, the data for such predictions are typically structured in tabular formats, containing a wealth of patient information (Nahian et al., 2022; Fang et al., 2024). However, the ultimate usefulness of a predictive model in a high-stakes clinical setting is determined by the twin requirements of predictive accuracy and the ability to explain its reasoning in a manner that clinicians can trust and act upon.

The major gap in current predictive systems lies in meeting both these needs at the same time. An effective system must not only predict correctly but also generate high-quality descriptions that explain its reasoning. Such an explanation must meet several key criteria derived from cognitive science and medical practice. First, it must have a low Cognitive Load (CL) (Sweller, 2011), presenting information clearly and concisely. Second, it must show sound Logical Argumentation (LA) (Toulmin, 2003), with a coherent reasoning process. Third, it must be based on Evidence-based Medicine (EBM) (Guyatt et al., 1992), matching both established medical knowledge and the statistical facts of the data. Finally, it must actively reduce Cognitive Biasing (CB) (Kahneman, 2011) by presenting a

balanced view. The failure of current AI to deliver accurate predictions paired with such high-quality explanations is a main obstacle to its adoption in clinical decision-making.

The ongoing challenge is that existing methods often fail to meet both requirements. Classical machine learning models, such as linear regression (Tibshirani, 1996) and tree-based approaches (Prokhorenkova et al., 2018), can achieve good results, but their explanatory ability is limited to statistical outputs like feature importance scores. These are not narrative explanations and need significant expert analysis, increasing cognitive load. On the other hand, the arrival of Large Language Models (LLMs) (Zhao et al., 2025) brought the promise of natural language explanations. However, when applied simply, they often lack a deep, detailed understanding of the specific dataset. Their reasoning can become "statistically unsupported," a weakness that leads to two problems: they produce explanations that seem medically believable but are not supported by the data, and this same shallow understanding often harms their predictive accuracy.

Our key insight is that high predictive accuracy and high-quality explanations are not conflicting goals but are two results of a single, deeper process: developing a direct, experience-based understanding of the data. This is like how a human expert dives deep into data before drawing conclusions. We rethink predictive accuracy not just as an end goal, but as a crucial reward signal that drives the model to build a more robust and fundamental "experience" with the underlying patterns. By optimizing for correct predictions and improving robustness against the data noise common in medical datasets, the model is forced to achieve a deep data understanding. This deep understanding is the necessary condition for generating explanations that are insightful, reliable, and clinically useful, with a high-performance predictive model appearing as a valuable, simultaneous output. The core objective is to predict and explain, but robustness to data noise should also be considered for real-world clinical applications.

To achieve this, we propose the **Reflective Cognitive Architecture (RCA)**, a framework designed to enable LLMs to learn directly from data through a process of experience and reflection. RCA includes two core mechanisms. The iterative rule refinement mechanism allows the model to learn from its mistakes, treating each incorrect prediction as an experience to be turned into abstract rules, thereby creating sound logical argumentation (LA). In addition, a distribution-aware rules check mechanism grounds these rules in the statistical reality of the training data. This mechanism uses a summary of the data distribution as a contextual "sanity check.", promoting evidence-based medicine (EBM) and reducing cognitive biases (CB).

We conducted extensive experiments on three disease prediction datasets, including a private real-world dataset for Catheter-Related Thrombosis (CRT), comparing RCA against 22 baselines. Our evaluation judges models on their ability to achieve both high predictive performance and high-quality explanations, as well as their robustness to data noise. The results demonstrate that RCA significantly outperforms existing methods on all fronts, confirming our main idea that a deeper, experience-based understanding of data is the key to achieving truly explainable and accurate AI for healthcare.

In summary, our main contributions are as follows:

- We rethink the problem of explainable disease prediction, arguing that predictive accuracy and high-quality descriptive statements are mutually reinforcing outcomes of a deep data understanding, which should be the main goal of the model.

- We propose RCA, A novel architecture featuring an iterative refinement mechanism to build understanding from experience and a distribution-aware check mechanism to make sure this understanding is statistically based and robust.

- We prepare a real-world dataset for CRT and create a complete evaluation framework judging predictive accuracy, robustness, and explanation quality based on principles of cognitive load, logical argumentation, evidence-based medicine, and cognitive bias.

- We demonstrate through extensive experiments that RCA outperforms a wide range of baselines in achieving a better balance of accuracy and explanation quality, supported by good robustness.

## 2 RELATED WORK

In this section, we place our work within two key areas: the evolution of explainability in disease prediction models (2.1) and the data interaction methods of LLM-based agents (2.2).

## 2.1 EXPLAINABILITY IN DISEASE PREDICTION

The quest for explainability in disease prediction is not new, but its definition and relationship with accuracy have changed over time (Sun et al., 2024). Early approaches favored models that were naturally understandable. For instance, methods like linear regression (Hoerl & Kennard, 2000; Tibshirani, 1996) and decision trees (Breiman, 2001; Prokhorenkova et al., 2018) were valued for their transparency and provided reasonable performance. However, their explanations were limited to statistical outputs, not narrative descriptions, which require significant expert analysis and place a high cognitive load on physicians.

Subsequently, more complex deep learning models emerged, achieving very high accuracy using Transformer (Hollmann et al., 2022) or FFN (Gorishniy et al., 2023). But most of them were often "black boxes," making their reasoning difficult to understand. This shifted the focus to post-hoc explanation techniques, which can be unfaithful to the model's actual reasoning process. Concurrently, Neural-Symbolic (NeSy) Networks like LTN(Badreddine et al., 2022) and LNN(Riegel et al., 2020) were explored to integrate logic into neural networks. While inherently transparent, they often struggle to generate narrative explanations or automatically discover complex rules from noisy data.

The emergence of LLMs, particularly those specialized for the medical domain (Medical LLMs) (Dou et al., 2025), offered a path toward generating natural language explanations. Initial efforts used LLMs to interpret statistical outputs or directly analyze tabular data. While these methods can produce fluent text, they often suffer from a disconnect from the data's statistics. This lack of grounding is a critical weakness, as it frequently leads to a dual failure: the explanations are not only invalid and untrustworthy, but the shallow reasoning process also harms the accuracy of the prediction itself. Furthermore, an explanation is still insufficient even if it is fluent and statistically grounded; in high-stakes settings like healthcare, it must be demonstrably usable by and useful to human experts Slack et al. (2023).Our work addresses the question by creating a direct link between the learning process for accuracy and the generation of explanations, ensuring they are two sides of the same coin.

## 2.2 LLM-BASED AGENT

LLM-based agents are increasingly being developed to tackle complex tasks by interacting with external environments or tools (Shen, 2024; Wang et al., 2024; Wu et al., 2024). The main methods for data analysis tasks involve optimizing for tool use or code generation.

**API-based agents** (Shen, 2024; SHEN et al., 2025; Shen et al., 2025) interact with data through a fixed set of functions. This approach creates a layer of abstraction between the agent and the raw data. The agent learns to become a skilled tool-caller, but it does not develop a detailed, instance-level understanding of the data's specifics. This abstraction hinders its ability to generate descriptive statements that are rich in detail, as it only perceives the data through the summarized lens of its tools.

**Code-generation agents** (OpenAI, 2023; Zhang et al., 2024; Guo et al., 2024) represent a more flexible approach, where the LLM writes and executes code (e.g., Python scripts) to perform analysis. While powerful, this method still maintains a degree of separation. The agent's core task becomes generating correct code, and the insights are derived from the code's output. The process of deep, repeated reflection on individual data points and their relationships is often bypassed in favor of executing a script that provides a summarized result.

RCA deliberately departs from these tool-focused methods. It is an agent that engages with data directly, without the mediation of external tools or code interpreters. Its core innovation lies in its internal, reflective process, which forces the model to build its understanding from direct "experience" with the data, imitating how a human analyst develops intuition. By avoiding the shortcuts of tool use, RCA optimizes for data comprehension directly, fostering a deeper and more robust understanding that serves as the essential foundation for both high accuracy and high-quality explanations.

## 3 METHODS

Our methodology is designed to build a model that is both highly accurate and produces superior explanatory statements. We posit that this dual objective is best achieved by forcing the model to develop a deep, experience-driven understanding of the data. To this end, we designed the Reflective

Cognitive Architecture (RCA), a framework that fosters this deep engagement rather than a superficial analysis. The final output for each patient $s_i$ is a prediction $\hat{\mathcal{Y}}_i$ that includes not only an accurate binary disease label $\hat{y}_i \in \{0, 1\}$ but also a high-quality explanatory statement $\hat{e}_i$. Formally, let $S = \{s_i\}_{i=1}^{N}$ be the dataset where each patient is represented by a vector of structured clinical features $f_i$ and a true disease state $y_i$. This section first provides an overview of the RCA architecture and then details its two core components: the Iterative Rules Optimization mechanism and the Distribution-aware Rules Check mechanism.

## 3.1 RCA OVERVIEW

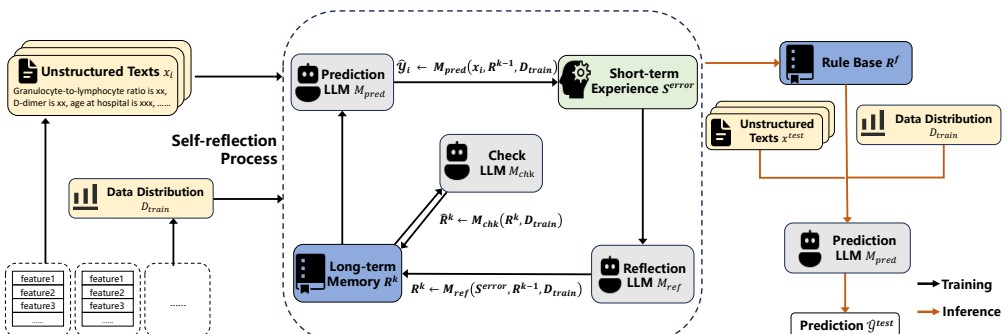

Figure 1: RCA pipeline. $S^{error}$ is a collection of misclassified samples (errors) from $M_{pred}$. $R^k$ is the rule base for the current iteration $k$, and $\hat{R}^k$ denotes the checked and refined rule base from iteration $k$. RCA uses reflective cycles and additional checks to directly analyze data, building a deep understanding that enhances both prediction accuracy and the generation of detailed, grounded explanatory statements.

As illustrated in Figure 1, the RCA pipeline systematically builds and refines data understanding. The process unfolds as follows:

**Data Narrative.** To enable an LLM to "read" and reason about patient data, we first transform the structured features $f_i$ into an unstructured text narrative $x_i$ as shown in Figure 1. For example, a data row is converted into a sentence like "Granulocyte-to-lymphocyte ratio is 4.88, D-dimer is 3.16, no chemotherapy, catheterization is CVC." This makes the data directly accessible to the LLM, with the patient record reformulated as $s_i = (x_i, y_i)$.

**Distribution Extraction.** To ensure the model's reasoning is grounded in Evidence-based Medicine (EBM) and to mitigate Cognitive Biasing (CB), we provide it with a global statistical context. Along with data transformation in Figure 1, we extract the data distribution $\mathcal{D}_{train}$ from the training set, which summarizes the statistical properties of the entire patient cohort (e.g., means, quantiles, frequencies). This global context prevents the model from over-interpreting individual data points. A concrete example of such a distribution summary is provided in Appendix A.15.

**Guided Prediction via a Dynamic Rule Base.** At the heart of RCA is a dynamic rule base, $R$, which serves as the model's evolving long-term memory. The "Self-reflection Process" of Figure 1 shows that, for any given patient $x_i$, a prediction LLM, $M_{pred}$, uses the current rule base $R^{k-1}$ and the global data distribution $\mathcal{D}_{train}$ to generate its output:

$$\hat{\mathcal{Y}}_i = (\hat{y}_i, \hat{e}_i) \leftarrow M_{pred}(x_i, R^{k-1}, \mathcal{D}_{train}) \tag{1}$$

By explicitly using the rule base, the model's predictions are guided by a consistent set of principles, fostering sound Logical Argumentation (LA) within the explanation $\hat{e}_i$.

**Dynamic Maintenance of the Rule Base.** The rule base $R$ is maintained through a dual-process feedback loop. The iterative rules optimization mechanism functions as a "reflective optimizer," refining logic based on instance-level prediction errors, while the distribution-aware rules check mechanism acts as a "statistical validator," auditing these updates against global evidence ($\mathcal{D}_{train}$). This cyclic interaction ensures $R$ evolves dynamically at each epoch, balancing the precision required to correct specific failures with the robustness needed to prevent overfitting.

**Training and Prediction Phases.** The training phase is a dynamic process where the rule base is iteratively refined by two LLMs, $M_{ref}$ and $M_{chk}$, based on prediction outcomes. In the testing phase, marked with orange lines in Figure 1, the final, optimized rule base $R^f$ is used to generate predictions and explanations $\hat{\mathcal{Y}}^{test}$ for unseen data. The resulting explanation is thus a product of a deep, iterative learning process that was driven by the pursuit of accuracy.

## 3.2 ITERATIVE RULES OPTIMIZATION: BUILDING LOGICAL ARGUMENTATION FROM EXPERIENCE

To generate explanations with sound Logical Argumentation (LA), a model must possess a coherent reasoning framework. The iterative rules optimization mechanism builds this framework by emulating experiential learning: treating prediction errors as opportunities to reflect and refine its understanding. This mechanism converts the short-term experience of specific errors into the long-term memory of abstract, generalizable rules.

**Iterative Reflection Loop.** The process is a feedback loop where $M_{pred}$ uses the rule base $R^{k-1}$ to make predictions. The instances where its predictions are incorrect constitute the direct feedback—the "experience"—that drives learning.

**Short-term Experience via Error Samples.** Misclassified samples are collected into a textual format, $S^{error}$:

$$S^{error} = \mathrm{conc}(s_j^{error})_{j=1}^{T} \tag{2}$$

where $T$ is the error batch capacity, and conc is concatenation function that stitches samples together. This aggregation of incorrect cases highlights deficiencies in the current rule base.

**Long-term Memory in the Rule Base.** This experience is processed by a reflection LLM, $M_{ref}$, to update the rule base, distilling specific errors into robust principles:

$$R^k \leftarrow M_{ref}(S^{error}, R^{k-1}, \mathcal{D}_{train}) \tag{3}$$

Through this process, the model's logical framework ($R^k$) evolves through trial and error. This ensures that the pursuit of higher accuracy directly sharpens the logical rules that will form the backbone of the final explanation.

## 3.3 DISTRIBUTION-AWARE RULES CHECK: GROUNDING LOGIC IN EVIDENCE

While iterative learning builds a logical framework, it risks creating rules that are statistically spurious. To ensure explanations are grounded in Evidence-based Medicine (EBM), mitigate Cognitive Biasing (CB), and improve robustness, the logic must be validated against the global data. The distribution-aware rules check mechanism serves as a safeguard, ensuring the model's reasoning is statistically sound.

**Additional Rules Check.** At the end of each epoch, a checking LLM, $M_{chk}$, reviews the rule base $R^k$ using the global data distribution $\mathcal{D}_{train}$ as a reference:

$$\hat{R}^k \leftarrow M_{chk}(R^k, \mathcal{D}_{train}) \tag{4}$$

$M_{chk}$ removes low-quality or overly specific rules and summarizes general rules for detecting outliers. This grounds the model's reasoning in the statistical properties of the dataset, directly promoting an evidence-based approach and strengthening the model against noisy or atypical data. The refined rule base $\hat{R}^k$ then replaces the previous version for the next epoch (denoted as $R^k$ for consistency).

**Mutual Enhancement.** Together, the iterative optimization and distribution-aware check form a synergistic, closed-loop system:

$$\hat{R}^k \underset{M_{chk}}{\overset{cover}{\rightleftarrows}} R^k \underset{M_{ref}}{\overset{M_{pred}}{\rightleftarrows}} \hat{\mathcal{Y}}_i \tag{5}$$

The iterative process ($M_{pred}$, $M_{ref}$) builds the core logical structure (LA) from the experience of pursuing accuracy, while the check mechanism ($M_{chk}$) ensures this structure is statistically robust and evidence-based (EBM, CB). This dual-process architecture ensures that RCA develops a deep, reliable understanding of the data, which is the essential foundation for achieving both high accuracy and generating trustworthy explanatory statements. To illustrate this synergistic process, a detailed walkthrough of the entire reflective cycle is presented in Appendix A.3.

## 4 EVALUATION

We designed our evaluation to test the central hypothesis: that a deeper, experience-driven data understanding leads to synergistic improvements in predictive accuracy, explanation quality, and robustness. For clinical decision support, both accuracy and explanations are critical. This section first details the experimental setup (4.1), then presents the main results correlating performance and explanation quality (4.2), tests the model's resilience against data noise (4.3), validates our architecture through an ablation study (4.4), and provides a qualitative case study (4.5).

### 4.1 SETUP

**Datasets.** To ensure a comprehensive evaluation, we selected three distinct datasets. For each dataset, we split it into training, validation, and test sets following a 3:1:1 ratio.

- **CRT:** We curated a real-world dataset for Catheter-Related Thrombosis (CRT) in collaboration with Feitian Hospital[1]. This proprietary dataset comprises 315 cancer patients, offering a high-stakes, clinically relevant challenge. This research was approved by the Medical Science Research Ethics Committee of the authors' institute.

- **Diabetes** (Pore, 2025): A public benchmark dataset for diabetes prediction with 8 highly correlated features. We use a subset of 415 cases to test the model's performance on a well-understood clinical problem.

- **Heart Disease** (Rdeki, 2025): A public dataset for heart disease prediction featuring 19 primarily categorical features, including lifestyle and biometric data. Its 965 cases challenge the model's ability to reason over heterogeneous data types.

More details of datasets are provided in Appendix A.4.

Furthermore, to thoroughly test the real-world scalability of our approach, we included two additional, large-scale datasets in our evaluation.

- **CRT_ex**: We significantly expanded our private **Catheter-Related Thrombosis (CRT)** dataset, increasing the sample size from 315 to **1,891 patients**.

- **Cardiovascular Disease** (Shihab, 2025): We introduced a new, large-scale public benchmark dataset for **Cardiovascular Disease** prediction, containing **70,000 patient records**.

The experimental setup for these datasets followed the same protocol as our main experiments. To maintain focus on our core contributions, the detailed results and analysis for this scalability study are presented in Appendix A.13.

**Baselines.** We compare RCA against 25 baselines representing diverse approaches.

- **Traditional ML Models:** We include Lasso regression (Tibshirani, 1996) and Catboost (Prokhorenkova et al., 2018). These models are standard for tabular data but produce statistical artifacts (e.g., coefficients, feature importance) rather than narrative explanations, representing a baseline for expert-driven interpretation. A 'Qwen3-235B' is used to generate an explanation for these models; we provide a detailed introduction in Appendix A.5.

- **Neural-symbolic Networks:** To address a key alternative to our approach, we include two prominent neural-symbolic methods: **Logic Tensor Networks (LTN)**(Badreddine et al., 2022) and **Logic Neural Networks (LNN)** (Riegel et al., 2020). These baselines test whether explicit logic integration outperforms RCA's emergent, LLM-based reasoning. We also used a 'Qwen3-235B' to generate an explanation for these models; we provide a detailed introduction in Appendix A.5

- **LLM-based Methods:** We test the ability of 4 leading non-reasoning LLMs ('Qwen2.5-72B-Instruct', 'DeepSeek-V3-64k', 'DeepSeek-Chat-V3.1', 'GPT-4.1-2025-04-14') to perform zero-shot prediction and explanation directly from tabular data.

- **Reasoning LLMs:** We evaluate 6 advanced reasoning-focused LLMs ('DeepSeek-R1', 'Qwen3-30B-A3B', 'Qwen3-235B-A22B-Instruct-2507', 'GPT-5-2025-08-07', 'o3-mini-2025-01-31', 'o4-mini-2025-04-16') to assess whether enhanced general reasoning capabilities translate to better data understanding and explanation in this specific domain.

---

[1]The hospital name has been anonymized to comply with the anonymity policy.

- **Medical LLM:** We also include 'Baichuan-M2'(Dou et al., 2025), a prominent large language model specifically optimized for the medical domain, to assess the performance of domain-specific models.
- **LLM-based Agents:** We include two agent paradigms that reflect the state-of-the-art in LLM-driven analysis, comprising 9 and 1 methods, respectively. The **LLM+Tools** approach equips models with predefined functions, while the **LLM+Code** approach utilizes a code interpreter (OpenAI, 2023). These baselines test the hypothesis that tool use abstracts away the fine-grained data interaction necessary for deep understanding. For details about the functions and code, please refer to the Appendix A.6.

All LLM baselines were provided with the same global data distribution ($\mathcal{D}_{train}$) and illustrative in-context examples from the training set that RCA utilized.

**Evaluation Metrics.** Our evaluation employs two categories of metrics.

- **Metrics for Predictive Performance:** We use accuracy, the Matthews Correlation Coefficient (MCC), and the F1-score. These metrics serve as quantitative proxies for the depth and correctness of the model's understanding, with MCC and F1-score being particularly informative for the imbalanced datasets common in medicine.
- **Metrics for Explanation Quality:** To directly measure the quality of the generated descriptive statements, we developed four criteria grounded in cognitive science and medical practice: Cognitive Load (CL), Logical Argumentation (LA), Evidence-based Medicine (EBM), and Cognitive Biasing (CB). These criteria have been recognized by 3 doctors as clinically valuable, as they align with real-world medical assessment needs for evaluating the clarity, rationality, and reliability of explanatory content. We invited 3 doctors to score 100 samples for each of the core methods: Traditional MLs ('CatBoost'), LLM-based Methods ('GPT-4.1'), Reasoning LLMs ('Qwen3-235B'), LLM-based Agents ('Qwen3-235B + tools'), and RCA ('RCA+GPT-4.1'). For all other methods, we used 'Qwen3-30B' with a carefully crafted prompt that achieved 90% scoring agreement with doctors.Then doctors and LLM scored each explanation on a scale of 1 to 10 for each criterion based on a detailed rubric. The rubric of these metrics are provided in Appendix A.7, while specific examples can be found in Appendix A.10.
- **Metrics for Explanation Usability:**To further validate usability, we conducted a formal human-centric study to assess practical usability. We asked 3 expert clinicians to review 50 sample explanations from each of the three datasets.In this study, participants compared the explanation from RCA against the explanation from strongest baseline 'Qwen3-235B'. For each pair, the clinicians were asked to identify the superior explanation across our four core criteria. This forced-choice design allows us to directly and formally quantify the expert preference for RCA's explanations.

**Implementation Details.** For our method, we implement RCA using both 'Qwen2.5-72B-Instruct' and 'GPT-4.1-2025-04-14' as the base LLMs(abbreviated as RCA+Qwen2.5 and RCA+GPT-4.1 to demonstrate its architectural benefits. The error batch capacity T is set to 25. The model is trained for 15 epochs on the CRT and Diabetes datasets, and 25 epochs on the Heart Disease dataset. All prompt templates used in RCAcan be found in Appendix A.16

## 4.2 MAIN RESULTS: DEEP UNDERSTANDING YIELDS SYNERGISTIC GAINS

In this section, we conduct qualitative analysis of main experiment on the CRT dataset, while the quantitative results of all datasets can be found in the Appendix A.11. Specifically, the qualitative analysis includes Lasso, Catboost, 'DeepSeek-Chat-V3.1'(abbreviated as 'DeepSeek-V3.1'), 'o4-mini-2025-04-16'(abbreviated as 'o4-mini'), 'Qwen3-235B-A22B-Instruct-2507' (abbreviated as 'Qwen3-235B') and 'o3-mini-2025-01-31+Code' (abbreviated as 'o3-mini+Code') as baselines .

Our central thesis posits that predictive accuracy and explanation quality are not in opposition but are synergistic outcomes of a model's deep, first-hand understanding of data. We test this by plotting model performance against explanation quality. As shown by the hollow points in Figure 2, which represent the main experimental results, the RCA-based approaches consistently occupy the top-right quadrant, achieving the highest predictive performance (Accuracy and MCC) while also delivering high-quality explanations (as indicated by the Cognitive Load score, with detailed metrics in Appendix A.11).

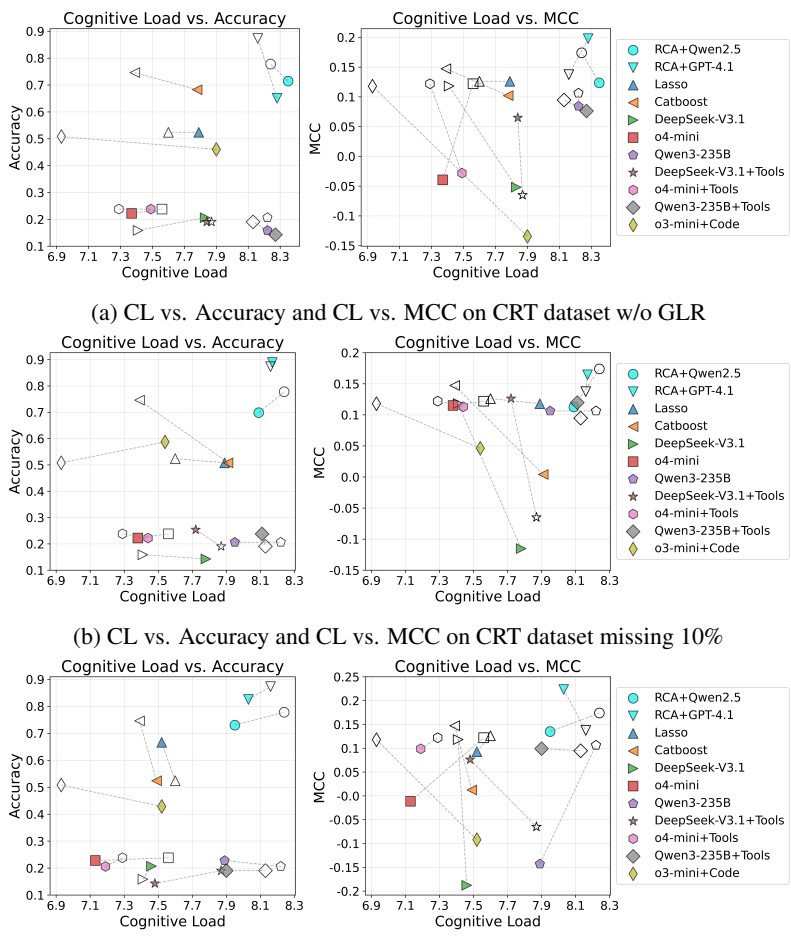

(a) CL vs. Accuracy and CL vs. MCC on CRT dataset w/o GLR

(b) CL vs. Accuracy and CL vs. MCC on CRT dataset missing 10%

(c) CL vs. Accuracy and CL vs. MCC on CRT dataset abnormal 10%

Figure 2: Results of the main experiment and the robustness experiment on CRT dataset. Hollow dots represent the main experiment, solid dots represent the robustness experiment, and dots of the same shape represent the same approach. The dashed line measures performance variation. RCA demonstrates not only the best results (4.2) but also gets little performance fluctuations(4.3, showing the resilience to data noise.

This result validates our core hypothesis. RCA's success is not coincidental; it is a direct consequence of its architecture, which forces a deep engagement with the data. The high predictive accuracy serves as confirmation that the internal model RCA has built is a correct representation of the underlying clinical patterns. The high-quality explanation is the natural articulation of this well-grounded understanding.

In contrast, other paradigms falter because they lack this deep engagement.

- **Traditional ML models** like Catboost achieve competitive accuracy, but explanations generated by 'Qwen3-235B' based on their prediction results perform poorly overall.

- **LLM-based agents (LLM+Tools, LLM+Code)** are hindered by their layer of abstraction. By interacting with data through APIs or code outputs, they become proficient tool-users but never develop a granular, instance-level "feel" for the data. Their understanding remains second-hand, limiting the depth and reliability of their explanations.

- **Standalone LLMs**, even powerful reasoning models like 'o4-mini', demonstrate the pitfalls of "statistically de-grounded" reasoning. While they may achieve a respectable MCC, their tendency to generate plausible but unsubstantiated narratives leads to lower accuracy and poorer explanation scores, highlighting a superficial grasp of the specific dataset.

Table 1: Results of ablation studies. The experimental results indicate that several core modules in RCA play irreplaceable roles.

| | Qwen2.5-72B | | | | GPT-4.1 | | | |
|---|---|---|---|---|---|---|---|---|
| | original | distribution | reflection | check | original | distribution | reflection | check |
| **CRT** | | | | | | | | |
| **Accuracy** | **0.7778** | 0.5873 | 0.5714 | 0.6032 | **0.8730** | 0.6508 | 0.7143 | 0.7937 |
| **MCC** | **0.1739** | −0.0702 | 0.1513 | 0.1691 | **0.1373** | 0.0824 | −0.0024 | 0.0517 |
| **F1-score** | **0.2222** | 0.0714 | 0.1818 | 0.1935 | **0.2000** | 0.1538 | 0.1000 | 0.1333 |
| **CL** | **8.24** | 7.70 | 7.54 | 7.75 | **8.16** | 7.45 | 7.57 | 7.40 |
| **Diabetes** | | | | | | | | |
| **Accuracy** | **0.7831** | 0.7711 | 0.7229 | 0.7349 | **0.7470** | 0.7229 | 0.6867 | 0.7349 |
| **MCC** | **0.5406** | 0.4926 | 0.4424 | 0.4169 | **0.4244** | 0.3857 | 0.3222 | 0.4232 |
| **F1-score** | **0.7097** | 0.6667 | 0.6567 | 0.6206 | **0.6038** | 0.5965 | 0.5667 | 0.5652 |
| **CL** | **8.13** | 7.93 | 7.93 | 7.77 | **8.03** | 7.18 | 7.34 | 7.73 |
| **Heart Disease** | | | | | | | | |
| **Accuracy** | **0.5647** | 0.3523 | 0.3575 | 0.3627 | **0.7461** | 0.5337 | 0.3471 | 0.4663 |
| **MCC** | **0.0547** | −0.0477 | −0.1001 | −0.0349 | **0.1493** | −0.0359 | −0.1134 | −0.0923 |
| **F1-score** | 0.1290 | 0.3169 | 0.2874 | **0.3204** | **0.2898** | 0.2623 | 0.2841 | 0.2481 |
| **CL** | **7.62** | 7.32 | 7.54 | 7.53 | **7.74** | 7.12 | 7.13 | 6.72 |

Ultimately, the results show that by prioritizing the development of a deep data understanding, RCA organically achieves state-of-the-art results in both prediction and explanation, demonstrating their synergistic relationship. Full results for all baselines across three datasets please refer to Appendix A.11.

## 4.3 ROBUSTNESS: DEEP UNDERSTANDING CONFERS RESILIENCE

Similarly, we conduct qualitative analysis of robust experiment on the CRT dataset, while the quantitative results can be found in the Appendix A.12.

Medical data is notoriously noisy and incomplete. A truly deep understanding should be resilient to such imperfections, distinguishing robust signals from spurious noise. We tested this by degrading the CRT dataset through feature removal ("GLR"), random value deletion (10%), and the introduction of outliers (10%).

The results, visualized by the solid points and connecting dashed lines in Figure 2, demonstrate RCA's superior robustness. The performance of RCA (both 'Qwen2.5-72B' and 'GPT-4.1' versions) shows minimal degradation across all noise conditions, as indicated by the short dashed lines connecting the hollow (original) and solid (noisy) points. This stability is a direct outcome of its design. The distribution-aware rules check grounds the model's logic in global statistics, preventing it from being misled by local anomalies or outliers. This mechanism is particularly effective at identifying and flagging extreme outliers that might otherwise corrupt predictions, as detailed in the edge case analysis in Appendix A.8. The iterative rules optimization builds a generalized understanding from cumulative experience, making the model less dependent on any single data point or feature.

In contrast, the performance of many baselines is far more volatile. For instance, both Catboost and 'DeepSeek-V3.1' suffer significant drops in MCC when faced with missing data, revealing that their underlying models may have overfit to patterns that are not robust. Their longer dashed lines signify a shallower understanding that shatters under data stress. The resilience of RCA is therefore not just a feature but further evidence of the foundational and robust nature of its data understanding.

## 4.4 ABLATION STUDY

To validate that RCA's performance stems directly from its proposed cognitive mechanisms, we conducted an ablation study by systematically removing its core components. As shown in Table 1, the removal of any key module leads to a significant performance collapse, confirming that each part is essential to the process of building a deep understanding. More explanation results are provided in Appendix A.14.

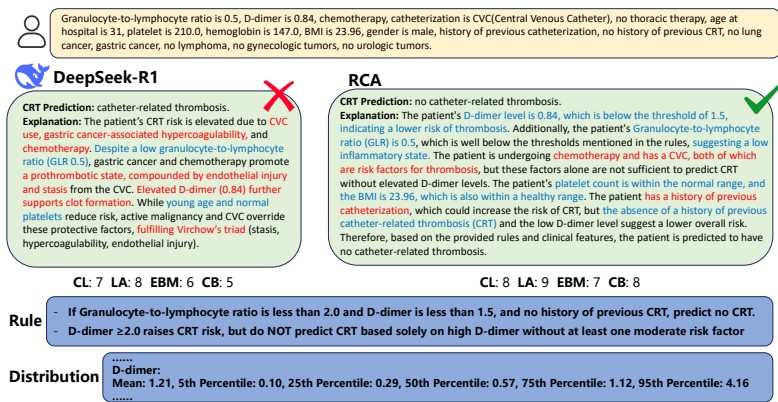

Figure 3: Comparison of explanations from 'DeepSeek-R1' and RCA for the same patient. RCA demonstrates superior reasoning by integrating quantitative thresholds and providing a balanced, evidence-based argument, a direct result of its deep data understanding. 'DeepSeek-R1' ś explanation, while fluent, is statistically ungrounded and leads to an incorrect prediction.

### 4.5 QUALITATIVE ANALYSIS: A PREDICTION CASE STUDY

Aggregate metrics validate our approach, but a case study reveals the practical difference between superficial and deep understanding. Figure 3 contrasts the explanations from RCA and a strong reasoning baseline, 'DeepSeek-R1', for the same patient from the CRT dataset.

'DeepSeek-R1' incorrectly predicts CRT, exemplifying the danger of statistically de-grounded reasoning. It constructs a plausible-sounding narrative by identifying risk factors (CVC, chemotherapy) but critically fails in its quantitative assessment. It misinterprets a D-dimer level of 0.84 mg/L as high risk, demonstrating a lack of awareness of the actual risk thresholds learned from the data distribution. This is a classic failure of a model that relies on general knowledge rather than a first-hand understanding of the specific dataset's statistical realities.

In contrast, RCA correctly predicts no CRT and generates an explanation that is a direct manifestation of its deep data understanding.

- Its reasoning is grounded in **Evidence-based Medicine (EBM)**, a result of the *distribution-aware rules check*. It explicitly compares the patient's D-dimer (0.84) and GLR (0.5) against the learned clinical risk thresholds (e.g., >1.5), correctly concluding they are "notably lower" and "well below the risk threshold."

- It exhibits strong **Logical Argumentation (LA)**, a product of the *iterative rules optimization*. It presents a balanced view, acknowledging risk factors but correctly reasoning that they are "insufficient for thrombosis without an elevated D-dimer."

This clear, logical, and evidence-based explanation is not a separate feature but the output of the same deep understanding that drove the accurate prediction. It is precisely this synergy that makes RCA a step towards truly trustworthy clinical AI.

## 5 CONCLUSION

In this paper, we challenged the conventional trade-off between predictive accuracy and explainability in clinical AI. We argued that these are not competing goals but synergistic outcomes of a model that develops a deep, first-hand understanding of the data. To achieve this, we introduced RCA, a novel framework that learns from experience through iterative rules optimization and grounds its reasoning in global statistics via a distribution-aware check. Our experiments demonstrate that by forcing the model to achieve this deeper comprehension, RCA not only attains state-of-the-art accuracy and robustness but also excels in generating clear, logical, and evidence-based explanatory statements. Future work could enhance RCA by integrating external knowledge bases, which would improve its reasoning on rare diseases or novel biomarkers where the LLM's pre-trained knowledge may be limited.

## 6 REPRODUCIBILITY STATEMENT

The work in this paper is well reproducible. The code used in the study is available at https://anonymous.4open.science/r/anonym107. Among the three datasets employed in the experiments, the Diabetes Dataset(https://www.kaggle.com/datasets/nanditapore/healthcare-diabetes/data), Heart Disease Dataset(https://www.kaggle.com/datasets/oktayrdeki/heart-disease) and Cardiovascular Disease Dataset(https://www.kaggle.com/datasets/alamshihab075/heart-failure-diagnosis-data-for-machine-learning/data) are public and can be accessed via the corresponding links; only the CRT Dataset is private, and its access requires submission of an application and subsequent approval. Data processing can refer to Section 3.1.

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

# A  APPENDIX / SUPPLEMENTAL MATERIAL

## A.1  DECLARATION ON THE USE OF LLMS

We declare that the use of LLMs during the preparation of this manuscript was strictly limited to language-related assistance, such as sentence refinement and grammatical correction. All substantive content was independently authored by the authors and rigorously reviewed and verified following any LLM-assisted modifications. During the experiments, all usage of LLMs was solely for academic research purposes, with no inappropriate applications. Detailed experimental settings are provided in the Experiments section of this paper. No other reliance on LLMs is involved in this work.

## A.2  ETHICS STATEMENT

All authors of this paper strictly adhere to the ICLR Code of Ethics throughout the entire research and manuscript preparation process. Prior to submitting this work, every author has carefully read and fully understood the content of the Code of Ethics, and confirms compliance with all its provisions.

This research strictly follows the ethical guidelines for conference participation outlined in the ICLR Code of Ethics, covering the integrity of paper submission, adherence to ethical standards in potential peer review , and constructive, respectful communication in any subsequent academic discussions related to this work. We further confirm that no content in this paper violates the ethical principles specified in the ICLR Code of Ethics, and that all research procedures are conducted with honesty, transparency, and respect for academic ethics.

## A.3  RCA REFLECTIVE PROCESS: A WALKTHROUGH

To provide a concrete illustration of the RCA framework, this section details the interaction between its two core mechanisms: **Iterative Rules Optimization** ($M_{ref}$) and **Distribution-aware Rules Check** ($M_{chk}$). We trace how the system processes a specific patient record during the training phase to demonstrate the synergy between instance-level learning and global-level statistical grounding.

### A.3.1  STEP 1: INITIAL STATE (CONTEXT)

Consider a scenario where the model is in the middle of the training process (Epoch $k$). It has already acquired some feature-coupled logic, but the rule base ($R^{k-1}$) lacks precision for borderline cases.

- **Current Rule Base ($R^{k-1}$):** Contains broad, clinically plausible rules, for example:
  - *Rule A:* "Patients undergoing **Chemotherapy** with a **Central Catheter (CVC/PICC)** are considered high risk if **D-dimer** $> 0.50$ **mg/L**, reflecting the prothrombotic effect of systemic treatment."
  - *Rule B:* "High inflammatory status, indicated by **GLR** $> 3.0$, coupled with **Age** $> 60$, predicts thrombosis even with normal D-dimer levels."
- **Global Data Distribution ($\mathcal{D}_{train}$):** The model possesses the statistical ground truth of the dataset. For instance, it identifies that for *no-CRT* patients in this specific cancer cohort, the 75th percentile of D-dimer is actually **1.12 mg/L** (notably higher than the standard 0.50 cutoff in Rule A), and the Granulocyte-to-lymphocyte ratio (GLR) typically ranges from **0.86 to 2.63**.

### A.3.2  STEP 2: THE PREDICTION ERROR (SHORT-TERM EXPERIENCE)

The prediction LLM ($M_{pred}$) encounters a specific instance, **Patient 4**, with the following profile:

- **Features:** GLR = 0.94 (Low), D-dimer = 0.58 (Marginally elevated), Chemotherapy = Yes, Catheter = PICC.
- **True Label:** No CRT.

**The Error:** Applying *Rule A*, the model identifies the risk factors (Chemotherapy + PICC) and notes the D-dimer (0.58) exceeds the 0.50 threshold. Consequently, it incorrectly predicts **"CRT"**. This failure constitutes the "short-term experience" ($S^{error}$) passed to the reflection module.

### A.3.3 STEP 3: ITERATIVE RULES OPTIMIZATION (BOTTOM-UP REFINEMENT)

The Reflection LLM ($M_{ref}$) analyzes this specific error. It recognizes that *Rule A* was too aggressive for this specific context: for Patient 4, a D-dimer of 0.58 was non-thrombotic, likely mitigated by the low inflammatory state (GLR=0.94).

To rectify this, $M_{ref}$ proposes a new, highly specific rule derived directly from the instance:

> ***Proposed Rule (Specific):*** *"If the patient has a GLR between **0.9–1.0** AND a D-dimer level between **0.4 and 0.6**, predict **No CRT**, overriding risk factors like Chemotherapy."*

While this rule successfully corrects the mistake by effectively "memorizing" Patient 4, it is dangerously over-fitted to the specific values of a single instance.

### A.3.4 STEP 4: DISTRIBUTION-AWARE RULES CHECK (TOP-DOWN SAFEGUARD)

At the end of the epoch, the Check LLM ($M_{chk}$) validates this proposed rule against the Global Data Distribution ($\mathcal{D}_{train}$).

- It observes that the proposed D-dimer range (0.4–0.6) falls well within the "safe zone" for the broader non-thrombosis population (where the 75th percentile is 1.12).
- It notes that the specific GLR range (0.9–1.0) is merely a narrow slice of the typical healthy range (0.86–2.63).

Recognizing that the specific rule is a subset of a broader statistical truth, $M_{chk}$ generalizes it into a robust, evidence-based rule that replaces the over-fitted proposal:

> ***Final Refined Rule (General):*** *"If the patient has a GLR between **0.86–2.63** (Normal Range) AND a D-dimer level **less than 1.12**, they are less likely to develop CRT, even in the presence of Chemotherapy."*

### A.3.5 SUMMARY

This walkthrough highlights the core philosophy of RCA: the **Refinement** mechanism uses specific errors to discover nuances (e.g., "Standard D-dimer cutoffs don't apply here"), while the **Check** mechanism uses global statistics to validate and generalize those discoveries (e.g., "Actually, anything under 1.12 is safe for this group"). This balance prevents overfitting to noise while ensuring the final rules are both accurate and robust.

## A.4 DATASET DETAILS

In this section, we will provide a detailed overview of the dataset.

- CRT: We collected a real dataset on Catheter-Related Thrombosis (CRT) for cancer patients from Feitian Hospital The dataset includes 315 cancer patients who underwent catheterization, with 32 diagnosed with CRT. The dataset contains a total of 17 features, 11 of which are categorical features and 6 are numerical features, including various tumor labels, laboratory test values, and other medically relevant data.
- Diabetes: This public dataset includes 8 features that are strongly associated with diabetes. We randomly selected 415 cases for diabetes prediction, of which 153 had diabetes. Among these 8 features, only "number of pregnancies" is a categorical feature, while the remaining 7 are numerical features.
- Heart Disease: This is a heart disease prediction dataset consisting of 19 features, most of which are categorical features. These features include lifestyle habits, blood tests, etc. We have 965 cases in this dataset, 193 of which were diagnosed with heart disease. In this dataset, numerical features are dominated by various laboratory test values, while categorical features mainly cover indicators related to different living habits.

### A.5 Explanation of Traditional MLs and Neural-symbolic Networks

Unlike LLMs, traditional machine learning and neural-symbolic algorithms cannot directly produce textual explanations. Therefore, for Lasso Regression and CatBoost we input the prediction results along with the feature coefficients or feature importance produced by each method, into a `Qwen3-235B`, which performs best among all LLMs in explanation scores. Then It's prompted to generate explanations based on prediction results and coefficients or importance. For our neural-symbolic networks, we input the final prediction and the truth values of its constituent logical predicates into a `Qwen3-235B` to generate a transparent, rule-based explanation.

### A.6 LLM-based Agents

#### A.6.1 LLM+Tools

We pre-defined a logistic regression function and a decision tree function to conduct relevant analysis,. These two functions can accept sample data and output feature correlation coefficients (for the logistic regression function) or feature importance (for the decision tree function) based on their built-in logic. For the LLM+tools method, when the LLM determines that tool invocation is necessary, it first organizes the data into a pre-specified format and sends it to the tool; after receiving the results returned by the tool, the LLM further uses these results to achieve data interpretation.

#### A.6.2 LLM+Code

The code interpreter of o3-mini is one of its core functional modules, supporting dynamic tool generation and code execution. As a cost-effective inference model released by OpenAI, o3-mini exhibits excellent performance in STEM (Science, Technology, Engineering, Mathematics) fields—including science, mathematics, and coding (OpenAI, 2023). To this end, we input data into the LLM + code agent, and explicitly instruct the LLM to analyze the target problem by writing code; this setup aims to investigate whether the code interpreter can enhance the LLM's understanding of the data.

### A.7 Rubric of Explanation Criteria

Considering the value of clinical application, We designed detailed scoring rubric on each criterion for doctors to reference. Cognitive Load, Logical Argumentation, Evidence-Based Medicine and Cognitive Biasing are presented sequentially in Section

#### A.7.1 Cognitive Load(CL)

Considering that the explanation is ultimately read by doctors, the definition of Cognitive Load is "Whether the explanation is easy for doctors to understand and analyze". Specific standards are as follows:

- **7-10 points:** Extremely easy to understand and analyze. The explanation uses concise, precise language; avoids redundant information; and structures content logically. Doctors can quickly grasp the core logic without additional effort.

- **5-7 points:** Moderately easy to understand and analyze. The explanation is mostly clear but may contain minor redundancies or slightly complex sentence structures. Doctors can grasp the core logic with minimal effort, without needing to re-read repeatedly.

- **3-5 points:** Difficult to understand and analyze. The explanation has confusing structure, ambiguous terminology, or excessive jargon. Doctors need to spend significant effort to understand the main content.

- **1-3 points:** Nearly impossible to understand and analyze. The explanation is disorganized, uses inaccurate or obscure language, and contains massive redundant or irrelevant information. Doctors cannot effectively grasp the logic even after repeated reading.

### A.7.2 LOGICAL ARGUMENTATION(LA)

The essence of an explanation lies in analyzing the reasoning process through logic, so the definition of Logical Argumentation is "Whether the expression is consistent and coherent, and whether the logic is clear and smooth".

- **7-10 points:** Fully consistent, coherent, and logically rigorous. The explanation has a clear logical thread; each statement connects naturally to the next; there are no contradictions or logical gaps; and the reasoning process from premises to conclusions is complete and persuasive.
- **5-7 points:** Mostly consistent, coherent, and logically clear. The overall logical thread is understandable, but there may be minor inconsistencies or weak transitions between statements. The reasoning process is generally complete with no major logical flaws.
- **3-5 points:** Inconsistent, incoherent, or logically confusing. The explanation has obvious logical gaps or occasional contradictions. The connection between statements is weak, making the overall logic difficult to follow.
- **1-3 points:** Severely inconsistent, incoherent, or logically invalid. The explanation has serious contradictions; the reasoning process is chaotic or nonexistent; and there is no clear logical connection between statements, leading to complete loss of persuasiveness.

### A.7.3 EVIDENCE-BASED MEDICINE(EBM)

The credibility of an explanation relies on the support of professional medical knowledge and evidence-based principles, so the definition of Evidence-Based Medicine is"Whether the explanation conforms to professional medical knowledge and evidence-based principles".

- **7-10 points:** Fully conforms to professional medical knowledge and evidence-based principles. All medical claims are accurate and supported by well-recognized evidence. No medical errors or misinformation exist.
- **5-7 points:** Mostly conforms to professional medical knowledge and evidence-based principles. Core medical claims are accurate, but minor details may lack strong evidence support or have slight imprecision. No critical medical errors.
- **3-5 points:** Partially conforms to professional medical knowledge and evidence-based principles. There are noticeable medical inaccuracies or over-reliance on low-quality evidence. These issues do not completely invalidate the explanation but reduce its professional credibility.
- **1-3 points:** Does not conform to professional medical knowledge and evidence-based principles. The explanation contains serious medical errors or promotes unsubstantiated claims. These issues make the explanation professionally unreliable.

### A.7.4 COGNITIVE BIASING(CB)

When generating explanations, if only supporting factors are listed, it is prone to falling into the trap of intuitive judgment (Kahneman, 2011). Therefore, the definition of CB is "Whether the evidence listed in the explanation is comprehensive, encompassing both factors supporting and opposing the final prediction".

- **7-10 points:** Extremely comprehensive evidence with no obvious bias. The explanation systematically lists key supporting factors and relevant opposing factors along with analysis. It also briefly discusses why opposing factors do not change the conclusion, showing balanced consideration.
- **5-7 points:** Mostly comprehensive evidence with minimal bias. The explanation lists and simply analyze some key supporting factors and some major opposing factors. While a few minor opposing factors may be omitted, the overall presentation is balanced, and the bias is not obvious.
- **3-5 points:** Incomplete evidence with noticeable bias. The explanation focuses primarily on supporting factors and only mentions opposing factors superficially or omits important ones. The one-sided presentation makes the explanation lean heavily toward justifying the conclusion.
- **1-3 points:** Highly incomplete evidence with severe bias. The explanation only lists factors supporting the final prediction and completely ignores all relevant opposing factors. It appears as a one-sided justification rather than a balanced explanation of the reasoning process.

## A.8 EDGE CASE ANALYSIS: HANDLING EXTREME ANOMALIES

While our primary design objective for RCA was to generate high-fidelity, evidence-based explanations for the general patient population, the architecture is inherently well-suited to identifying and handling statistical anomalies. A critical challenge in real-world medical data is distinguishing between robust clinical signals and extreme noise (e.g., data entry errors or sensor malfunctions).

To demonstrate the robustness of RCA in such scenarios, we analyze a specific real-world case from the CRT dataset that contains extreme outlier values.

### A.8.1 CASE DESCRIPTION

The patient record presents with the following clinical features, including two physiologically improbable values:

- **Platelet Count:** 2300.0 (Population Mean $\approx$ 230.0)
- **Hemoglobin:** 1350.0 (Population Mean $\approx$ 118.0)
- **Other Features:** CVC catheterization, undergoing chemotherapy, D-dimer 0.89, Age 64.
- **Ground Truth:** No catheter-related thrombosis.

Baseline methods, lacking a grounded understanding of the global data distribution, often misinterpreted these extreme values as high-risk indicators due to their magnitude, leading to incorrect predictions of thrombosis.

### A.8.2 RCA PERFORMANCE AND ANALYSIS

In contrast, RCA correctly predicted "no catheter-related thrombosis." The **distribution-aware rules check** mechanism (§3.3) successfully utilized the global statistical context ($\mathcal{D}_{train}$) to flag these values as deviations falling far outside the 99th percentile.

The explanation generated by RCA explicitly articulates this reasoning:

> "The patient has a moderately elevated D-dimer (0.89), a **very high platelet count (2300.0)** and an **extremely high hemoglobin (1350.0)**, both of which are **clear outliers likely due to data entry errors or extreme physiological abnormalities**; such values far exceed physiologic ranges. ... Laboratory outlier values alone should not be used as the primary basis for CRT prediction; **this case warrants immediate clinical verification**."

### A.8.3 DISCUSSION ON ANOMALY HANDLING

This case highlights a key safety feature of our framework. RCA does not attempt to algorithmically distinguish between a "genuine rare edge case" and "data noise/error," as both manifest as significant deviations from the learned distribution. Instead, it adopts a clinically responsible strategy:

1. **Detection:** Identifying the anomaly via the distribution check.
2. **Logging:** Explicitly noting the discrepancy in the explanation.
3. **Flagging:** Reducing the predictive weight of the outlier features and advising human review rather than forcing an unsupported high-confidence prediction.

This approach ensures that RCA remains robust to noise while providing necessary alerts for expert intervention.

## A.9 IMPLICIT SUB-POPULATION ANALYSIS VIA DISTRIBUTION-AWARE REASONING

While RCA does not include an explicit module for pre-defined sub-group analysis, its architecture enables it to organically identify and apply tailored logic to distinct sub-populations. This emergent capability stems from its core process of generating and verifying rules against the complete statistical

context of the data. By weighing evidence within the global distribution, the model implicitly learns context-dependent rules that behave differently for different data regimes.

A clear illustration of this is found in the final rule base generated for the CRT dataset. The model effectively stratified patients into three distinct physiological sub-populations based on their D-dimer levels, a key biomarker for thrombosis. For each group, it learned a unique diagnostic standard, demonstrating a nuanced understanding that goes beyond simple thresholding. The sub-populations and their corresponding logic are as follows:

- **High-Risk Sub-population (D-dimer $\geq$ 2.0):** For patients in this stratum, the model identified that the high D-dimer level is a strong intrinsic signal for CRT. Consequently, it learned a rule **(Rule 3)** requiring only **one additional moderate risk factor** (e.g., chemotherapy) to confirm a positive diagnosis. The model correctly inferred that the baseline risk in this group is already high, lowering the evidence bar for a final prediction.

- **Intermediate-Risk Sub-population (D-dimer 1.0–1.99):** In this group, the model recognized a greater degree of ambiguity, as the D-dimer level is elevated but not definitive. As a result, it learned a more conservative rule **(Rule 4)** that necessitates a higher burden of proof: **at least two moderate risk factors** must be present to predict CRT. This reflects an understanding that more corroborating evidence is needed to reach a conclusion in this ambiguous zone.

- **Low-Risk Sub-population (D-dimer $<$ 1.0):** For these patients, the model treated the low D-dimer level as strong contra-evidence for CRT. To overcome this negative signal, it learned its strictest rule **(Rule 5)**, which requires the accumulation of **three or more moderate risk factors** before making a positive prediction. This demonstrates the model's ability to demand a high volume of evidence to overturn a strong baseline indication.

As these rules demonstrate, RCA does not treat D-dimer as a biomarker with a single, universal threshold. Instead, by grounding its reasoning in the complete data distribution, it successfully identified distinct risk profiles and learned context-dependent diagnostic standards for each sub-population. This ability to implicitly stratify a population and apply tailored logic is a key feature of its data-driven approach, enhancing both the accuracy and the interpretability of its conclusions.

## A.10 AN EXPLANATION CASE

The patient has several risk factors contributing to catheter-related thrombosis. The GLR of 1.82 is within a borderline range, with some studies suggesting values above 1.8 may reflect subclinical inflammation associated with increased thrombotic risk. Although D-dimer is normal at 0.1 mg/L, it may not reliably exclude CRT, especially in cancer patients where baseline levels can vary. The patient has lung cancer, a known hypercoagulable condition that significantly increases VTE and CRT risk. He is undergoing thoracic therapy, which can cause vascular damage and immobility, further elevating thrombosis risk. The use of a PICC line adds mechanical endothelial injury as a contributory factor. Advanced age (74 years) is an independent risk factor for VTE due to decreased venous compliance and increased comorbidity burden. While platelet and hemoglobin levels are normal, and BMI is in the normal range, the combination of malignancy, PICC use, age, and thoracic therapy collectively heighten the likelihood of CRT despite the absence of chemotherapy and prior catheterization history.

**Cognitive Load**: 7
**Logical Argumentation**: 6
**Evidence-based Medicine**: 7
**Cognitive Biasing**: 5

The patient's GLR of 0.73 is low, which may reflect a less pronounced inflammatory state and could be associated with a lower thrombotic risk, although the exact threshold for GLR in CRT remains debated. D-dimer level is 0.33 mg/L, well within the normal range, indicating low current fibrin turnover and minimal evidence of active thrombosis. The patient is receiving chemotherapy and has a PICC line, both of which are established risk factors for CRT due to endothelial injury and hypercoagulability. However, the patient is 46 years old, which confers a moderate age-related risk, and her platelet count (209.0 × 10⁹/L) and hemoglobin (136.0 g/L) are within normal ranges, suggesting no significant thrombocytosis or anemia. BMI is 25.88, indicating overweight status, which is a mild risk factor for VTE. The patient is female with no prior catheterization or history of CRT, and has no known high-risk malignancies such as lung, gastric, or gynecologic cancers. Overall, despite the presence of chemotherapy and PICC, the absence of elevated D-dimer, low GLR, and lack of other strong risk factors suggest a low likelihood of CRT.

**Cognitive Load**: 9
**Logical Argumentation**: 10
**Evidence-based Medicine**: 10
**Cognitive Biasing**: 9

Figure 4: Two samples of explanation generated are provided to help better understand the criteria used in explanation experiment. Protective factors in the text are highlighted in blue, while risk factors are highlighted in red.

Figure 4 presents the explanations generated for two patients respectively. The explanation on the left scored lower, with scores in the four dimensions (Cognitive Load, Logical Argumentation, Evidence-Based Medicine, Cognitive Biasing) being 7, 6, 7, and 5 in sequence. Specifically:

Table 2: Expert Preference for RCA. The values represent the percentage of times clinicians (N=5) chose RCA's explanation as superior to the baseline's.

| Preference for RCA | CRT | Diabetes | Heart Disease |
|---|---|---|---|
| **Cognitive Load(CL)** | 82.7% | 81.3% | 84.7% |
| **Logical Argumentation(LA)** | 78.0% | 75.3% | 72.0% |
| **Evidence-Based Medicine(EBM)** | 72.7% | 74.7% | 73.3% |
| **Cognitive Biasing(CB)** | 86.7% | 89.3% | 86.0% |

- *Cognitive Biasing*: Although the text mentions supportive evidence for thrombotic risk and supplements features associated with lower risk, basically meeting the requirement of evidence comprehensiveness, it insufficiently analyzes the opposing factors. It merely lists evidence of lower risk without providing valid information such as the association between this evidence and reduced CRT risk.
- *Logical Argumentation*: It generally follows a "total-subtotal-total" logic: first clarifying that the patient has multiple CRT risk factors, then analyzing each factor one by one, and finally concluding that "the superposition of multiple factors increases risk", with clear expression. However, there are deficiencies in the logical connection between some factors, and the argumentative relationship between certain features and "CRT risk" is not smooth enough.
- *Evidence-Based Medicine*: The overall prediction is based on evidence and conforms to professional medical knowledge, but when listing features associated with lower risk, it lacks professional support.

  Based on the above three dimensions, although the text is generally logically coherent, with clear evidence-based core and evidence covering both positive and negative aspects, making it easy for doctors to read and understand; in the key part of the final summary, a large number of low-risk evidence contrary to the conclusion of "increased risk" are listed, which significantly increases the understanding pressure on doctors. Therefore, the overall Cognitive Load score is 7.

In contrast, the explanation on the right scored higher, with scores in the four dimensions being 9, 10, 10, and 9 in sequence:

- *Cognitive Biasing*: The text details two categories of features—"those increasing CRT risk" and "those reducing CRT risk", conducts in-depth analysis of each feature, and does not omit key influencing factors, with comprehensive and balanced evidence.
- *Logical Argumentation*: It adopts a "subtotal-total" logic for argumentation, clearly explaining how each feature directly or indirectly affects CRT risk, and finally draws the conclusion that "CRT risk is reduced", with a clear and coherent argumentation process.
- *Evidence-Based Medicine*: The analysis of risk factors conforms to recognized evidence-based conclusions, and the analysis of protective factors (i.e., factors reducing risk) also meets clinical testing standards; at the same time, it specifically mentions the controversy that "the threshold of GLR for CRT diagnosis has not been clearly defined", which not only respects evidence-based principles but also does not affect the professionalism of the overall argumentation.

  Overall, the understanding cost for professional doctors to read this text is extremely low. They can quickly grasp the core argument framework without the need to additionally verify the professionalism of the information, and there is basically no cognitive burden. Therefore, the Cognitive Load score is 9.

## A.11   Main Experiment Results

This section supplements Section 4.2. Table 3 presents the predictive performance of all methods in the main experiment. As can be seen from the table, RCA+Qwen2.5 and RCA+GPT-4.1 have nearly outperformed all baselines across the three metrics on the three datasets. Notably, on the heart disease dataset, the accuracy of RCA+GPT-4.1 is 20% higher than that of the top-performing baseline, accompanied by excellent MCC and F1-score. It is worth noting that LLM-based methods generally perform poorly on the CRT dataset and the Heart Disease dataset. However, o3-mini+Code achieves promising results on the CRT dataset, which indicates that code assistance enhances the reasoning ability of the o3-mini.

Tables 4- 6 present a detailed evaluation of explanation quality across all methods on the CRT, Diabetes, and Heart Disease datasets, respectively. The results offer two crucial insights into the performance of our RCA framework.

First, an analysis of the mean scores demonstrates that RCA-based methods consistently achieve the highest average quality. Across all three datasets, RCA+Qwen2.5 and RCA+GPT-4.1 outperform all baselines, securing top ranks in Cognitive Load (CL), Logical Argumentation (LA), Evidence-Based Medicine (EBM), and Cognitive Biasing (CB). This indicates that, on average, the explanations generated by RCA are judged by clinical experts to be clearer, more logical, and more reliable.

Second, the variance data reveals a critical advantage in consistency. The RCA framework consistently exhibits significantly lower variance compared to other LLM-based approaches, particularly the standalone Reasoning LLMs. For instance, in Table 4, the variance for RCA's CL and LA scores (e.g., 0.3-0.5) is often two to three times lower than that of the baseline LLMs (e.g., 0.9-1.3). This low variance signifies a high degree of inter-rater agreement and confirms that RCA's explanation quality is not just high, but also stable and predictable. In contrast, the high variance of standalone LLMs suggests their output quality is erratic, making them less trustworthy for critical applications. This dual achievement of delivering both the highest mean quality and the greatest consistency underscores the effectiveness of RCA's structured, data-grounded reasoning process.

To further validate these quantitative findings from a user-centric perspective, we conducted a head-to-head preference study, asking clinicians to directly choose the better explanation between RCA and the baselines. As shown in Table 2, the results provide a resounding endorsement of our framework. Across all datasets and criteria, clinicians overwhelmingly preferred RCA's explanations, with preference rates consistently exceeding 70% and often surpassing 80%. Notably, the highest preference for RCA was observed in Cognitive Biasing (CB), with rates reaching up to 89.3%. This is particularly significant because it shows that while generating a perfectly balanced argument is difficult for all models, clinicians find RCA's structured approach to be substantially more balanced and less biased compared to the unstructured outputs of other LLMs. This direct preference data serves as powerful, qualitative proof of the practical superiority of our framework.

Finally, a notable cross-model trend is that scores for Cognitive Biasing (CB) are generally the lowest among the four metrics for nearly all models. This suggests a common tendency for LLMs to generate one-sided justifications, highlighting the inherent difficulty of producing truly balanced clinical arguments.

## A.12 ROBUST EXPERIMENT RESULTS

Table 7 presents the predictive performance across all datasets, while Tables 8-10 respectively show the explanation scores for each of the three datasets. As can be seen from the table data, RCA+GPT-4.1 achieves the best performance in nearly all predictive metrics while maintaining competitive explanation scores, with RCA+Qwen2.5 performing slightly worse in prediction task. Such minor fluctuations indicate that RCAenables LLMs to maintain robustness against data noise. Furthermore, a comprehensive analysis of the aforementioned tables reveals no direct correlation between predictive performance and explanation scores: for instance, Qwen3-235B, which performs poorly in terms of accuracy, achieves impressive scores across all four explanation metrics. This phenomenon demonstrates that modern LLMs possess strong capabilities in generating explanatory texts, regardless of whether their predictive results are correct or not.

## A.13 SCALABILITY EXPERIMENT

To address the important question of real-world scalability, we conducted additional experiments on two larger and more diverse datasets. This evaluation was designed to test RCA's performance and robustness when applied to datasets that are an order of magnitude larger than those in our main experiments. The results of our scalability experiments are presented in Table 11.

The results in Table 11 strongly validate RCA's scalability.

- On the large-scale **Cardiovascular Disease (CD) dataset** (70,000 samples), **RCA+Qwen2.5** achieves the highest **Accuracy (0.7177)** and the highest **MCC (0.4476)**, outperforming all

Table 3: Accuracy, MCC and F1-score results in main experiment. RCA achieve almost best performance across all datasets, with Accuracy and MCC scores that rival all those of tree-based methods known for their effectiveness with tabular data. Moreover, it significantly outperforms LLM and LLM-based agents.

| Datasets | | CRT | | | Diabetes | | | Heart Disease | | |
|---|---|---|---|---|---|---|---|---|---|---|
| **Algorithm** | | **Acc** | **MCC** | **F1-score** | **Acc** | **MCC** | **F1-score** | **Acc** | **MCC** | **F1-score** |
| **Traditional** | **Lasso** | 0.5238 | 0.1261 | 0.1667 | 0.6867 | 0.3596 | 0.2857 | 0.5337 | -0.0065 | 0.1212 |
| **MLs** | **Catboost** | 0.7460 | 0.1472 | 0.2000 | 0.7590 | 0.5252 | 0.3478 | 0.5440 | 0.0182 | 0.1212 |
| **Neural-symbolic** | **LTN** | 0.5161 | 0.0085 | 0.1176 | 0.7635 | 0.5247 | 0.6946 | 0.6010 | -0.0386 | 0.2222 |
| **Networks** | **LNN** | 0.2581 | 0.1287 | 0.1481 | 0.7437 | 0.4974 | 0.6787 | 0.5389 | -0.0598 | 0.2393 |
| | **Qwen2.5-72B** | 0.2698 | -0.0073 | 0.1154 | 0.7470 | 0.4244 | 0.3333 | 0.3101 | -0.0870 | 0.0851 |
| **LLM-based** | **DeepSeekV3** | 0.2698 | 0.0111 | 0.1154 | 0.5542 | 0.4797 | 0.2222 | 0.2228 | -0.0383 | 0.1091 |
| **Method** | **DeepSeekV3.1** | 0.1587 | 0.1181 | 0.1212 | 0.7229 | 0.3687 | 0.5660 | 0.2073 | -0.1877 | 0.3377 |
| | **GPT-4.1** | 0.1746 | -0.0797 | 0.1034 | 0.7470 | 0.4187 | 0.3333 | 0.2280 | -0.0502 | 0.1091 |
| | **DeepSeek-R1** | 0.1429 | 0.0805 | 0.1290 | 0.7229 | 0.3615 | 0.3200 | 0.2124 | -0.1367 | 0.0741 |
| | **o3-mini** | 0.3175 | 0.0232 | 0.1224 | 0.5060 | -0.1352 | 0.0606 | 0.2176 | -0.0996 | 0.1091 |
| **Reasoning** | **o4-mini** | 0.2381 | 0.1220 | 0.1455 | 0.7470 | 0.4173 | 0.4878 | 0.2280 | -0.0114 | 0.3550 |
| **LLMs** | **Qwen3-30B** | 0.1270 | 0.0678 | 0.1270 | 0.7470 | 0.4187 | 0.5882 | 0.2228 | -0.0353 | 0.3534 |
| | **Qwen3-235B** | 0.2063 | 0.1063 | 0.1380 | 0.7108 | 0.3451 | 0.5556 | 0.2280 | -0.1430 | 0.3258 |
| | **GPT-5** | 0.2857 | 0.0024 | 0.1176 | 0.7470 | 0.4106 | 0.5333 | 0.2280 | -0.1602 | 0.3196 |
| **Medical LLM** | **Baichuan-M2** | 0.1587 | 0.0845 | 0.1311 | 0.6988 | 0.2772 | 0.4186 | 0.0725 | -0.7779 | 0.1095 |
| | **Qwen2.5-72B+Tools** | 0.2857 | 0.0023 | 0.1176 | 0.7470 | 0.4244 | 0.3333 | 0.3161 | -0.0797 | 0.0851 |
| | **DeepSeekV3+Tools** | 0.2381 | 0.1242 | 0.1429 | 0.7470 | 0.4187 | 0.3333 | 0.2280 | 0.0543 | 0.1404 |
| | **DeepSeekV3.1+Tools** | 0.1905 | -0.0650 | 0.1053 | 0.7349 | 0.4002 | 0.5926 | 0.2228 | -0.0352 | **0.3534** |
| | **GPT-4.1+Tools** | 0.2698 | -0.0073 | 0.1154 | 0.7590 | 0.4454 | 0.3478 | 0.2694 | -0.1203 | 0.0800 |
| **LLM-based** | **DeepSeek-R1+Tools** | 0.1111 | 0.0582 | 0.1250 | 0.7349 | 0.3873 | 0.3200 | 0.2435 | 0.0351 | 0.1111 |
| **Agent** | **o3-mini + Tools** | 0.2063 | -0.0577 | 0.1071 | 0.4359 | -0.2441 | 0.0000 | 0.2265 | -0.1003 | 0.1091 |
| | **o4-mini+Tools** | 0.2381 | 0.1220 | 0.1455 | 0.7108 | 0.3169 | 0.4783 | 0.2228 | -0.0721 | 0.3478 |
| | **Qwen3-30B+Tools** | 0.0952 | 0.0471 | 0.1231 | 0.7108 | 0.3169 | 0.4783 | 0.2073 | -0.1877 | 0.3377 |
| | **Qwen3-235B+Tools** | 0.1905 | 0.0946 | 0.1091 | 0.6987 | 0.3223 | 0.5455 | 0.2073 | -0.2282 | 0.3139 |
| | **o3-mini+Code** | 0.5079 | 0.1179 | 0.1622 | 0.6747 | 0.3861 | 0.2857 | 0.2850 | -0.0594 | 0.1176 |
| **RCA** | **RCA+Qwen2.5** | 0.7778 | **0.1739** | **0.2222** | **0.7831** | **0.5406** | **0.7097** | 0.5647 | 0.0547 | 0.1290 |
| | **RCA+GPT-4.1** | **0.8730** | 0.1373 | 0.2000 | 0.7470 | 0.4244 | 0.6038 | **0.7461** | **0.1493** | 0.2898 |

30 baseline methods, including traditional ML, LLM-based agents, and the newly added neural-symbolic models.

- On the expanded **CRT_ex dataset** (1,891 samples), RCA methods again demonstrate top-tier performance. **RCA+GPT-4.1** achieves the highest **Accuracy (0.8335)**, while **RCA+Qwen2.5** secures the best **MCC (0.1053)** and **F1-score (0.2338)**.

These experiments confirm that RCA's architecture, which fosters a deep, experience-driven understanding of data, is not limited to small datasets. It effectively scales to larger and more diverse clinical prediction tasks, maintaining its state-of-the-art performance in both accuracy and robustness.

## A.14 ABLATION STUDY

As shown in Table 12, removing any module from RCA leads to a significant drop in the four explanation scores, which indicates that each module plays an irreplaceable role in the process of explanatory reasoning. As demonstrated in Tables 1 and 12, the removal of the data distribution resulted in a marked decrease in predictive and explanatory performance, yet the framework maintained a respectable level of efficacy.

Table 4: Explanation experiment results on CRT dataset

| Metrics | | CL | | LA | | EBM | | CB | |
|---|---|---|---|---|---|---|---|---|---|
| Algorithms | | Mean. | Var. | Mean. | Var. | Mean. | Var. | Mean. | Var. |
| Traditional | Lasso | 7.60 | 0.39 | 8.33 | 0.36 | 7.31 | 0.63 | 7.21 | 0.56 |
| MLs | Catboost | 7.39 | 0.50 | 8.13 | 0.53 | 7.31 | 0.78 | 7.08 | 0.71 |
| Neural-symbolic | LTN | 7.47 | 0.23 | 8.25 | 0.32 | 8.05 | 0.61 | 7.27 | 0.59 |
| Networks | LNN | 7.51 | 0.28 | 8.35 | 0.33 | 7.63 | 0.74 | 7.06 | 0.54 |
| LLM-based Methods | Qwen2.5-72B | 7.92 | 0.96 | 8.16 | 1.23 | 8.25 | 1.31 | 7.14 | 1.07 |
| | DeepSeek-V3 | 7.98 | 0.68 | 8.27 | 0.73 | 8.22 | 0.83 | 7.19 | 0.70 |
| | DeepSeek-V3.1 | 7.41 | 0.89 | 8.11 | 1.02 | 8.13 | 1.28 | 6.73 | 1.02 |
| | GPT-4.1 | 7.79 | 0.69 | 8.25 | 0.72 | 8.30 | 1.13 | 7.31 | 0.88 |
| Reasoning LLMs | DeepSeek-R1 | 7.82 | 1.03 | 7.97 | 1.29 | 8.27 | 1.60 | 7.25 | 1.07 |
| | o3-mini | 6.84 | 0.96 | 7.22 | 1.23 | 6.24 | 1.62 | 6.19 | 1.21 |
| | o4-mini | 7.56 | 1.18 | 8.06 | 1.36 | 8.25 | 1.80 | 6.81 | 1.49 |
| | Qwen3-30B | 7.87 | 0.97 | 8.49 | 1.06 | 8.41 | 1.28 | 7.06 | 1.19 |
| | Qwen3-235B | 8.22 | 0.94 | 8.81 | 1.15 | 8.71 | 1.25 | 7.49 | 1.08 |
| | GPT-5 | 7.63 | 0.97 | 8.22 | 1.06 | 8.40 | 1.32 | 7.05 | 1.05 |
| Medical LLM | Baichuan-M2 | 7.82 | 0.54 | 8.38 | 0.52 | 8.32 | 0.75 | 6.96 | 0.55 |
| LLM-Based Agents | Qwen2.5-72B+Tools | 7.87 | 0.75 | 8.35 | 0.92 | 8.29 | 1.18 | 7.16 | 0.95 |
| | DeepSeek-V3.1+Tools | 7.87 | 0.69 | 8.41 | 0.93 | 8.35 | 1.22 | 6.51 | 1.01 |
| | GPT-4.1+Tools | 7.89 | 0.48 | 8.30 | 0.54 | 8.32 | 0.76 | 7.35 | 0.67 |
| | DeepSeek-R1+Tools | 7.87 | 1.03 | 8.00 | 1.29 | 8.17 | 1.60 | 7.14 | 1.07 |
| | o3-mini+Tools | 6.22 | 0.91 | 7.30 | 1.07 | 6.20 | 1.31 | 6.16 | 1.15 |
| | o4-mini+Tools | 7.29 | 1.05 | 7.87 | 1.21 | 8.05 | 1.60 | 5.73 | 0.59 |
| | Qwen3-30B+Tools | 7.79 | 0.86 | 8.63 | 0.99 | 8.41 | 1.13 | 6.57 | 0.83 |
| | Qwen3-235B+Tools | 8.13 | 0.88 | 8.86 | 0.84 | 8.86 | 1.22 | 7.11 | 0.94 |
| | o3-mini+Code | 6.93 | 1.04 | 7.60 | 1.19 | 6.68 | 1.57 | 6.17 | 1.23 |
| RCA | RCA+Qwen2.5 | **8.24** | 0.42 | **8.89** | 0.54 | 8.47 | 0.92 | 7.61 | 0.74 |
| | RCA+GPT-4.1 | 8.16 | 0.34 | 8.59 | 0.34 | **8.87** | 0.76 | **7.62** | 0.56 |

Table 5: Explanation experiment results on Diabetes dataset

| Metrics | | CL | | LA | | EBM | | CB | |
|---|---|---|---|---|---|---|---|---|---|
| Algorithms | | Mean. | Var. | Mean. | Var. | Mean. | Var. | Mean. | Var. |
| Traditional | Lasso | 7.83 | 0.32 | 8.39 | 0.26 | 8.33 | 0.30 | 6.24 | 0.40 |
| MLs | Catboost | 7.71 | 0.30 | 8.27 | 0.24 | 8.19 | 0.28 | 6.12 | 0.36 |
| Neural-symbolic | LTN | 7.68 | 0.35 | 8.22 | 0.29 | 8.28 | 0.33 | 6.18 | 0.43 |
| Networks | LNN | 7.82 | 0.33 | 8.35 | 0.27 | 8.45 | 0.31 | 6.28 | 0.39 |
| LLM-based Methods | Qwen2.5-72B | 7.61 | 0.92 | 8.20 | 0.80 | 8.10 | 0.87 | 5.98 | 1.02 |
| | DeepSeek-V3 | 7.55 | 0.99 | 8.04 | 0.87 | 8.29 | 0.94 | 5.99 | 1.09 |
| | DeepSeek-V3.1 | 7.85 | 0.95 | 8.23 | 0.83 | 8.12 | 0.90 | 6.04 | 1.05 |
| | GPT-4.1 | 7.84 | 0.95 | 8.14 | 0.83 | 8.33 | 0.90 | 5.85 | 1.05 |
| Reasoning LLMs | DeepSeek-R1 | 7.73 | 1.02 | 8.12 | 0.90 | 7.99 | 0.97 | 5.88 | 1.12 |
| | o3-mini | 7.67 | 1.05 | 8.12 | 0.93 | 8.16 | 1.00 | 5.89 | 1.15 |
| | o4-mini | 7.82 | 1.00 | 8.13 | 0.88 | 8.08 | 0.95 | 6.13 | 1.10 |
| | Qwen3-30B | 7.94 | 0.98 | 8.43 | 0.86 | 8.51 | 0.93 | 6.72 | 1.08 |
| | Qwen3-235B | 7.84 | 0.89 | 8.41 | 0.77 | 8.31 | 0.84 | 6.30 | 0.99 |
| | GPT-5 | 7.66 | 1.05 | 8.24 | 0.93 | 8.53 | 1.00 | 6.24 | 1.15 |
| Medical LLM | Baichuan-M2 | 7.79 | 0.40 | 8.18 | 0.33 | 8.52 | 0.37 | 6.05 | 0.46 |
| LLM-Based Agents | Qwen2.5-72B+Tools | 7.86 | 0.55 | 8.29 | 0.43 | 8.66 | 0.50 | 6.43 | 0.65 |
| | DeepSeek-V3.1+Tools | 7.83 | 0.62 | 8.23 | 0.50 | 8.55 | 0.56 | 6.41 | 0.72 |
| | GPT-4.1+Tools | 7.81 | 0.56 | 8.25 | 0.44 | 8.01 | 0.51 | 6.12 | 0.66 |
| | DeepSeek-R1+Tools | 7.73 | 0.75 | 8.24 | 0.63 | 8.11 | 0.70 | 5.93 | 0.82 |
| | o3-mini+Tools | 7.62 | 0.79 | 8.25 | 0.66 | 8.13 | 0.73 | 5.90 | 0.86 |
| | o4-mini+Tools | 7.85 | 0.82 | 8.27 | 0.69 | 8.12 | 0.76 | 6.03 | 0.89 |
| | Qwen3-30B+Tools | 7.82 | 0.73 | 8.13 | 0.61 | 8.08 | 0.68 | 6.13 | 0.80 |
| | Qwen3-235B+Tools | 7.88 | 0.69 | 8.33 | 0.57 | 8.11 | 0.64 | 6.38 | 0.76 |
| | o3-mini+Code | 7.17 | 1.09 | 7.60 | 0.97 | 6.76 | 1.04 | 5.49 | 1.19 |
| RCA | RCA+Qwen2.5 | **8.13** | 0.36 | **8.57** | 0.51 | **8.74** | 0.54 | 6.43 | 0.43 |
| | RCA+GPT-4.1 | 8.03 | 0.43 | 8.38 | 0.49 | 8.63 | 0.62 | **6.94** | 0.40 |

## A.15 EXAMPLE OF GLOBAL DATA DISTRIBUTION

To make the concept of "global data distribution" more concrete, this section provides a detailed example of the statistical summary provided to the RCA framework (specifically to $M_{pred}$ and $M_{chk}$). This information, first extracted in Section 3.1, serves as the statistical grounding for all reasoning, rule generation, and rule validation.

The following is the data distribution ($\mathcal{D}_{train}$) for the key features of the Diabetes dataset.

Table 6: Explanation experiment results on Heart Disease dataset

| Metrics | | CL | | LA | | EBM | | CB | |
|---|---|---|---|---|---|---|---|---|---|
| Algorithms | | Mean. | Var. | Mean. | Var. | Mean. | Var. | Mean. | Var. |
| Traditional | Lasso | 7.60 | 0.35 | 8.27 | 0.28 | 8.63 | 0.32 | 5.94 | 0.42 |
| MLs | Catboost | 7.50 | 0.33 | 8.21 | 0.25 | 8.72 | 0.29 | 5.94 | 0.38 |
| Neural-symbolic | LTN | 7.45 | 0.38 | 8.15 | 0.31 | 8.35 | 0.35 | 6.20 | 0.45 |
| Networks | LNN | 7.65 | 0.36 | 8.32 | 0.29 | 8.58 | 0.33 | 6.35 | 0.41 |
| LLM-based | Qwen2.5-72B | 7.63 | 0.95 | 8.33 | 0.82 | 8.77 | 0.89 | 6.14 | 1.05 |
| Methods | DeepSeek-V3 | 7.52 | 1.02 | 8.25 | 0.89 | 8.62 | 0.96 | 6.05 | 1.12 |
| | DeepSeek-V3.1 | 7.35 | 1.02 | 8.04 | 0.89 | 8.55 | 0.96 | 5.69 | 1.12 |
| | GPT-4.1 | 7.45 | 0.98 | 8.08 | 0.85 | 8.64 | 0.92 | 5.77 | 1.08 |
| | DeepSeek-R1 | 7.40 | 1.05 | 7.98 | 0.92 | 8.59 | 0.99 | 5.75 | 1.15 |
| | o3-mini | 7.33 | 1.08 | 7.89 | 0.95 | 8.84 | 1.02 | 5.46 | 1.18 |
| Reasoning | o4-mini | 7.31 | 1.03 | 7.83 | 0.90 | 8.73 | 0.97 | 5.38 | 1.13 |
| LLMs | Qwen3-30B | 7.41 | 1.01 | 8.06 | 0.88 | 8.49 | 0.95 | 5.82 | 1.11 |
| | Qwen3-235B | 7.67 | 0.92 | 8.34 | 0.79 | 8.84 | 0.86 | 6.36 | 1.02 |
| | GPT-5 | 7.36 | 1.08 | 8.07 | 0.95 | 8.59 | 1.02 | 5.62 | 1.18 |
| Medical LLM | Baichuan-M2 | 7.28 | 0.42 | 8.08 | 0.35 | 8.25 | 0.39 | 5.85 | 0.48 |
| | Qwen2.5-72B+Tools | 7.63 | 0.58 | 8.34 | 0.45 | 8.74 | 0.52 | 6.00 | 0.68 |
| | DeepSeek-V3.1+Tools | 7.37 | 0.65 | 7.97 | 0.52 | 8.46 | 0.58 | 5.72 | 0.75 |
| | GPT-4.1+Tools | 7.42 | 0.59 | 8.01 | 0.46 | 8.58 | 0.53 | 5.70 | 0.69 |
| LLM-Based | DeepSeek-R1+Tools | 7.38 | 0.78 | 7.87 | 0.65 | 8.55 | 0.72 | 5.73 | 0.85 |
| Agents | o3-mini+Tools | 7.31 | 0.82 | 7.90 | 0.68 | 8.76 | 0.75 | 5.48 | 0.89 |
| | o4-mini+Tools | 7.35 | 0.85 | 7.96 | 0.71 | 8.51 | 0.78 | 5.66 | 0.92 |
| | Qwen3-30B+Tools | 7.39 | 0.76 | 7.94 | 0.63 | 8.56 | 0.70 | 5.73 | 0.83 |
| | Qwen3-235B+Tools | 7.58 | 0.72 | 8.28 | 0.59 | 8.91 | 0.66 | 6.20 | 0.79 |
| | o3-mini+Code | 7.28 | 1.02 | 7.88 | 0.99 | 8.67 | 1.06 | 5.38 | 1.12 |
| RCA | RCA+Qwen2.5 | 7.62 | 0.28 | 8.47 | 0.33 | **8.94** | 0.49 | 6.18 | 0.68 |
| | RCA+GPT-4.1 | **7.74** | 0.35 | **8.53** | 0.42 | 8.79 | 0.55 | **6.42** | 0.53 |

Table 7: Accuracy, MCC and F1-score results in robust experiment. RCA achieve almost best performance across all datasets, with Accuracy and MCC scores that rival all those of tree-based methods known for their effectiveness with tabular data. Moreover, it significantly outperforms LLM and LLM-based agents.

| Datasets | | w/o GLR | | | Missing | | | Abnormal | | |
|---|---|---|---|---|---|---|---|---|---|---|
| Algorithm | | Acc | MCC | F1-score | Acc | MCC | F1-score | Acc | MCC | F1-score |
| Traditional | Lasso | 0.5238 | 0.1261 | 0.1667 | 0.5079 | 0.1179 | 0.1622 | 0.6667 | 0.0927 | 0.1600 |
| MLs | Catboost | 0.6825 | 0.1021 | 0.1667 | 0.5079 | 0.0041 | 0.1143 | 0.5238 | 0.0124 | 0.1176 |
| LLM-based | Qwen2.5 | 0.1277 | 0.0678 | 0.1270 | 0.3333 | -0.0921 | 0.0870 | 0.3175 | -0.1021 | 0.0851 |
| Method | DeepSeekV3 | 0.2539 | -0.0152 | 0.1132 | 0.2857 | 0.0048 | 0.1176 | 0.2063 | -0.1082 | 0.1071 |
| | DeepSeekV3.1 | 0.2063 | -0.0517 | 0.1071 | 0.1429 | -0.1151 | 0.1000 | 0.2073 | -0.1877 | 0.3377 |
| | GPT-4.1 | 0.2381 | -0.0280 | 0.1111 | 0.2222 | -0.0395 | 0.1091 | 0.1905 | 0.0993 | 0.1356 |
| | DeepSeek-R1 | 0.0793 | 0.0331 | 0.1212 | 0.1429 | 0.0764 | 0.1290 | 0.0635 | 0.0000 | 0.0000 |
| | o3-mini | 0.3968 | -0.0315 | 0.0952 | 0.3175 | 0.1001 | 0.1569 | 0.2063 | 0.0971 | 0.1379 |
| Reasoning | o4-mini | 0.2222 | -0.0394 | 0.1091 | 0.2222 | 0.1151 | 0.1429 | 0.2280 | -0.0114 | **0.3550** |
| LLMs | Qwen3-30B | 0.1270 | 0.0678 | 0.1270 | 0.1269 | 0.0678 | 0.1270 | 0.2228 | -0.0353 | 0.3534 |
| | Qwen3-235B | 0.1587 | 0.0844 | 0.1311 | 0.2063 | 0.1063 | 0.1379 | 0.2280 | -0.1430 | 0.3258 |
| | GPT-5 | 0.3333 | -0.0921 | 0.0870 | 0.3333 | 0.0293 | 0.1250 | 0.2381 | -0.0281 | 0.1111 |
| | Qwen2.5-72B+Tools | 0.2539 | 0.1263 | 0.1455 | 0.3175 | -0.1021 | 0.0851 | 0.2222 | 0.1131 | 0.1404 |
| | DeepSeekV3+Tools | 0.1746 | -0.2155 | 0.0714 | 0.2222 | 0.1267 | 0.1404 | 0.1905 | 0.1089 | 0.1356 |
| | DeepSeekV3.1+Tools | 0.1905 | 0.0650 | 0.1053 | 0.2540 | 0.1263 | 0.1455 | 0.1429 | 0.0764 | 0.1290 |
| | GPT-4.1+Tools | 0.2222 | -0.0395 | 0.1091 | 0.2857 | -0.1235 | 0.0816 | 0.1905 | -0.0650 | 0.1053 |
| LLM-based | DeepSeek-R1+Tools | 0.1111 | 0.0582 | 0.1250 | 0.1746 | 0.0921 | 0.1333 | 0.0793 | 0.0331 | 0.1212 |
| Agent | o3-mini + Tools | 0.3175 | -0.1099 | 0.0851 | 0.2063 | 0.0971 | 0.1379 | 0.1746 | -0.6240 | 0.0851 |
| | o4-mini+Tools | 0.2381 | -0.0281 | 0.1111 | 0.2222 | 0.1131 | 0.1404 | 0.2063 | 0.0989 | 0.1091 |
| | Qwen3-30B+Tools | 0.1587 | 0.0845 | 0.1311 | 0.1429 | 0.0764 | 0.1290 | 0.1746 | 0.0921 | 0.1333 |
| | Qwen3-235B+Tools | 0.1429 | 0.0764 | 0.1290 | 0.2381 | 0.1198 | 0.1429 | 0.1905 | 0.0993 | 0.1356 |
| | o3-mini+Code | 0.4603 | -0.1343 | 0.0556 | 0.5873 | 0.0462 | 0.1333 | 0.4286 | -0.0919 | 0.1000 |
| RCA | RCA+Qwen2.5 | **0.7143** | 0.1235 | 0.1818 | 0.6984 | 0.1126 | 0.1739 | 0.7302 | 0.1350 | 0.1904 |
| | RCA+GPT-4.1 | 0.6507 | **0.1979** | **0.2143** | **0.8889** | **0.1644** | **0.2222** | **0.8254** | **0.2232** | 0.2666 |

Table 8: Explanation experiment results on CRT dataset w/o GLR

|  |  | CL | LA | EBM | CB |
|---|---|---|---|---|---|
| **Traditional MLs** | **Lasso** | 7.79 | 8.52 | 8.27 | 6.94 |
|  | **Catboost** | 7.78 | 8.48 | 8.41 | 6.82 |
| **LLM-Based Methods** | **Qwen2.5-72B** | 7.68 | 8.41 | 8.32 | 6.43 |
|  | **DeepSeek-V3** | 7.71 | 8.41 | 8.42 | 6.77 |
|  | **DeepSeek-V3.1** | 7.83 | 8.56 | 8.48 | 6.71 |
|  | **GPT-4.1** | 7.71 | 8.43 | 8.52 | 6.51 |
| **Reasoning LLMs** | **DeepSeek-R1** | 7.98 | 8.76 | 8.63 | 6.82 |
|  | **o3-mini** | 7.56 | 8.22 | 8.27 | 6.13 |
|  | **o4-mini** | 7.37 | 8.08 | 8.21 | 6.03 |
|  | **Qwen3-30B** | 7.91 | 8.49 | 8.67 | 6.76 |
|  | **Qwen3-235B** | 8.22 | 8.75 | 8.86 | 7.12 |
|  | **GPT-5** | 7.63 | 8.44 | 8.14 | 6.81 |
| **LLM-Based Agents** | **Qwen2.5-72B+Tools** | 7.76 | 8.48 | 8.56 | 6.52 |
|  | **DeepSeek-V3+Tools** | 7.73 | 8.52 | 8.37 | 6.73 |
|  | **DeepSeek-V3.1+Tools** | 7.84 | 8.49 | 8.51 | 6.69 |
|  | **GPT-4.1+Tools** | 7.73 | 8.56 | 8.76 | 6.42 |
|  | **DeepSeek-R1+Tools** | 8.11 | 8.79 | 8.70 | 6.84 |
|  | **o3-mini+Tools** | 7.46 | 8.05 | 7.98 | 6.05 |
|  | **o4-mini+Tools** | 7.49 | 8.21 | 8.22 | 6.41 |
|  | **Qwen3-30B+Tools** | 7.78 | 8.56 | 8.57 | 6.49 |
|  | **Qwen3-235B+Tools** | 8.27 | 8.92 | 9.06 | 7.16 |
|  | **o3-mini+Code** | 7.90 | 8.54 | 8.68 | 6.67 |
| **RCA** | **RCA + Qwen2.5** | **8.35** | **8.97** | 8.99 | **7.26** |
|  | **RCA + GPT-4.1** | 8.28 | 8.75 | **9.13** | 7.21 |

NO DIABETES

- **Glucose**: Mean: 110.10, 5th Percentile: 75.00, 25th Percentile: 93.00, 50th Percentile: 107.00, 75th Percentile: 124.00, 95th Percentile: 157.00

- **BloodPressure**: Mean: 67.62, 5th Percentile: 44.00, 25th Percentile: 62.00, 50th Percentile: 70.00, 75th Percentile: 78.00, 95th Percentile: 90.00

- **SkinThickness**: Mean: 20.06, 5th Percentile: 0.00, 25th Percentile: 0.00, 50th Percentile: 22.00, 75th Percentile: 31.00, 95th Percentile: 42.00

- **Insulin**: Mean: 68.45, 5th Percentile: 0.00, 25th Percentile: 0.00, 50th Percentile: 40.00, 75th Percentile: 105.00, 95th Percentile: 265.00

- **BMI**: Mean: 30.33, 5th Percentile: 20.40, 25th Percentile: 25.40, 50th Percentile: 29.90, 75th Percentile: 35.40, 95th Percentile: 42.40

- **DiabetesPedigreeFunction**: Mean: 0.42, 5th Percentile: 0.14, 25th Percentile: 0.23, 50th Percentile: 0.32, 75th Percentile: 0.55, 95th Percentile: 0.95

- **Age**: Mean: 31.16, 5th Percentile: 21.00, 25th Percentile: 23.00, 50th Percentile: 26.00, 75th Percentile: 37.00, 95th Percentile: 58.00

- **Pregnancies**:
  - 1 pregnancy: 220
  - 2 pregnancy: 196
  - 0 pregnancy: 168
  - 3 pregnancy: 114
  - 4 pregnancy: 98
  - 5 pregnancy: 66
  - 6 pregnancy: 61

Table 9: Explanation experiment results on CRT dataset missing 10%

|  |  | CL | LA | EBM | CB |
|---|---|---|---|---|---|
| **Traditional MLs** | **Lasso** | 7.89 | 8.24 | 8.13 | 6.90 |
|  | **Catboost** | 7.91 | 8.33 | 8.27 | 6.93 |
| **LLM-Based Methods** | **Qwen2.5-72B** | 7.69 | 8.37 | 8.02 | 6.39 |
|  | **DeepSeek-V3** | 7.65 | 8.31 | 8.11 | 6.28 |
|  | **DeepSeek-V3.1** | 7.78 | 8.29 | 8.17 | 6.32 |
|  | **GPT-4.1** | 7.52 | 8.22 | 8.19 | 6.52 |
| **Reasoning LLMs** | **DeepSeek-R1** | 7.79 | 8.24 | 8.21 | 6.75 |
|  | **o3-mini** | 7.33 | 7.92 | 7.71 | 5.87 |
|  | **o4-mini** | 7.38 | 7.95 | 7.76 | 5.97 |
|  | **Qwen3-30B** | 7.68 | 8.41 | 8.21 | 6.59 |
|  | **Qwen3-235B** | 7.95 | 8.65 | 8.70 | 7.03 |
|  | **GPT-5** | 7.82 | 8.30 | 8.14 | 6.73 |
| **LLM-Based Agents** | **Qwen2.5-72B+Tools** | 7.63 | 8.25 | 8.05 | 6.38 |
|  | **DeepSeek-V3+Tools** | 7.58 | 8.22 | 8.17 | 6.08 |
|  | **DeepSeek-V3.1+Tools** | 7.72 | 8.37 | 8.13 | 6.33 |
|  | **GPT-4.1+Tools** | 7.60 | 8.37 | 8.26 | 6.49 |
|  | **DeepSeek-R1+Tools** | 7.62 | 8.35 | 8.25 | 6.67 |
|  | **o3-mini+Tools** | 7.41 | 8.07 | 7.81 | 6.03 |
|  | **o4-mini+Tools** | 7.44 | 7.94 | 8.14 | 6.08 |
|  | **Qwen3-30B+Tools** | 7.83 | 8.43 | 8.46 | 6.79 |
|  | **Qwen3-235B+Tools** | 8.11 | 8.78 | 8.49 | 7.05 |
|  | **o3-mini+Code** | 7.54 | 8.19 | 8.13 | 6.21 |
| **RCA** | **RCA + Qwen2.5** | 8.09 | 8.79 | 8.67 | **7.13** |
|  | **RCA + GPT-4.1** | **8.17** | **8.93** | **8.72** | 7.01 |

- – 7 pregnancy: 45
- – 8 pregnancy: 34
- – 10 pregnancy: 26
- – 9 pregnancy: 24
- – 12 pregnancy: 11
- – 11 pregnancy: 10
- – 13 pregnancy: 8

## DIABETES

- **Glucose**: Mean: 143.29, 5th Percentile: 97.00, 25th Percentile: 120.75, 50th Percentile: 142.00, 75th Percentile: 168.00, 95th Percentile: 193.00

- **BloodPressure**: Mean: 70.52, 5th Percentile: 0.00, 25th Percentile: 66.00, 50th Percentile: 74.00, 75th Percentile: 82.00, 95th Percentile: 94.00

- **SkinThickness**: Mean: 21.89, 5th Percentile: 0.00, 25th Percentile: 0.00, 50th Percentile: 26.00, 75th Percentile: 36.00, 95th Percentile: 46.00

- **Insulin**: Mean: 102.00, 5th Percentile: 0.00, 25th Percentile: 0.00, 50th Percentile: 0.00, 75th Percentile: 168.00, 95th Percentile: 395.65

- **BMI**: Mean: 35.39, 5th Percentile: 26.38, 25th Percentile: 30.90, 50th Percentile: 34.30, 75th Percentile: 38.50, 95th Percentile: 48.30

- **DiabetesPedigreeFunction**: Mean: 0.54, 5th Percentile: 0.14, 25th Percentile: 0.26, 50th Percentile: 0.44, 75th Percentile: 0.73, 95th Percentile: 1.22

- **Age**: Mean: 36.90, 5th Percentile: 22.00, 25th Percentile: 28.00, 50th Percentile: 36.00, 75th Percentile: 44.00, 95th Percentile: 57.05

Table 10: Explanation experiment results on CRT dataset abnormal 10%

| | | CL | LA | EBM | CB |
|---|---|---|---|---|---|
| **Traditional MLs** | **Lasso** | 7.52 | 8.27 | 8.16 | 6.37 |
| | **Catboost** | 7.49 | 8.21 | 8.29 | 6.21 |
| **LLM-Based Methods** | **Qwen2.5-72B** | 7.56 | 8.25 | 8.32 | 6.73 |
| | **DeepSeek-V3** | 7.48 | 8.29 | 8.30 | 6.02 |
| | **DeepSeek-V3.1** | 7.46 | 8.30 | 8.37 | 6.27 |
| | **GPT-4.1** | 7.32 | 8.30 | 8.11 | 6.37 |
| **Reasoning LLMs** | **DeepSeek-R1** | 7.83 | 8.63 | 8.24 | 6.81 |
| | **o3-mini** | 7.25 | 7.86 | 7.89 | 5.75 |
| | **o4-mini** | 7.13 | 7.87 | 7.84 | 5.59 |
| | **Qwen3-30B** | 7.62 | 8.29 | 7.90 | 6.43 |
| | **Qwen3-235B** | 7.89 | 8.62 | 8.34 | 6.97 |
| | **GPT-5** | 7.60 | 8.21 | 8.13 | 6.59 |
| **LLM-Based Agents** | **Qwen2.5-72B+Tools** | 7.59 | 8.32 | 8.32 | 6.83 |
| | **DeepSeek-V3+Tools** | 7.52 | 8.31 | 8.22 | 6.19 |
| | **DeepSeek-V3.1+Tools** | 7.48 | 8.30 | 8.08 | 6.21 |
| | **GPT-4.1+Tools** | 7.29 | 8.31 | 8.08 | 6.25 |
| | **DeepSeek-R1+Tools** | 7.86 | 8.54 | 8.26 | 6.92 |
| | **o3-mini+Tools** | 7.22 | 7.92 | 7.98 | 5.71 |
| | **o4-mini+Tools** | 7.19 | 7.94 | 7.95 | 5.90 |
| | **Qwen3-30B+Tools** | 7.44 | 8.14 | 8.22 | 6.44 |
| | **Qwen3-235B+Tools** | 7.90 | 8.68 | 8.65 | 6.93 |
| | **o3-mini+Code** | 7.52 | 8.33 | 8.0 | 6.84 |
| **RCA** | **RCA + Qwen2.5** | 7.95 | 8.75 | 8.53 | 6.88 |
| | **RCA + GPT-4.1** | **8.03** | **8.84** | **8.69** | **6.96** |

- **Pregnancies**:
  - 0 pregnancy: 91
  - 1 pregnancy: 66
  - 3 pregnancy: 59
  - 4 pregnancy: 55
  - 7 pregnancy: 55
  - 8 pregnancy: 45
  - 5 pregnancy: 43
  - 2 pregnancy: 39
  - 9 pregnancy: 34
  - 6 pregnancy: 32
  - 10 pregnancy: 19
  - 11 pregnancy: 14
  - 13 pregnancy: 11
  - 12 pregnancy: 9
  - 14 pregnancy: 5
  - 17 pregnancy: 2
  - 15 pregnancy: 1

## A.16 PROMPT TEMPLATES

In this section we will provide all prompt templates used in RCA.

Table 11: Scalability experiment results on CRT_ex dataset and Cardiovascular Disease dataset.

| Datasets | | | CRT_ex | | | CD | | |
|---|---|---|---|---|---|---|---|---|
| Algorithm | | Acc | MCC | F1-score | Acc | MCC | F1-score |
| **Traditional** | **Lasso** | 0.4634 | 0.0456 | 0.2056 | 0.5086 | 0.0634 | 0.6683 |
| **MLs** | **Catboost** | 0.6470 | 0.0938 | 0.2287 | 0.5779 | 0.2495 | 0.6965 |
| **Neural-symbolic** | **LTN** | 0.5680 | 0.0006 | 0.1745 | 0.6850 | 0.3724 | 0.6998 |
| **Networks** | **LNN** | 0.7859 | -0.0245 | 0.0964 | 0.6836 | 0.3728 | 0.6548 |
| | **Qwen2.5-72B** | 0.3789 | 0.0076 | 0.1848 | 0.6450 | 0.3096 | 0.5697 |
| **LLM-based** | **DeepSeekV3** | 0.2398 | 0.0699 | 0.2149 | 0.6771 | 0.3597 | 0.6463 |
| **Method** | **DeepSeekV3.1** | 0.6803 | -0.1074 | 0.0667 | 0.4171 | -0.2260 | 0.5650 |
| | **GPT-4.1** | 0.2778 | 0.0530 | 0.2085 | 0.6871 | 0.3781 | 0.6667 |
| | **DeepSeek-R1** | 0.0200 | -0.9087 | 0.0077 | 0.0650 | -0.8700 | 0.0603 |
| **Reasoning** | **o3-mini** | 0.3081 | 0.0585 | 0.2106 | 0.7093 | 0.4188 | **0.7044** |
| **LLMs** | **Qwen3-30B** | 0.5509 | -0.1892 | 0.0484 | 0.4429 | -0.1144 | 0.4323 |
| | **Qwen3-235B** | 0.5109 | 0.0408 | 0.2017 | 0.4300 | -0.1404 | 0.4071 |
| | **GPT-5** | 0.3787 | -0.1780 | 0.1067 | 0.6921 | 0.3843 | 0.6901 |
| **Medical LLM** | **Baichuan-M2** | 0.1713 | 0.0598 | 0.2104 | 0.6800 | 0.3601 | 0.6836 |
| | **Qwen2.5-72B+Tools** | 0.3245 | -0.0321 | 0.1801 | 0.6807 | 0.3632 | 0.6642 |
| | **DeepSeekV3+Tools** | 0.8183 | -0.0199 | 0.0571 | 0.6779 | 0.3558 | 0.6810 |
| | **DeepSeekV3.1+Tools** | 0.2312 | -0.2295 | 0.1236 | 0.6600 | 0.3210 | 0.6730 |
| **LLM-based** | **GPT-4.1+Tools** | 0.5050 | 0.0748 | 0.2192 | 0.5929 | 0.2106 | 0.4673 |
| **Agents** | **DeepSeek-R1+Tools** | 0.1418 | 0.0210 | 0.2004 | 0.1050 | -0.7900 | 0.1044 |
| | **o3-mini + Tools** | 0.4319 | 0.0381 | 0.2029 | 0.7157 | 0.4349 | 0.6966 |
| | **Qwen3-30B+Tools** | 0.2721 | -0.2095 | 0.1197 | 0.4579 | -0.0847 | 0.4840 |
| | **Qwen3-235B+Tools** | 0.6003 | 0.0626 | 0.2105 | 0.6757 | 0.3527 | 0.6890 |
| | **o3-mini+Code** | 0.4386 | 0.0430 | 0.2051 | 0.7085 | 0.4215 | 0.6862 |
| **RCA** | **RCA+Qwen2.5** | 0.7069 | **0.1053** | **0.2338** | **0.7177** | **0.4476** | 0.6931 |
| | **RCA+GPT-4.1** | **0.8335** | 0.0564 | 0.1463 | 0.7107 | 0.4285 | 0.6818 |

Table 12: Complete explanation experiment results in ablation study

| | | CRT | | | | Diabetes | | | | Heart Disease | | | |
|---|---|---|---|---|---|---|---|---|---|---|---|---|---|
| | | CL | LA | EBM | CB | CL | LA | EBM | CB | CL | LA | EBM | CB |
| **Qwen2.5-72B** | original | **8.24** | **8.89** | **8.47** | **7.61** | **8.13** | **8.57** | **8.74** | **6.43** | **7.62** | **8.47** | **8.94** | **6.18** |
| | w/o distribution | 7.70 | 8.35 | 8.10 | 6.29 | 7.93 | 8.47 | 8.27 | 6.36 | 7.32 | 7.87 | 8.21 | 5.74 |
| | w/o reflection | 7.54 | 8.21 | 7.83 | 6.24 | 7.93 | 8.49 | 8.11 | 6.33 | 7.54 | 8.23 | 8.46 | 5.92 |
| | w/o check | 7.75 | 8.32 | 8.13 | 6.45 | 7.77 | 8.19 | 8.52 | 6.18 | 7.53 | 8.16 | 8.30 | 5.89 |
| **GPT-4.1** | original | **8.16** | **8.59** | **8.87** | **7.62** | **8.03** | **8.38** | **8.63** | **6.94** | **7.74** | **8.53** | **8.79** | **6.42** |
| | w/o distribution | 7.45 | 8.24 | 7.97 | 6.19 | 7.18 | 7.72 | 8.24 | 6.85 | 7.12 | 7.93 | 7.51 | 5.61 |
| | w/o reflection | 7.57 | 8.54 | 8.06 | 6.86 | 7.34 | 7.94 | 8.42 | 6.48 | 7.13 | 7.86 | 7.89 | 5.76 |
| | w/o check | 7.40 | 8.24 | 7.83 | 6.37 | 7.73 | 8.06 | 8.59 | 6.52 | 6.72 | 7.43 | 6.70 | 5.10 |

### A.16.1 RULES OPTIMIZATION

Table 13 15 shows the prompt template for $M_{ref}$ to iteratively extract rules in the self-reflection process. In the prompt, incorrectly predicted samples, along with previous rules and data distribution are fed to $M_{ref}$, including features and true labels. Then $M_{ref}$ will consider what caused the wrong predictions and optimized the rule base. If the previous rule is empty, it means extracting initial rules. We specifically emphasized adherence to medical knowledge in the prompts and incorporated negative example to standardize rule generation.

### A.16.2 RULES CHECK

Table 16-18 shows the prompt template for $M_{chk}$ to check and delete the rules. At the end of each epoch, $M_{chk}$ checks the rule base to maintain the quality of the rules. The prompt lists several major errors we have identified and provides example rules tailored to specific diseases and features. We can

see that the prompt also include previous distribution and previous rules. It instructs $M_{chk}$ to examine each rule and remove incorrect or low-quality rule, preventing them from affecting predictions.

### A.16.3 DISEASE PREDICTION

Table 19-21 show the prompt template for $M_{pred}$ to generate prediction and explanation for the patient. We can see that the prompts are largely consistent across the three datasets, with only minor differences in wording. The prompt on all three datasets contains both positive and negative examples to provide demonstrations for the $M_{pred}$.

Table 13: Prompt template for reflective rules extraction on CRT dataset

```
You are an advanced reasoning agent that can improve based on self reflection.
The original task is "Given clinical features of tumor patient, estimate whether
the patient has the catheter related thrombosis(CRT) or not and explain your
reasoning.". Now you will be given the previous rules and some wrong samples
that you have attempted to predict CRT but failed. Considering patients' clinical
features and their true CRT results, you need to reflect on and revise rules to
help CRT prediction.
The rules must be supported by medical knowledge. Note that rules like "If the
patient has a BMI value between 10 and 50 and a history of previous CRT, predict
no catheter-related thrombosis." are not reasonable - you cannot predict CRT only
based on BMI and previous CRT history.
There might be some outliers in the data. You will also be given the features
distribution on the whole dataset.  You can determine the outliers based on
distribution before summarizing the rules.  Don't utilize relationship between
outliers and CRT, and don't let your rules be destroyed by outliers easily. Don't
exclude patients with outliers, you should use other features to predict CRT for
them.
Keep the rules brief. Only output rules. Your rules must be general enough for
any patients. Give your response in this format:

Rules, which should be a list of rules, each rule is a short sentence.

Previous distribution:
{distribution}

Previous rules:
{rules}
(If it is empty, it means summarizing the initial rules)

Wrong samples:
{samples}

Rules:
```

Table 14: Prompt template for reflective rules extraction on Diabetes dataset

```
You are an advanced reasoning agent that can improve based on self reflection.
The original task is "Given clinical features of patient, estimate whether the
patient has the diabetes or not and explain your reasoning.".  Now you will
be given the previous rules and some wrong samples that you have attempted to
predict diabetes but failed. Considering patients' clinical features and their
true diabetes results, you need to reflect on and revise rules to help diabetes
prediction.
The rules must be supported by medical knowledge. Note that rules like "If the
patient has a BMI value greater than 25, predict diabetes." are not reasonable -
you cannot predict diabetes only based on BMI.
```

There might be some outliers in the data. You will also be given the features
distribution on the whole dataset. You can determine the outliers based on
distribution before summarizing the rules. Don't utilize relationship between
outliers and diabetes, and don't let your rules be destroyed by outliers easily.
Don't exclude patients with outliers, you should use other features to predict
diabetes for them.
Keep the rules brief. Only output rules. Your rules must be general enough for
any patients. Give your response in this format:

Rules, which should be a list of rules, each rule is a short sentence.

Data distribution:
{distribution}

Previous rules:
{rules}
(If it is empty, it means summarizing the initial rules)

Wrong samples:
{samples}

Rules:

Table 15: Prompt template for reflective rules extraction on Heart Disease dataset

You are an advanced reasoning agent that can improve based on self reflection.
The original task is "Given clinical features of patient, estimate whether the
patient has the heart disease or not and explain your reasoning.". Now you will
be given the previous rules and some wrong samples that you have attempted to
predict heart disease but failed. Considering patients' clinical features and
their true heart disease results, you need to reflect on and revise rules to help
heart disease prediction.
The rules must be supported by medical knowledge. Note that rules like "If the
patient has a blood pressure greater than 160 and has diabetes, predict no heart
disease." are not reasonable — you cannot predict heart disease only based on
blood pressure and diabetes.
There might be some outliers in the data. You will also be given the features
distribution on the whole dataset. You can determine the outliers based on
distribution before summarizing the rules. Don't utilize relationship between
outliers and heart disease, and don't let your rules be destroyed by outliers
easily. Don't exclude patients with outliers, you should use other features to
predict heart disease for them.
Keep the rules brief. Only output rules. Your rules must be general enough for
any patients. Give your response in this format:

Rules, which should be a list of rules, each rule is a short sentence.

Data distribution:
{distribution}

Previous rules:
{rules}
(If it is empty, it means summarizing the initial rules)

Wrong samples:
{samples}

Rules:

Table 16: Prompt template for rules check on CRT dataset

You are an advanced reasoning agent that can improve based on self reflection. The original task is "Given clinical features of tumor patient and some prediction rules, estimate whether the patient has the catheter related thrombosis(CRT) or not and explain your reasoning" Given the previous rules and the features distribution, you need to check and delete the error rules.
Rules like "If the patient has a BMI value between 10 and 50 and a history of previous CRT, predict no catheter-related thrombosis." are not reasonable, because it's inconsistent with medical knowledge – you cannot predict CRT only based on BMI and previous CRT history.
Also, there might be some outliers in data. Rules that utilize relationship between outliers and disease, like "If the patient has a D-dimer level between 0.1 and 0.79, but any numerical feature is an extreme outlier, they are less likely to develop CRT." is forbidden. However, outliers could mislead prediction, so you should indicate in the rules how to identify outliers. Don't exclude patients with outliers, you should use other features to support disease prediction for them.
And rules that are too specific for certain patient are awful. You need to delete rules similar to those listed above. Give your response in this format:

Rules, which should be a list of rules, each rule is a short sentence.

Previous distribution:
{distribution}

Previous rules:
{rules}

Rules:

Table 17: Prompt template for rules check on Diabetes dataset

You are an advanced reasoning agent that can improve based on self reflection. The original task is "Given clinical features of patient and some prediction rules, estimate whether the patient has the diabetes or not and explain your reasoning" Given the previous rules and the features distribution, you need to check and delete the error rules.
Rules like "If the patient has a BMI value greater than 25, predict diabetes." are not reasonable, because it's inconsistent with medical knowledge – you cannot predict diabetes only based on BMI.
Also, there might be some outliers in data. Rules that utilize relationship between outliers and disease, like "If the patient has a Diastolic blood pressure between 80 mmHg and 90 mmHg, but any numerical feature is an extreme outlier, they are less likely to develop diabetes." is forbidden. However, outliers could mislead prediction, so you should indicate in the rules how to identify outliers. Don't exclude patients with outliers, you should use other features to support disease prediction for them.
And rules that are too specific for certain patient are awful. You need to delete rules similar to those listed above. Give your response in this format:

Rules, which should be a list of rules, each rule is a short sentence.

Previous distribution:
{distribution}

Previous rules:
{rules}

Rules:

Table 18: Prompt template for rules check on Heart Disease dataset

You are an advanced reasoning agent that can improve based on self reflection. The original task is "Given clinical features of tumor patient and some prediction rules, estimate whether the patient has the heart disease or not and explain your reasoning" Given the previous rules and the features distribution, you need to check and delete the error rules.

Rules like "If the patient has a blood pressure greater than 160 and has diabetes, predict no heart disease." are not reasonable, because it's inconsistent with medical knowledge – you cannot predict heart disease only based on blood pressure and diabetes.

Also, there might be some outliers in data. Rules that utilize relationship between outliers and disease, like "If the patient has a CRP level between 10 and 12, but any numerical feature is an extreme outlier, they are less likely to develop heart disease." is forbidden. However, outliers could mislead prediction, so you should indicate in the rules how to identify outliers. Don't exclude patients with outliers, you should use other features to support disease prediction for them.

And rules that are too specific for certain patient are awful. You need to delete rules similar to those listed above. Give your response in this format:

Rules, which should be a list of rules, each rule is a short sentence.

Previous distribution:
{distribution}

Previous rules:
{rules}

Rules:

Table 19: Prompt template for disease prediction on CRT dataset

Given clinical features of tumor patient, estimate whether the patient has the catheter related thrombosis(CRT) or not and explain your reasoning. You will be given some rules for prediction and distribution of training dataset. You can refer to the following rules, but don't limit yourself to them. Remember there are some outliers in the data. Give your response in this format:
(1) CRT Prediction, which should be either "no catheter-related thrombosis" or "catheter-related thrombosis".
(2) Explanation, which should be in a single, short paragraph.

Here are some examples:
Features:Granulocyte-to-lymphocyte ratio is 1.44, D-dimer is 0.19, chemotherapy, catheterization is CVC(Central Venous Catheter), no thoracic therapy, age at hospital is 29, platelet is 353.0, hemoglobin is 138.0, BMI is 18.83, gender is male, history of previous catheterization, no history of previous catheter related thrombosis, no lung cancer, no gastric cancer, lymphoma, no gynecologic tumors, no urologic tumors.

CRT Prediction: no catheter-related thrombosis

Explanation:The GLR of the patients was 1.44. Some studies have suggested that GLR is associated with thrombosis, but the normal threshold value of GLR is still under debate and the range of normal values of GLR varies from study to study. The patient's GLR value of 1.44 was within the normal range, which may indicate a lower risk of thrombosis. The patient's D-dimer level was 0.19 mg/L, which is within the normal range (less than 0.5 mg/L is generally considered normal), and lower D-dimer levels are usually associated with lower thrombotic risk. The patient is receiving chemotherapy, a known risk factor for VTE. Chemotherapy patients have a 6.5-fold elevated risk of thrombosis. The patient is using a CVC (Central Venous Catheter). The use of a central venous catheter is itself a risk factor for VTE, especially in oncology patients. The patient was relatively young at 29 years of age, and usually younger patients have a lower risk of thrombosis. The patient's platelet level was 353.0, slightly above the normal range, and thrombocytosis is a predictor of VTE. The patient's hemoglobin level was 138.0 g/L, which is in the normal range.BMI: The patient's BMI was 18.83, which is in the underweight range, and it is generally accepted that higher BMIs are more likely to result in CRT.The patient was male, and the effect of gender on thrombotic risk has varied in different studies. The patient had a history of previous catheterization, which may increase the risk of CRT. The patient had no history of prior thrombosis and did not develop certain tumors, which may indicate a lower risk of thrombosis. In summary, the patient was predicted to have a low risk of CRT.

Features:Granulocyte-to-lymphocyte ratio is 2.73, D-dimer is 0.1, chemotherapy, catheterization is PICC(Peripherally Inserted Central Catheter), no thoracic therapy, age at hospital is 30, platelet is 267.0, hemoglobin is 108.0, BMI is 26.04, gender is female, no history of previous catheterization, no history of previous catheter related thrombosis, no lung cancer, no gastric cancer, no lymphoma, no gynecologic tumors, no urologic tumors.

CRT Prediction: catheter-related thrombosis

Explanation:GLR is an indicator of inflammation and immune status.GLR 2.73 is a relatively high value and may indicate the presence of an inflammatory response, which may be associated with an increased risk of thrombosis.D-dimer is a marker of coagulation and fibrinolysis.A level of 0.1 is usually considered normal or only slightly elevated and is not sufficient to directly diagnose VTE.Therefore, this level of D-dimer is unlikely to indicate the presence of CRT. chemotherapy may increase a patient's coagulation status because it can cause vascular endothelial injury and inflammation, which can increase the risk of thrombosis. the use of a PICC is a known risk factor for CRT because catheters can cause vascular endothelial injury and inflammation, which can promote thrombosis. Younger age is associated with a relatively lower risk of VTE. Platelet counts above the normal range may indicate a risk of inflammation or thrombosis. A slightly lower hemoglobin level may indicate mild anemia, but this level usually does not directly affect the risk of thrombosis. A slightly higher body mass index (BMI) indicates that the patient may be overweight, which is a risk factor for VTE. Gender is not an independent risk factor for CRT. There was no history of previous catheterization, which reduced the patient's risk of CRT. There is no history of catheter-related thrombosis, which reduces the patient's risk of CRT. No history of certain malignancies, which are known risk factors for VTE and CRT. In summary, the patient's risk of having catheter-related thrombosis is relatively high.

(END OF EXAMPLES)

Here are some rules:
{rules}
(If it is empty, it means there is no rule.)
(END OF RULES)

```
Here is the distribution:
{distribution}
(END OF DISTRIBUTION)

Features:
{features}

CRT Prediction:
```

Table 20: Prompt template for disease prediction on Diabetes dataset

```
Given clinical features of patient, estimate whether the patient has the diabetes
or not and explain your reasoning. You will be given some rules for prediction
and distribution of training dataset. You can refer to the following rules, but
don't limit yourself to them. Remember there are some outliers in the data. Give
your response in this format:
(1) Diabetes Prediction, which should be either "no diabetes" or "diabetes".
(2) Explanation, which should be in a single, short paragraph.

Here are some examples:
Features: Number of pregnancies is 1, Plasma glucose concentration (2-hour
test) level is 135, Diastolic blood pressure is 54 mm Hg, Triceps skin fold
thickness is 0 mm, 2-Hour serum insulin level is 0 mu U/ml, BMI is 26.7,
DiabetesPedigreeFunction(Genetic diabetes score) is 0.687, Age is 62.

Diabetes Prediction: no diabetes

Explanation: The patient's 2-hour plasma glucose level is 135 mg/dL, which is
below the diagnostic threshold for diabetes (>=200 mg/dL) and even below the
range for prediabetes (140-199 mg/dL). While factors like age (62), overweight
BMI (26.7), and a moderate genetic risk score (0.687) increase diabetes risk, the
absence of elevated glucose levels within diagnostic ranges and other features
(e.g., low triceps skin fold thickness, low insulin level) do not meet criteria
for diabetes. Diagnosis primarily relies on glucose levels, which here are within
normal limits.

Features: Number of pregnancies is 4, Plasma glucose concentration (2-hour
test) level is 171, Diastolic blood pressure is 72 mm Hg, Triceps skin fold
thickness is 0 mm, 2-Hour serum insulin level is 0 mu U/ml, BMI is 43.6,
DiabetesPedigreeFunction(Genetic diabetes score) is 0.479, Age is 26.

Diabetes Prediction: diabetes

Explanation: The patient's plasma glucose concentration (171 mg/dL) exceeds the
prediabetes threshold (>=140 mg/dL) and approaches the diabetes range, combined
with a markedly elevated BMI (43.6, class III obesity), a major risk factor for type
2 diabetes. The genetic risk score (0.479) and history of 4 pregnancies (potential
gestational diabetes risk) further support this prediction. While triceps skinfold
thickness and insulin levels of 0 suggest possible data anomalies, the high glucose
and BMI strongly indicate diabetes likelihood despite the patient's younger age
(26).
(END OF EXAMPLES)

Here are some rules:
{rules}
(If it is empty, it means there is no rule.)
(END OF RULES)
```

```
Here is the distribution:
{distribution}
(END OF DISTRIBUTION)

Features:
{features}

Diabetes Prediction:
```

Table 21: Prompt template for disease prediction on Heart Disease dataset

```
Given clinical features of tumor patient, estimate whether the patient has the
heart disease or not and explain your reasoning. You will be given some rules for
prediction and distribution of training dataset. You can refer to the following
rules, but don't limit yourself to them. Remember there are some outliers in the
data. Give your response in this format:
(1) Heart Disease Prediction, which should be either "no heart disease" or "heart
disease".
(2) Explanation, which should be in a single, short paragraph.

Here are some examples:
Features: Age is 62.0, Gender is Female, Blood Pressure is 133.0, Cholesterol
Level is 166.0, Exercise Habits is Medium, Smoking is No, Family Heart Disease is
No, Diabetes is No, BMI is 25.739170533963147, High Blood Pressure is No, Low HDL
Cholesterol is Yes, High LDL Cholesterol is No, Stress Level is Low, Sleep Hours
is 5.493276805328829, Sugar Consumption is Medium, Triglyceride Level is 126.0,
Fasting Blood Sugar is 102.0, CRP Level is 11.60991435489297, Homocysteine Level
is 8.297757016065253

Heart disease Prediction: heart disease

Explanation: The patient has several risk factors for heart disease. At 62 years
old, the patient has a cholesterol level of 166, and despite having normal blood
pressure according to the "High Blood Pressure" marker, a blood pressure of 133
is relatively close to the elevated range. The presence of low HDL cholesterol
is a risk factor for heart disease.  The C-reactive protein (CRP) level of
11.60991435489297 is elevated, indicating possible inflammation in the body,
which is associated with heart disease. Although the patient has a medium level
of exercise and no family history of heart disease or diabetes, the combination
of age, low HDL cholesterol, and elevated CRP level increases the likelihood of
having heart disease.

Features: Age is 35.0, Gender is Male, Blood Pressure is 159.0, Cholesterol Level
is 261.0, Exercise Habits is Low, Smoking is No, Family Heart Disease is No,
Diabetes is Yes, BMI is 21.63849835899007, High Blood Pressure is No, Low HDL
Cholesterol is Yes, High LDL Cholesterol is No, Stress Level is High, Sleep Hours
is 4.296875738592791, Sugar Consumption is Medium, Triglyceride Level is 385.0,
Fasting Blood Sugar is 136.0, CRP Level is 1.9462702594315329, Homocysteine Level
is 11.140952179886469

Heart disease Prediction: no heart disease
```

Explanation: Although the patient presented with multiple risk factors such as elevated blood pressure, high cholesterol levels, diabetes, high triglycerides, high stress, low sleep hours, elevated CRP, and low HDL cholesterol, it has been determined that he has no heart disease. It is possible that there are mitigating factors not mentioned, such as effective medical management or significant lifestyle changes that reduce the impact of these risk factors on the heart.

(END OF EXAMPLES)

Here are some rules:
{rules}
(If it is empty, it means there is no rule.)
(END OF RULES)

Here is the distribution:
{distribution}
(END OF DISTRIBUTION)

Features:
{features}

Heart Disease Prediction:

