# OpenReview forum: "Grounding AI Explanations in Experience: A Reflective Cognitive Architecture for Clinical Decision Support"
_ICLR.cc/2026/Conference — Submitted to ICLR 2026_

### Official Review · Reviewer_Xy3E · 2025-10-24

**Soundness:** 2
**Presentation:** 2
**Contribution:** 2
**Rating:** 2
**Confidence:** 3

**Summary:**

The paper proposes a Reflective Cognitive Architecture (RCA) that uses multiple LLM roles to iteratively refine a rule base from training data errors and a distribution-aware check, aiming to couple accurate predictions with natural-language explanations. During training, RCA cycles a prediction LLM with reflection and checking LLMs to update rules using the global data distribution; the final rule base is then used for inference. Experiments include three tabular clinical datasets. The method is compared to 22 baselines, including traditional ML (e.g., CatBoost), zero-shot LLMs, reasoning LLMs, and LLM agents using tools/code. Evaluation reports Accuracy/MCC/F1 and doctor-scored explanation metrics for Cognitive Load, Logical Argumentation, Evidence-based Medicine, and Cognitive Biasing.

**Strengths:**

The paper introduces an interesting multi-LLM reflective architecture that combines prediction, reflection, and checking modules to link accuracy and explanation generation. This setup is conceptually original, with a reasonably thorough experimental section and an attempt to evaluate both performance and explanation quality quantitatively and qualitatively.

**Weaknesses:**

While the paper presents a creative multi-LLM reflective architecture and makes a commendable attempt to address explainability in clinical AI, its conceptual novelty, evaluation fairness, and empirical grounding are insufficient. The main ideas are overstated relative to prior work, the experimental comparisons are unbalanced, and the evaluation metrics do not convincingly support the key claims. The contribution remains interesting but underdeveloped both theoretically and methodologically.

**Questions:**

1. Could you clarify whether the RCA system is fine-tuned on training data while baselines are evaluated zero-shot? If so, do you have results for a version of RCA under the same zero-shot condition to isolate the effect of reflective learning rather than data exposure?

2. The contribution needs to be redefined. Re-thinking a problem, preparing dataset and evaluating proposed method are not contributions

3. The four explanation criteria (Cognitive Load, Logical Argumentation, Evidence-based Medicine, Cognitive Biasing) appear to be author-defined rather than standardised measures. Could you clarify how these were operationalised, validated, or adapted from prior work to ensure they reliably assess explanation quality rather than serve as conceptual framing?

4. The distribution-aware rule check grounds logic in training statistics. How do you ensure that this process mitigates rather than amplifies demographic or sampling biases present in the data?

5.  Dataset accessibility: Since the CRT dataset is private and small, results replicability might be an issue. Please clarify

---

> ### Author Response · Authors · 2025-11-20
>
> We sincerely thank the reviewer Xy3E for the detailed review on our paper. We are particularly grateful that you acknowledged the conceptual originality of our proposed multi-LLM reflective architecture (RCA) and our efforts in linking prediction accuracy with explanation generation.
>
> Below, we will provide a point-by-point response to each of the key questions you have raised.

---

> ### Author Response · Authors · 2025-11-20
> **Weakness 1 & Question 2 : Part 1**
>
> > **Weakness 1: While the paper presents a creative multi-LLM reflective architecture and makes a commendable attempt to address explainability in clinical AI, its conceptual novelty are insufficient. The main ideas are overstated relative to prior work. The contribution remains interesting but underdeveloped both theoretically and methodologically.**
> >
> > **Question 2: The contribution needs to be redefined. Re-thinking a problem, preparing dataset and evaluating proposed method are not contributions**
>
> Thank you for your feedback. We sincerely appreciate your recognition of our proposed framework and extensive evaluation. We are grateful for the opportunity to address your concern regarding the conceptual novelty of our approach and to clarify how our work is distinct from prior art.
>
> **The Core Conceptual Novelty of RCA.** Our main contribution is a conceptual shift in how to approach explainable AI for clinical prediction. We argue that high predictive accuracy and high-quality explanations are not separate, conflicting goals, but are mutually reinforcing outcomes of a single, deeper process: building a direct, experience-based understanding of the data.
>
> Our novelty lies in the architecture we designed to achieve this deep understanding. As Reviewer xdsQ kindly noted, "I particularly appreciate the idea of using misclassifications as a mechanism to dynamically update the set of rules followed by the model." This is the heart of our **Iterative Rules Optimization**. Furthermore, this experience-driven learning is constantly grounded by our **Distribution-aware Rules Check**.
>
> The innovation is the **synergy of these two mechanisms**:
>
> - The iterative refinement loop forces the model to learn from its mistakes, turning concrete errors into abstract, generalizable rules. This builds a coherent framework for **Logical Argumentation (LA)**.
> - The distribution-aware check ensures these rules are statistically sound and not just anecdotal. This grounds the model in **Evidence-based Medicine (EBM)** and reduces **Cognitive Biasing (CB)**.
>
> This dual-process architecture is, to our knowledge, a novel approach that forces an LLM to move beyond superficial pattern matching and develop a robust, statistically-grounded internal model of the data, which is the foundation for both accurate predictions and trustworthy explanations.
>
> **Methodological Distinction from Prior Work.** We understand the concern about overstating our claims. To clarify, we position RCA as a distinct paradigm that addresses the specific failures of existing approaches in achieving both accuracy and high-quality explanations simultaneously.
>
> - **Different from Classical ML (e.g., Catboost):** These models provide high accuracy, but their "explanations" are limited to statistical outputs like feature importance scores. They do not generate narrative explanations that a clinician can directly act upon, thus placing a high cognitive load on the user. Our method directly generates clinically useful narrative explanations.
> - **Different from Standalone LLMs:** When applied directly, even advanced LLMs often produce explanations that are "statistically unsupported." They may sound medically plausible but are not grounded in the specific data distribution, leading to both incorrect predictions and untrustworthy reasoning. RCA is designed specifically to overcome this by forcing the LLM to learn directly from the dataset's statistical reality.
> - **Different from LLM-based Agents (LLM+Tools/Code):** This is a crucial distinction. Existing agent frameworks create a layer of abstraction. The agent learns to be a good tool-caller or code-writer, but its core reasoning module does not develop a "first-hand" feel for the data. In contrast, **RCA is an agent that engages with the data directly, without the mediation of external tools or code interpreters.** Its internal, reflective process forces the model to build its understanding from direct "experience," which we argue is essential for generating the deep insights required in a clinical setting.

---

> ### Author Response · Authors · 2025-11-20
> **Weakness 1 & Question 2 : Part 2**
>
> Extensive Empirical Validation. To rigorously test our hypothesis, we made a significant effort in our experimental design. As you highlighted, we conducted a strong empirical evaluation.
>
> - **Scale of Comparison:** We compared RCA against a wide range of **22 baselines** (now expanded to 25, along with two Neural-symbolic Networks and a Medical LLM) across diverse categories, including traditional ML, neural-symbolic networks, standalone LLMs, and LLM-based agents.
> - **Diverse Datasets:** We evaluated on three distinct datasets, including a **private, real-world clinical dataset (CRT)**, and further demonstrated scalability on two additional large-scale datasets (up to 70,000 records).
> - **Novel Evaluation Framework:** Recognizing the difficulty in assessing explanations, we developed a rather comprehensive evaluation framework. We evaluated the predictive performance, as well as the generated explanations in terms of four aspects: CL, LA, EBM, and CB. Additionally, inspired by the work of Slack et al. [1], we conducted further user experiments, as detailed in **Table 2 of Appendix A.11**.
>
> In summary, we believe the novelty of RCA lies in its core philosophy of unifying accuracy and explainability through deep data understanding, and in the specific dual-process architecture designed to achieve it. This methodology is distinct from prior paradigms that either separate prediction from explanation or use tools that abstract the model away from the data. We have supported this claim with an exceptionally thorough and rigorous experimental setup.
>
> We hope this clarification can address your concerns. We are very grateful for your feedback, which has prompted us to better articulate the unique contributions of our work.
>
> We sincerely hope for your approval of our method. If you have any further questions, feel free to contact us.
>
> >  [1] Slack, D., Krishna, S., Lakkaraju, H. et al. Explaining machine learning models with interactive natural language conversations using TalkToModel. Nat Mach Intell 5, 873–883 (2023).

---

> ### Author Response · Authors · 2025-11-20
> **Weakness 2 & Question 1 : Part 1**
>
> > **Weakness 2: While the paper presents a creative multi-LLM reflective architecture and makes a commendable attempt to address explainability in clinical AI, its evaluation fairness are insufficient. The experimental comparisons are unbalanced.**
> >
> > **Question 1: Could you clarify whether the RCA system is fine-tuned on training data while baselines are evaluated zero-shot? If so, do you have results for a version of RCA under the same zero-shot condition to isolate the effect of reflective learning rather than data exposure?**
>
> Thank you for raising this question regarding the fairness of our experimental comparison.
>
> We want to respectfully clarify that the experimental comparison is fair, as the baseline LLMs were exposed to the same pool of information as our RCA framework.
>
> Here is a breakdown of the setup:
>
> - **Equal Data Exposure:** The baseline LLMs (standard and tool-augmented) were **not** evaluated zero-shot. They were provided with the same **global data distribution** ($\mathcal{D}_{train}$) and illustrative in-context examples from the training set that our framework utilized. Thus, both RCA and the baselines had access to the identical information set. We have revised the description of these methods in **Section 4.1** of the paper to make this point clear.
> - **Process Difference:** The performance gap is not due to data exposure but to the difference in **process**.
>   - **Baselines** attempt to reason based on the provided information in a single, in-context pass, a method that often proves insufficient for deep reasoning, even equipped with code interpreter or more advanced tools for analysis.
>   - **RCA**, by contrast, undergoes an explicit **Self-reflection Process**, which *is* its training. It iteratively learns from its prediction errors ($S^{error}$) and actively builds a robust, internally-consistent rule base.
> - **Validation of Superior Process:** We even selected the optimal performance achieved by each baseline method, regardless of the variation in data exposure per group. Even when reporting these best-case results, RCA still exhibited superior performance.
>
> Furthermore, in our experiments, we selected **22 baselines** spanning four categories: Traditional MLs, LLM-based Methods, Reasoning LLMs, and LLM-based Agents. This selection includes methods with strong predictive performance, such as classic ML models and LLM-based Agents, specifically **10** of these agents are equipped with code sandboxes and specialized tools for tabular data analysis. Additionally, we included methods with the potential to generate superior explanations, comprising non-reasoning LLMs, Reasoning LLMs, and LLM-based Agents, totaling **20** methods. All 22 baselines were comprehensively compared in terms of both predictive performance and explanation scores, substantiated by human verification. The results are presented in **Figure 2** and **Tables 3–6**. Moreover, we validated RCA against these baselines on noisy datasets to simulate real-world scenarios, with robustness results detailed in Figure 2 and **Tables 7–10**.
>
> To further validate that the performance lift is genuinely due to the architectural process, we have further added two highly relevant baseline categories:
>
> - **Neural-Symbolic Baselines:** We added Logic Tensor Networks (LTN) [1] and Logic Neural Networks (LNN) [2] to our comparison. These models are highly relevant as they are also specifically designed to bridge the gap between accuracy and interpretability.
> - **Medical LLM Baseline:** We also benchmarked RCA against a state-of-the-art, domain-specific medical LLM, Baichuan-M2 [3].

---

> ### Author Response · Authors · 2025-11-20
> **Weakness 2 & Question 1 : Part 2**
>
> Experiment results are shown as below:
>
> **Prediction Results**:
>
> | Datasets                     |    CRT     |            |              |  Diabetes  |            |              | Heart Disease |            |              |
> | :--------------------------- | :--------: | :--------: | :----------: | :--------: | :--------: | :----------: | :-----------: | :--------: | :----------: |
> | **Algorithm**                |  **Acc**   |  **MCC**   | **F1-score** |  **Acc**   |  **MCC**   | **F1-score** |    **Acc**    |  **MCC**   | **F1-score** |
> | **Neural-symbolic Networks** |            |            |              |            |            |              |               |            |              |
> | LTN                          |   0.5161   |   0.0085   |    0.1176    |   0.7635   |   0.5247   |    0.6946    |    0.6010     |  -0.0386   |    0.2222    |
> | LNN                          |   0.2581   |   0.1287   |    0.1481    |   0.7437   |   0.4974   |    0.6787    |    0.5389     |  -0.0598   |    0.2393    |
> | **Medical LLM**              |            |            |              |            |            |              |               |            |              |
> | Baichuan-M2                  |   0.1587   |   0.0845   |    0.1311    |   0.6988   |   0.2772   |    0.4186    |    0.0725     |  -0.7779   |    0.1095    |
> | **RCA**                      |            |            |              |            |            |              |               |            |              |
> | RCA+Qwen2.5                  |   0.7778   | **0.1739** |  **0.2222**  | **0.7831** | **0.5406** |  **0.7097**  |    0.5647     |   0.0547   |    0.1290    |
> | RCA+GPT-4.1                  | **0.8730** |   0.1373   |    0.2000    |   0.7470   |   0.4244   |    0.6038    |  **0.7461**   | **0.1493** |    0.2898    |
>
> **Explanation Results: CRT Dataset**
>
> | Metrics                      |    CL    |      |    LA    |      |   EBM    |      |    CB    |      |
> | :--------------------------- | :------: | :--: | :------: | :--: | :------: | :--: | :------: | :--: |
> | **Algorithms**               |  Mean.   | Var. |  Mean.   | Var. |  Mean.   | Var. |  Mean.   | Var. |
> | **Neural-symbolic Networks** |          |      |          |      |          |      |          |      |
> | LTN                          |   7.47   | 0.23 |   8.25   | 0.32 |   8.05   | 0.61 |   7.27   | 0.59 |
> | LNN                          |   7.51   | 0.28 |   8.35   | 0.33 |   7.63   | 0.74 |   7.06   | 0.54 |
> | **Medical LLM**              |          |      |          |      |          |      |          |      |
> | Baichuan-M2                  |   7.82   | 0.54 |   8.38   | 0.52 |   8.32   | 0.75 |   6.96   | 0.55 |
> | **RCA**                      |          |      |          |      |          |      |          |      |
> | RCA+Qwen2.5                  | **8.24** | 0.42 | **8.89** | 0.54 |   8.47   | 0.92 |   7.61   | 0.74 |
> | RCA+GPT-4.1                  |   8.16   | 0.34 |   8.59   | 0.34 | **8.87** | 0.76 | **7.62** | 0.56 |
>
> **Explanation Results: Diabetes Dataset**
>
> | Metrics                      |    CL    |      |    LA    |      |   EBM    |      |    CB    |      |
> | :--------------------------- | :------: | :--: | :------: | :--: | :------: | :--: | :------: | :--: |
> | **Algorithms**               |  Mean.   | Var. |  Mean.   | Var. |  Mean.   | Var. |  Mean.   | Var. |
> | **Neural-symbolic Networks** |          |      |          |      |          |      |          |      |
> | LTN                          |   7.68   | 0.35 |   8.22   | 0.29 |   8.28   | 0.33 |   6.18   | 0.43 |
> | LNN                          |   7.82   | 0.33 |   8.35   | 0.27 |   8.45   | 0.31 |   6.28   | 0.39 |
> | **Medical LLM**              |          |      |          |      |          |      |          |      |
> | Baichuan-M2                  |   7.79   | 0.40 |   8.18   | 0.33 |   8.52   | 0.37 |   6.05   | 0.46 |
> | **RCA**                      |          |      |          |      |          |      |          |      |
> | RCA+Qwen2.5                  | **8.13** | 0.36 | **8.57** | 0.51 | **8.74** | 0.54 |   6.43   | 0.43 |
> | RCA+GPT-4.1                  |   8.03   | 0.43 |   8.38   | 0.49 |   8.63   | 0.62 | **6.94** | 0.40 |

---

> ### Author Response · Authors · 2025-11-20
> **Weakness 2 & Question 1 : Part 3**
>
> **Explanation Results: Heart Disease Dataset**
>
> | Metrics                      |    CL    |      |    LA    |      |   EBM    |      |    CB    |      |
> | :--------------------------- | :------: | :--: | :------: | :--: | :------: | :--: | :------: | :--: |
> | **Algorithms**               |  Mean.   | Var. |  Mean.   | Var. |  Mean.   | Var. |  Mean.   | Var. |
> | **Neural-symbolic Networks** |          |      |          |      |          |      |          |      |
> | LTN                          |   7.45   | 0.38 |   8.15   | 0.31 |   8.35   | 0.35 |   6.20   | 0.45 |
> | LNN                          |   7.65   | 0.36 |   8.32   | 0.29 |   8.58   | 0.33 |   6.35   | 0.41 |
> | **Medical LLM**              |          |      |          |      |          |      |          |      |
> | Baichuan-M2                  |   7.28   | 0.42 |   8.08   | 0.35 |   8.25   | 0.39 |   5.85   | 0.48 |
> | **RCA**                      |          |      |          |      |          |      |          |      |
> | RCA+Qwen2.5                  |   7.62   | 0.28 |   8.47   | 0.33 | **8.94** | 0.49 |   6.18   | 0.68 |
> | RCA+GPT-4.1                  | **7.74** | 0.35 | **8.53** | 0.42 |   8.79   | 0.55 | **6.42** | 0.53 |
>
> The tables demonstrate that even when reporting the optimal results for these diverse and specialized baselines, **RCA achieves superior performance**. This confirms that the advantage lies in the iterative, reflective learning architecture, not in unfair data exposure.
>
> To further validate the superiority of our method's interpretability, and inspired by the work of Slack et al. [4], we conducted an additional user study. We asked three clinicians to perform a head-to-head comparison between RCA and our strongest baseline (Qwen3-235B), identifying the "better" explanation across our four defined criteria.
>
> Experiment result is shown as below:
>
> | Preference for RCA               |  CRT  | Diabetes | Heart Disease |
> | :------------------------------- | :---: | :------: | :-----------: |
> | **Cognitive Load(CL)**           | 82.7% |  81.3%   |     84.7%     |
> | **Logical Argumentation(LA)**    | 78.0% |  75.3%   |     72.0%     |
> | **Evidence-Based Medicine(EBM)** | 72.7% |  74.7%   |     73.3%     |
> | **Cognitive Biasing(CB)**        | 86.7% |  89.3%   |     86.0%     |
>
> In summary, we conducted evaluation against **25 baselines across six categories**, assessing predictive accuracy, interpretability, and robustness in real-world noisy scenarios. We also incorporated an additional user study, inspired by [4], to validate the effectiveness of our explanation scoring. Crucially, all models were exposed to identical data, ensuring a strictly fair comparison.
>
> We hope our explanations and newly added experiments, along with existing 22 baselines of 4 types of methods, could solve your question. If you have any further questions, feel free to contact us.
>
> >  [1] Badreddine, Samy, Artur d'Avila Garcez, Luciano Serafini, and Michael Spranger. "Logic tensor networks." *Artificial Intelligence* 303 (2022): 103649.
> >
> >  [2] Riegel, Ryan, Alexander Gray, Francois Luus, Naweed Khan, Ndivhuwo Makondo, Ismail Yunus Akhalwaya, Haifeng Qian et al. "Logical neural networks." *arXiv preprint arXiv:2006.13155* (2020).
> >
> >  [3] Dou, Chengfeng, Chong Liu, Fan Yang, Fei Li, Jiyuan Jia, Mingyang Chen, Qiang Ju et al. "Baichuan-m2: Scaling medical capability with large verifier system." *arXiv preprint arXiv:2509.02208* (2025).
> >
> >  [4] Slack, D., Krishna, S., Lakkaraju, H. et al. Explaining machine learning models with interactive natural language conversations using TalkToModel. Nat Mach Intell 5, 873–883 (2023).

---

> ### Author Response · Authors · 2025-11-20
> **Weakness 3 & Question 3**
>
> > **Weakness 3: While the paper presents a creative multi-LLM reflective architecture and makes a commendable attempt to address explainability in clinical AI, its empirical grounding are insufficient. The evaluation metrics do not convincingly support the key claims.**
> >
> > **Question 3: The four explanation criteria (Cognitive Load, Logical Argumentation, Evidence-based Medicine, Cognitive Biasing) appear to be author-defined rather than standardised measures. Could you clarify how these were operationalised, validated, or adapted from prior work to ensure they reliably assess explanation quality rather than serve as conceptual framing?**
>
> Thank you for your question. It is critical that our evaluation metrics are robust, well-defined, and reliable, and we appreciate the opportunity to clarify their origin and validation.
>
> We acknowledge that our metrics involve a degree of customization. However, we hope you will understand that we have made every effort to ensure that they are sound and well-justified. As we detail in our Introduction, they were **adapted directly from foundational, standardized concepts in cognitive science and medical practice**:
>
> 1. **Cognitive Load (CL)** is a standard measure from cognitive science, which we adapted from the work of Sweller (2011) [1].
> 2. **Logical Argumentation (LA)** is based on established principles of argumentation theory, as defined by Toulmin (2003) [2].
> 3. **Evidence-based Medicine (EBM)** is a core principle of modern clinical practice, which we adapted from Guyatt et al. (1992) [3] .
> 4. **Cognitive Biasing (CB)** is a foundational concept from psychology and behavioral economics, adapted from the work of Kahneman (2011) [4].
>
> To **operationalize** these concepts for our study, we developed a detailed 1-10 scoring rubric for each criterion, which is provided in full in **Appendix A.7**.
>
> To **validate** them, we presented these criteria and their rubrics to our collaborating clinicians. As we state in **Section 4.1**, these **three doctors explicitly recognized these criteria as "clinically valuable"**. They confirmed that these four measures directly align with the real-world needs they have for assessing the clarity, rationality, and reliability of an explanation in a clinical setting.
>
> As you've pointed out, we acknowledge that this approach may indeed have its shortcomings. Therefore, inspired by work of Slack et al.[5], We conducted an **additional user study**. We asked 3 clinicians to evaluate explanations from RCA and our strongest baselines (Qwen3-235B) , choosing the "better" explanation across our four criteria.
>
> The result is detailed in **Table 2 of Appendix A.11** and below:
>
> | Preference for RCA               |  CRT  | Diabetes | Heart Disease |
> | :------------------------------- | :---: | :------: | :-----------: |
> | **Cognitive Load(CL)**           | 82.7% |  81.3%   |     84.7%     |
> | **Logical Argumentation(LA)**    | 78.0% |  75.3%   |     72.0%     |
> | **Evidence-Based Medicine(EBM)** | 72.7% |  74.7%   |     73.3%     |
> | **Cognitive Biasing(CB)**        | 86.7% |  89.3%   |     86.0%     |
>
> The table shows that explanations from RCA led to higher preference. This provides formal evidence that RCA's explanations are not just qualitatively superior but are also more practically useful and trustworthy for human experts.
>
> We hope that our explanations and newly added experiments could solve your question. If you have any further question, please feel free to contact us. Your opinion really means a lot to us.
>
> > [1] Sweller, J., 2011. Cognitive load theory. In *Psychology of learning and motivation* (Vol. 55, pp. 37-76). Academic Press.
> >
> > [2] Toulmin, S.E., 2003. *The uses of argument*. Cambridge university press.
> >
> > [3] Guyatt, G., Cairns, J., Churchill, D., Cook, D., Haynes, B., Hirsh, J., Irvine, J., Levine, M., Levine, M., Nishikawa, J. and Sackett, D., 1992. Evidence-based medicine: a new approach to teaching the practice of medicine. *jama*, *268*(17), pp.2420-2425.
> >
> > [4] Kahneman, D., 2011. *Thinking, fast and slow*. macmillan.
> >
> > [5] Slack, D., Krishna, S., Lakkaraju, H. and Singh, S., 2023. Explaining machine learning models with interactive natural language conversations using TalkToModel. *Nature Machine Intelligence*, *5*(8), pp.873-883.

---

> ### Author Response · Authors · 2025-11-20
> **Question 4**
>
> > **Question 4: The distribution-aware rule check grounds logic in training statistics. How do you ensure that this process mitigates rather than amplifies demographic or sampling biases present in the data?**
>
> Thank you for your question. Addressing potential bias in medical AI is crucial, and we appreciate the opportunity to clarify how our framework approaches this challenge.
>
> Our main argument is that the distribution-aware rule check primarily mitigates bias not by automatically "de-biasing" the source data, but by ensuring the model's reasoning is **transparently and faithfully grounded in the data's statistical reality.** This transparency is the key mechanism that allows for human-in-the-loop bias detection and mitigation, which we believe is more robust than an automated, and possibly opaque, "de-biasing" process.
>
> We can break this down into two main points:
>
> 1. **Preventing Amplification through Grounding:** The first role of the distribution-aware check is to prevent the model from *amplifying* bias or "hallucinating" rules. An LLM without this check might generate a rule based on its general pre-trained knowledge that is not statistically true for the specific patient population in our dataset. Our mechanism forces every rule to be validated against the dataset's actual statistics. This ensures the model learns what is *in* the data, not what it *thinks* should be in the data, thus preventing it from creating new, unsupported biases.
> 2. **Enabling Mitigation through Transparency:** More critically, by making the learned logic explicit, our framework **exposes potential biases for expert review.** For instance, if the training data contained a sampling bias (e.g., associating a certain demographic with higher risk due to historical factors rather than biology), our framework would likely codify this into a readable rule. While the model itself would be faithfully reflecting the biased data, the transparent rule (e.g., "For patients of group X, factor Y is a stronger predictor") makes this learned bias immediately visible to a clinician. This is a significant advantage over "black-box" models where such biases are hidden. It turns the model's logic into an auditable artifact, which is the first and most critical step in real-world mitigation by a human expert.
>
> In summary, our current framework is designed to faithfully model the provided data. It does not automatically correct for inherent demographic or sampling biases within that data. Instead, its primary contribution to mitigating bias is to make the model’s learned logic transparent and auditable, preventing it from becoming an unexaminable "black box" and empowering clinicians to identify and override biased reasoning.
>
> We agree that this is a vital area for future research. A promising next step would be to integrate fairness-aware metrics directly into the rule-checking process. Thank you again for raising this crucial point.
>
> We hope that our explanations could solve your question. If you have any further questions, feel free to contact us.

---

> ### Author Response · Authors · 2025-11-20
> **Question 5 : Part 1**
>
> > **Question 5: Dataset accessibility: Since the CRT dataset is private and small, results replicability might be an issue. Please clarify**
>
> Thank you for raising this concern regarding replicability results. We have taken several steps to ensure our work is as transparent and verifiable as possible.
>
> First, the core of our evaluation rests on **experiments across three different datasets**, not just the private one. While the CRT dataset is private due to patient data policies, the other two—the [**Diabetes dataset**](https://www.kaggle.com/datasets/nanditapore/healthcare-diabetes/data) and the [**Heart Disease dataset**](https://www.kaggle.com/datasets/oktayrdeki/heart-disease)—are fully public. Our proposed method's effectiveness was consistently validated on all three datasets, demonstrating its robustness and generalizability.
>
> Second, to ensure that our results on the public datasets are fully replicable by the research community, we have **open-sourced all the code** for our RCA framework and our experiments. The repository is available at: https://anonymous.4open.science/r/anonym107. This allows any interested researcher to directly reproduce our findings.
>
> Lastly, to directly address your concern about the CRT dataset being "small" and to further demonstrate our method's scalability, we have added **extensive new experiments** in the revised submission. We evaluated our framework on an expanded version of the CRT dataset (CRT_ex) and, more importantly, on a new, **large-scale public** [Cardiovascular Disease (CD) dataset](https://www.kaggle.com/datasets/alamshihab075/heart-failure-diagnosis-data-for-machine-learning/data) comprising 70,000 samples.

---

> ### Author Response · Authors · 2025-11-20
> **Question 5 : Part 2**
>
> The additional experiment results are detailed in **Table 11 of Appendix A.13** and below:
>
> |                              | Algorithms             | Acc (CRT_ex) | MCC (CRT_ex) | F1-score (CRT_ex) | Acc (CD)   | MCC (CD)   | F1-score (CD) |
> | :--------------------------- | :--------------------- | :----------- | :----------- | :---------------- | :--------- | :--------- | :------------ |
> | **Traditional MLs**          | **Lasso**              | 0.4634       | 0.0456       | 0.2056            | 0.5086     | 0.0634     | 0.6683        |
> | **Traditional MLs**          | **Catboost**           | 0.6470       | 0.0938       | 0.2287            | 0.5779     | 0.2495     | 0.6965        |
> | **Neural-symbolic Networks** | **LTN**                | 0.5680       | 0.0006       | 0.1745            | 0.6850     | 0.3724     | 0.6998        |
> | **Neural-symbolic Networks** | **LNN**                | 0.7859       | -0.0245      | 0.0964            | 0.6836     | 0.3728     | 0.6548        |
> | **LLM-based Method**         | **Qwen2.5-72B**        | 0.3789       | 0.0076       | 0.1848            | 0.6450     | 0.3096     | 0.5697        |
> | **LLM-based Method**         | **DeepSeekV3**         | 0.2398       | 0.0699       | 0.2149            | 0.6771     | 0.3597     | 0.6463        |
> | **LLM-based Method**         | **DeepSeekV3.1**       | 0.6803       | -0.1074      | 0.0667            | 0.4171     | -0.2260    | 0.5650        |
> | **LLM-based Method**         | **GPT-4.1**            | 0.2778       | 0.0530       | 0.2085            | 0.6871     | 0.3781     | 0.6667        |
> | **Reasoning LLMs**           | **DeepSeek-R1**        | 0.0200       | -0.9087      | 0.0077            | 0.0650     | -0.8700    | 0.0603        |
> | **Reasoning LLMs**           | **o3-mini**            | 0.3081       | 0.0585       | 0.2106            | 0.7093     | 0.4188     | **0.7044**    |
> | **Reasoning LLMs**           | **Qwen3-30B**          | 0.5509       | -0.1892      | 0.0484            | 0.4429     | -0.1144    | 0.4323        |
> | **Reasoning LLMs**           | **Qwen3-235B**         | 0.5109       | 0.0408       | 0.2017            | 0.4300     | -0.1404    | 0.4071        |
> | **Reasoning LLMs**           | **GPT-5**              | 0.3787       | -0.1780      | 0.1067            | 0.6921     | 0.3843     | 0.6901        |
> | **Medical LLM**              | **Baichuan-M2**        | 0.1713       | 0.0598       | 0.2104            | 0.6800     | 0.3601     | 0.6836        |
> | **LLM-based Agents**         | **Qwen2.5-72B+Tools**  | 0.3245       | -0.0321      | 0.1801            | 0.6807     | 0.3632     | 0.6642        |
> | **LLM-based Agents**         | **DeepSeekV3+Tools**   | 0.8183       | -0.0199      | 0.0571            | 0.6779     | 0.3558     | 0.6810        |
> | **LLM-based Agents**         | **DeepSeekV3.1+Tools** | 0.2312       | -0.2295      | 0.1236            | 0.6600     | 0.3210     | 0.6730        |
> | **LLM-based Agents**         | **GPT-4.1+Tools**      | 0.5050       | 0.0748       | 0.2192            | 0.5929     | 0.2106     | 0.4673        |
> | **LLM-based Agents**         | **DeepSeek-R1+Tools**  | 0.1418       | 0.0210       | 0.2004            | 0.1050     | -0.7900    | 0.1044        |
> | **LLM-based Agents**         | **o3-mini + Tools**    | 0.4319       | 0.0381       | 0.2029            | 0.7157     | 0.4349     | 0.6966        |
> | **LLM-based Agents**         | **Qwen3-30B+Tools**    | 0.2721       | -0.2095      | 0.1197            | 0.4579     | -0.0847    | 0.4840        |
> | **LLM-based Agents**         | **Qwen3-235B+Tools**   | 0.6003       | 0.0626       | 0.2105            | 0.6757     | 0.3527     | 0.6890        |
> | **LLM-based Agents**         | **o3-mini+Code**       | 0.4386       | 0.0430       | 0.2051            | 0.7085     | 0.4215     | 0.6862        |
> | **RCA**                      | **RCA+Qwen2.5**        | 0.7069       | **0.1053**   | **0.2338**        | **0.7177** | **0.4476** | 0.6931        |
> | **RCA**                      | **RCA+GPT-4.1**        | **0.8335**   | 0.0564       | 0.1463            | 0.7107     | 0.4285     | 0.6818        |
>
> The experimental results show that our method is also effective on these two new datasets.
>
> We hope that our explanations and newly added experiments could solve your question. If you have any further questions, feel free to contact us.

---

> ### Author Response · Authors · 2025-11-20
>
> Your feedback is extremely valuable to us, and if you have any further questions or thoughts, we are always happy to discuss them with you!
>
> Sincerely, The Authors

---

> ### Author Response · Authors · 2025-11-26
> **A quick request for your valuable feedback**
>
> Dear Reviewer Xy3E,
>
> Thank you once again for your thoughtful suggestions, which have been very helpful in refining our work.
>
> As the discussion period continues, we wanted to ensure our responses were helpful and to ask if there are any lingering concerns we can address. We are committed to making this paper as strong as possible and welcome any further feedback you might have.
>
> We sincerely strive to earn your continued recognition for our efforts.
>
> Sincerely,
>
> Authors

---

> ### Comment · Reviewer_Xy3E · 2025-11-26
> **I thank the authors fro the extensive answer. I increased my rating.**
>
> I thank the authors for their extensive and thoughtful response. I have increased my rating, primarily because I initially misunderstood how some of the baselines were evaluated—specifically, whether they were run in a zero-shot manner or with additional fine-tuning. The clarification resolved that concern.
>
> That said, I still have reservations regarding the theoretical soundness of the approach, particularly the justification and validation of the four explanation criteria. I also remain uncertain about how well the proposed framework would generalize beyond the datasets examined in the current study, as well as the potential for bias amplification.

---

> > ### Author Response · Authors · 2025-11-26
> > **Further Response to Review's Question : Part 2**
> >
> > We are very grateful for your valuable feedback and for engaging with our responses. If you have any other questions, we would be more than happy to provide further clarification. Our code is fully open-sourced at https://anonymous.4open.science/r/anonym107 to ensure transparency and reproducibility. We hope that these additional explanations might fully address your remaining concerns, and we would be honored if you would further consider our work favorably.
> >
> > Sincerely,
> >
> > The Authors

---

> > ### Author Response · Authors · 2025-11-28
> >
> > Dear reviewer Xy3E,
> >
> > We have incorporated your suggestions and revised the paper. For a detailed breakdown of the modifications, please refer to the 'Summary of revisions' we added above.
> >
> > If you have any further questions, feel free to contact us. Thank you again for your time and guidance.

---

> ### Author Response · Authors · 2025-11-26
> **Further Response to Review's Question : Part 1**
>
> Dear Reviewer Xy3E,
>
> Thank you so much for your thoughtful follow-up and for raising your rating. We truly appreciate your time and the feedback you've provided throughout this process. We also appreciate you sharing your remaining reservations. They are important points, and we would like to offer some further clarification in the hope of fully resolving your concerns.
>
> > the justification and validation of the four explanation criteria
>
> Regarding the justification and validation of our four explanation criteria, we hope we can elaborate on the points we shared in our response to your "Weakness 3 & Question 3" concern. **First**, we did our best to ground these metrics in established literature from cognitive science and medical practice, and they were validated as "clinically valuable" by our collaborating doctors. **Second**, we recognize that evaluating natural language explanations is inherently subjective, and we hope you can appreciate that the initial scoring process was a significant undertaking for the clinicians, involving a great deal of their time and effort. **Third**, to more formally address this very challenge, we followed the methodology of a study in Nature Machine Intelligence [1] and conducted an additional user study. This study, detailed in our previous response "Weakness 3 & Question 3", directly compared our explanations against the strongest baseline in a head-to-head preference test. We sincerely hope that this combination of a grounded methodology and the additional formal user study helps to address your concerns.
>
> > generalize beyond the datasets examined in the current study
>
> Regarding the concern about the framework's ability to generalize beyond the datasets examined, we aimed to demonstrate its broad applicability. As we detailed in our response to your "Question 5", in addition to the initial three datasets (two public and one private for real-world validation), we conducted further experiments on two new, large-scale datasets. These included an expanded version of our private CRT dataset and a large public cardiovascular disease dataset with 70,000 patient records. We sincerely hope that these additional results on larger and more diverse datasets provide stronger evidence for RCA's potential to generalize.
>
> > the potential for bias amplification
>
> We also understand that our previous response regarding the potential for bias amplification may not have been fully convincing, and we would like to offer a more detailed perspective. **First**, we want to re-emphasize that our paper’s central thesis is the synergy between high accuracy and high-quality explanations. The robustness mechanism, while important, is a secondary design consideration, added as a pragmatic feature to ensure our framework is effective in real-world clinical settings where data is often imperfect. **Second**, our approach to this is grounded in a first-principles understanding of the problem. By definition, an "outlier" or "noise" is a data point that deviates significantly from the overall data distribution. Therefore, any method that aims to be robust against such data must, in some way, consider this distribution. Our design makes this process explicit and elegant. Instead of adding complex, specialized filters, we simply provide the automatically computed statistics of the training data as context. **Crucially**, this global distribution is not a hand-crafted prior or a form of weak supervision; it is calculated directly from the real, and potentially noisy, training data. This makes our approach general-purpose and not a case-by-case solution for medicine. The framework doesn't blindly follow these statistics. They serve as a "sanity check" for our checking LLM, which uses its reasoning abilities to weigh evidence from specific errors against the statistical reality of the entire dataset. This is what makes the process both simple and effective. **Finally**, as shown in our ablation study, RCA's core iterative learning mechanism has a degree of inherent robustness even without this explicit check, because it naturally learns from batches of errors rather than overreacting to single instances.
>
> In summary, we followed the principle of simple and direct usage to enhance the robustness of RCA. We also validated the effectiveness of this straightforward approach through extensive experiments. This comparison is also fair, as we also provided the same distribution to all baselines and used the same prompt for explanation. We sincerely hope this can solve your concern.
>
> We will update the paper's PDF within the next two days to further clarify this point. Our revisions will be fully detailed in our 'Summary of Revisions'.
>
> [1] Slack, D., Krishna, S., Lakkaraju, H. and Singh, S., 2023. Explaining machine learning models with interactive natural language conversations using TalkToModel. Nature Machine Intelligence, 5(8), pp.873-883.

---

### Official Review · Reviewer_xdsQ · 2025-10-30

**Soundness:** 3
**Presentation:** 4
**Contribution:** 3
**Rating:** 8
**Confidence:** 3

**Summary:**

The paper proposes Reflective Cognitive Architecture (RCA), a framework that leverages multiple instantiations of a large language model to iteratively generate improved predictions along with corresponding explanations. The authors evaluate their approach against several strong baselines on three real-world datasets. They further test the framework under challenging conditions with noisy or suboptimal data, demonstrating consistent and significant improvements over all baselines across these scenarios.

**Strengths:**

I find this an interesting approach that leverages additional data to enhance LLM performance on specialised, data-grounded reasoning tasks. I particularly appreciate the idea of using misclassifications as a mechanism to dynamically update the set of rules followed by the model. The paper is very well written and exceptionally easy to follow. Moreover, the experimental setup is extensive and thorough, evaluating multiple metrics against common baselines and demonstrating that the method remains robust even under moderately noisy conditions. Each component of the proposed framework is shown to contribute meaningfully to the overall performance.

**Weaknesses:**

- Since the rule extraction process relies heavily on the available data, the model may become more susceptible to exploiting dataset-specific biases. It would be interesting to analyse how the proposed framework behaves under such conditions. One potential mitigation could be to complement the data-driven rule extraction with external knowledge sources, for example, a curated corpus of research papers, to guide the reasoning process using human-grounded knowledge.
- The comparison with standard LLMs (and, to some extent, with tool-augmented models) may not be entirely fair, as those models are not "trained" or exposed to the same data used in the proposed framework. This difference in data exposure could partially explain the observed performance gap. It would be useful to clarify this distinction and discuss its implications for the fairness of the comparison.

**Questions:**

- How would the proposed method perform on datasets with strongly biased features? Have you considered evaluating the robustness of your framework under such conditions?
- Given that the baseline LLMs were not exposed to the same training data used in your approach, how do you address the issue of fairness in the comparison? Do you think this difference in data access could have influenced the observed performance gap?

---

> ### Author Response · Authors · 2025-11-20
>
> Thank you for the detailed and constructive feedback on our work. We were very encouraged that Reviewer xdsQ recognized the merits of RCA's novelty, our comprehensive empirical studies, and the formal approach to evaluating explanations. Your insightful questions and pointed-out weaknesses were instrumental in helping us strengthen the paper. We have thoroughly revised the paper in response to your suggestions and will outline the specific changes for each point below.

---

> ### Author Response · Authors · 2025-11-20
> **Weakness 1**
>
> > **Weakness 1: Since the rule extraction process relies heavily on the available data, the model may become more susceptible to exploiting dataset-specific biases. It would be interesting to analyse how the proposed framework behaves under such conditions. One potential mitigation could be to complement the data-driven rule extraction with external knowledge sources, for example, a curated corpus of research papers, to guide the reasoning process using human-grounded knowledge.**
>
> Thank you for this insightful and important point. This is a critical consideration, and we gave this trade-off significant thought during our design process.
>
> You are absolutely correct that any purely data-driven rule extraction process risks learning and exploiting dataset-specific biases. Your suggestion to complement this with external knowledge sources, like research papers, is an excellent and very valid approach to guide the model with human-grounded knowledge.
>
> Our primary motivation for RCA, however, stemmed from a different core observation. We found that modern LLMs, due to their vast pre-training, **already possess an enormous amount of general medical knowledge**. They are almost all capable of generating fluent, medically-plausible-sounding explanations.
>
> The problem, as our baseline results show, is that they often do this badly.The critical gap we identified is not a lack of *general* medical knowledge which external papers would supplement, but a lack of *deep, specific understanding of the dataset at hand*. They fail to grasp the unique statistical thresholds, feature interactions, and cohort-specific patterns present in the *specific* data. This is precisely why standard LLMs produce "statistically unsupported" narratives that are not grounded in the evidence of the dataset itself.
>
> Therefore, RCA was explicitly designed to solve this **data-understanding gap**, not a general knowledge gap. It forces the model to learn from direct "experience" with the data, using accuracy as a signal to build a robust internal model. Our results show that RCA's explanations are rated higher for Evidence-Based Medicine (EBM) and are more robust to data noise. This suggests that this deep data-grounding is the more pressing issue for this task.
>
> That said, your point remains highly relevant. In scenarios involving very rare diseases or novel biomarkers where the LLM may *not* have sufficient pre-trained knowledge, your suggested approach of integrating an external knowledge corpus would be an excellent and likely necessary solution to establish a foundational understanding. We appreciate this valuable direction and have added a note of this to our Conclusion (**Section 5**).

---

> ### Author Response · Authors · 2025-11-20
> **Weakness 2 & Question 2 : Part 1**
>
> > **Weakness 2: The comparison with standard LLMs (and, to some extent, with tool-augmented models) may not be entirely fair, as those models are not "trained" or exposed to the same data used in the proposed framework. This difference in data exposure could partially explain the observed performance gap. It would be useful to clarify this distinction and discuss its implications for the fairness of the comparison.**
> >
> > **Question 2: Given that the baseline LLMs were not exposed to the same training data used in your approach, how do you address the issue of fairness in the comparison? Do you think this difference in data access could have influenced the observed performance gap?**
>
> Thank you for raising this extremely important point about experimental fairness. We sincerely apologize if our paper was not clear on this, as it's a critical detail of our evaluation.
>
> We would like to respectfully clarify that the comparison is fair, as the baseline LLMs were exposed to the same training data and information as our RCA framework.
>
> To be specific, the baseline LLMs were not just given a single test instance in a zero-shot manner. They were provided with the same global data distribution $\mathcal{D}_{train}$ and illustrative in-context examples from the training set that our framework used. Thus, both RCA and the baselines had access to the identical pool of information.
>
> The critical difference—and the very core of our paper's contribution—is not in *data access* but in the *process* of learning from it.
>
> - **Baseline LLMs** attempt to process this information in a single, in-context pass, which, as our results show, is often insufficient for deep reasoning, even equipped with code interpreter or more advanced tools for analysis.
> - **RCA**, by contrast, undergoes an explicit Self-reflection Process, which is its training. It iteratively learns from its prediction errors $S^{error}$ and actively builds a robust, internally-consistent rule base.
>
> In fact, when presenting the baseline results, we selected the optimal performance achieved by each baseline method, regardless of the variation in data exposure per group. Even when reporting these best-case results, RCA still exhibited the superior performance.
>
> To further address any potential concerns, we have added two more baselines: a neuro-symbolic network and a Medical LLM specifically pretrained on extensive healthcare data.
>
> - **Neural-Symbolic Baselines:** We have added two representative neural-symbolic models, **Logic Tensor Networks (LTN) [1]** and **Logic Neural Networks (LNN) [2]**, to our comparison (see **Section 4.1**). These models are designed to bridge the gap between accuracy and interpretability, making them highly relevant baselines.
> - **Medical LLM Baseline:** We have also benchmarked RCA against a state-of-the-art, domain-specific medical LLM: **Baichuan-M2 [3]**.
>
> Experiment results are shown as below:
>
> **Prediction Results**
>
> | Datasets                     |    CRT     |            |              |  Diabetes  |            |              | Heart Disease |            |              |
> | :--------------------------- | :--------: | :--------: | :----------: | :--------: | :--------: | :----------: | :-----------: | :--------: | :----------: |
> | **Algorithm**                |  **Acc**   |  **MCC**   | **F1-score** |  **Acc**   |  **MCC**   | **F1-score** |    **Acc**    |  **MCC**   | **F1-score** |
> | **Neural-symbolic Networks** |            |            |              |            |            |              |               |            |              |
> | LTN                          |   0.5161   |   0.0085   |    0.1176    |   0.7635   |   0.5247   |    0.6946    |    0.6010     |  -0.0386   |    0.2222    |
> | LNN                          |   0.2581   |   0.1287   |    0.1481    |   0.7437   |   0.4974   |    0.6787    |    0.5389     |  -0.0598   |    0.2393    |
> | **Medical LLM**              |            |            |              |            |            |              |               |            |              |
> | Baichuan-M2                  |   0.1587   |   0.0845   |    0.1311    |   0.6988   |   0.2772   |    0.4186    |    0.0725     |  -0.7779   |    0.1095    |
> | **RCA**                      |            |            |              |            |            |              |               |            |              |
> | RCA+Qwen2.5                  |   0.7778   | **0.1739** |  **0.2222**  | **0.7831** | **0.5406** |  **0.7097**  |    0.5647     |   0.0547   |    0.1290    |
> | RCA+GPT-4.1                  | **0.8730** |   0.1373   |    0.2000    |   0.7470   |   0.4244   |    0.6038    |  **0.7461**   | **0.1493** |    0.2898    |

---

> ### Author Response · Authors · 2025-11-20
> **Weakness 2 & Question 2 : Part 2**
>
> **Explanation Results: CRT Dataset**
>
> | Metrics                      |    CL    |      |    LA    |      |   EBM    |      |    CB    |      |
> | :--------------------------- | :------: | :--: | :------: | :--: | :------: | :--: | :------: | :--: |
> | **Algorithms**               |  Mean.   | Var. |  Mean.   | Var. |  Mean.   | Var. |  Mean.   | Var. |
> | **Neural-symbolic Networks** |          |      |          |      |          |      |          |      |
> | LTN                          |   7.47   | 0.23 |   8.25   | 0.32 |   8.05   | 0.61 |   7.27   | 0.59 |
> | LNN                          |   7.51   | 0.28 |   8.35   | 0.33 |   7.63   | 0.74 |   7.06   | 0.54 |
> | **Medical LLM**              |          |      |          |      |          |      |          |      |
> | Baichuan-M2                  |   7.82   | 0.54 |   8.38   | 0.52 |   8.32   | 0.75 |   6.96   | 0.55 |
> | **RCA**                      |          |      |          |      |          |      |          |      |
> | RCA+Qwen2.5                  | **8.24** | 0.42 | **8.89** | 0.54 |   8.47   | 0.92 |   7.61   | 0.74 |
> | RCA+GPT-4.1                  |   8.16   | 0.34 |   8.59   | 0.34 | **8.87** | 0.76 | **7.62** | 0.56 |
>
> **Explanation Results: Diabetes Dataset**
>
> | Metrics                      |    CL    |      |    LA    |      |   EBM    |      |    CB    |      |
> | :--------------------------- | :------: | :--: | :------: | :--: | :------: | :--: | :------: | :--: |
> | **Algorithms**               |  Mean.   | Var. |  Mean.   | Var. |  Mean.   | Var. |  Mean.   | Var. |
> | **Neural-symbolic Networks** |          |      |          |      |          |      |          |      |
> | LTN                          |   7.68   | 0.35 |   8.22   | 0.29 |   8.28   | 0.33 |   6.18   | 0.43 |
> | LNN                          |   7.82   | 0.33 |   8.35   | 0.27 |   8.45   | 0.31 |   6.28   | 0.39 |
> | **Medical LLM**              |          |      |          |      |          |      |          |      |
> | Baichuan-M2                  |   7.79   | 0.40 |   8.18   | 0.33 |   8.52   | 0.37 |   6.05   | 0.46 |
> | **RCA**                      |          |      |          |      |          |      |          |      |
> | RCA+Qwen2.5                  | **8.13** | 0.36 | **8.57** | 0.51 | **8.74** | 0.54 |   6.43   | 0.43 |
> | RCA+GPT-4.1                  |   8.03   | 0.43 |   8.38   | 0.49 |   8.63   | 0.62 | **6.94** | 0.40 |
>
> **Explanation Results: Heart Disease Dataset**
>
> | Metrics                      |    CL    |      |    LA    |      |   EBM    |      |    CB    |      |
> | :--------------------------- | :------: | :--: | :------: | :--: | :------: | :--: | :------: | :--: |
> | **Algorithms**               |  Mean.   | Var. |  Mean.   | Var. |  Mean.   | Var. |  Mean.   | Var. |
> | **Neural-symbolic Networks** |          |      |          |      |          |      |          |      |
> | LTN                          |   7.45   | 0.38 |   8.15   | 0.31 |   8.35   | 0.35 |   6.20   | 0.45 |
> | LNN                          |   7.65   | 0.36 |   8.32   | 0.29 |   8.58   | 0.33 |   6.35   | 0.41 |
> | **Medical LLM**              |          |      |          |      |          |      |          |      |
> | Baichuan-M2                  |   7.28   | 0.42 |   8.08   | 0.35 |   8.25   | 0.39 |   5.85   | 0.48 |
> | **RCA**                      |          |      |          |      |          |      |          |      |
> | RCA+Qwen2.5                  |   7.62   | 0.28 |   8.47   | 0.33 | **8.94** | 0.49 |   6.18   | 0.68 |
> | RCA+GPT-4.1                  | **7.74** | 0.35 | **8.53** | 0.42 |   8.79   | 0.55 | **6.42** | 0.53 |
>
> The tables indicate that neural-symbolic methods offer commendable predictive performance, and Medical LLMs attain high scores for explanation quality. Nevertheless, RCA consistently demonstrates a superior balance in achieving both high predictive accuracy and overall explanation quality compared to these approaches.
>
> We hope our explanations and newly added experiments could solve your question. If you have new further questions, feel free to contact us.
>
> >  [1] Badreddine, Samy, Artur d'Avila Garcez, Luciano Serafini, and Michael Spranger. "Logic tensor networks." *Artificial Intelligence* 303 (2022): 103649.
> >
> >  [2] Riegel, Ryan, Alexander Gray, Francois Luus, Naweed Khan, Ndivhuwo Makondo, Ismail Yunus Akhalwaya, Haifeng Qian et al. "Logical neural networks." *arXiv preprint arXiv:2006.13155* (2020).
> >
> >  [3] Dou, Chengfeng, Chong Liu, Fan Yang, Fei Li, Jiyuan Jia, Mingyang Chen, Qiang Ju et al. "Baichuan-m2: Scaling medical capability with large verifier system." *arXiv preprint arXiv:2509.02208* (2025).

---

> ### Author Response · Authors · 2025-11-20
> **Question 1**
>
> > **Question 1: How would the proposed method perform on datasets with strongly biased features? Have you considered evaluating the robustness of your framework under such conditions?**
>
> Thank you for this question. When designing RCA, the "strong bias features" we considered were indeed the 10% feature shift, 10% missing values, and full-column loss, as detailed in our robustness experiments.
>
> This 10% threshold was determined through our consultations with clinicians, and we believe it effectively covers the vast majority of error scenarios encountered in real-world hospital data. For context, the incidence of such shifts or missing values in our own CRT dataset is approximately 5%.
>
> Regarding the introduction of biases of a much greater magnitude, we acknowledge that this was not explored in the current study. Our rationale is that when a bias becomes so prevalent, it arguably ceases to be "noise" and transforms into a "feature" of the data itself. This poses a fundamental challenge that is not unique to our method but is shared by virtually all anomaly detection approaches. A conflict arises because detecting a feature with such a large bias is at odds with the very definition of an "outlier" or "anomaly." In such extreme cases, we believe the issue is best resolved by domain experts.
>
> In summary, we have made a concerted effort to test our method against conditions far more challenging than those typically seen in practice, demonstrating its advantages in generating both predictions and useful explanations. We hope this clarifies the scope and reasoning behind our approach.

---

> ### Author Response · Authors · 2025-11-20
>
> Thank you once again for your valuable and constructive feedback. Your suggestions have helped us identify important areas for improvement and future exploration. We hope our responses have adequately addressed your concerns.
>
> If you have any questions, feel free to contact us!

---

> > ### Comment · Reviewer_xdsQ · 2025-11-24
> >
> > Thank you for the answers you provided. I am happy to maintain my (already) high score.

---

> > > ### Author Response · Authors · 2025-11-24
> > >
> > > We are absolutely thrilled to receive your message! Thank you so much for your positive feedback and continued support for our paper.
> > >
> > > Throughout this process, your incisive questions and brilliant suggestions pushed us to refine our arguments and have significantly strengthened the paper. We are deeply grateful for your championship of our work and for your generous investment of time and intellect.
> > >
> > > Wishing you a wonderful day.
> > >
> > > Sincerely,
> > >
> > > the authors

---

### Official Review · Reviewer_Vm76 · 2025-10-31

**Soundness:** 2
**Presentation:** 3
**Contribution:** 2
**Rating:** 4
**Confidence:** 3

**Summary:**

This paper proposes a novel Reflective Cognitive Architecture (RCA) framework to address the important issue in AI-assisted diagnosis, where doctors require both high predictive accuracy and truth-worthy explanations. RCA has two main modules: an iterative rule refinement mechanism and a distribution-aware rules check mechanism. Such modules are implemented with LLMs, where a reflection LLM learns from prediction errors to update a "long-term memory" rule base, and a checking LLM validates these rules against the dataset's global statistics to ensure they are evidence-based and robust. Finally, a prediction LLM generates the predictions and explanations based on the optimized rule base as well as the global data distribution.

For the experiment, the authors evaluate RCA on three medical datasets, including a private one and two public ones. The authors proposed four metrics to evaluate the quality of explorations, and invited three doctors to score each explanation on a scale of
1 to 10 for each criterion. The results compared with 22 baselines claim that RCA achieved the best prediction accuracy and the explanation quality.

**Strengths:**

1. The paper focused on the important interprebabilty question in AI healthcare area. The proposed model is noval, consisting of learning between three LLMs. The authors rethink predictive accuracy not just as an end goal, but as a crucial reward signal that drives the model to build a more robust and fundamental "experience" with the underlying patterns.

2. The paper propose to quantify the explanations with four criterions, i.e., low CL, sound LA, EBM, reduced CB, which can be widely-used in AI Interpretable research.

3. The paper conducts comprehensive experiments on three datasets and 22 baselines, toghther with ablation study and qualitative analysis. The code and prompt is provided for reproducibility.

**Weaknesses:**

1. The paper mentioned two parts in the related work, the traditional ML methods and LLM methods. However, neural-symbolic is also a very related research line. Some works in neural-symbolic AI also try to solve the gap between prediction accuracy and meaningful explanations. Such work can be added to the related work as well as the baselines. Meanwhile, there are some medical LLM like Med-Gemini. More baselines on such domain specific LLMs are recommended.

2. The rules are not showed in the paper. As the key component of this paper are highly-related to rules refinement, some motivating examples of rules are important to understand the model effectiveness. Meanwhile, how to check the justifiability of such rules can be explained. For example, we can invite domain experts to check, or compare with public guidelines.

3. The paper also does not show the detailed example of global data distribution.

4. All the reported results in the table don't have variance. For instance, three doctors are invited to score the explanations, so the variance of these three doctors can be reported.

5. The results on Table 3-5 shows the score of explanations from three doctors. The rubric of such scores are in A.7. However, the rubric only has wide ranges like 5-7/7-10 points, however, in the Table 3-5, we can find that most scores of each baseline models and RCA can fall into the same range. Thus, it is not convincing that he small difference of explanations can indicate the quality of interpretability.

**Questions:**

1. The figure 1 can be polished. For instance, some key components like the meaning of $R_k$ and $R_f$ can be added in the caption or in the figure.

2. The meaning of 'conc' is not defined in the paper.

3. The case study in 4.5 can include more details. For example, what are the rules during the self-reflection process? What is the score of explanations for these four metrics.

4. The location of Reproducibility Statement is recommended at the end of the main text (before references).

---

> ### Author Response · Authors · 2025-11-20
>
> We sincerely thank the Reviewer for their detailed and constructive feedback. We are encouraged that the reviewer recognized the importance of our work, the novelty of our proposed RCA framework, and the comprehensiveness of our experiments. We appreciate the insightful suggestions, which have helped us to significantly improve our paper.

---

> ### Author Response · Authors · 2025-11-20
> **Weakness 1 : Part 1**
>
> > **Weakness 1: The paper mentioned two parts in the related work, the traditional ML methods and LLM methods. However, neural-symbolic is also a very related research line. Some works in neural-symbolic AI also try to solve the gap between prediction accuracy and meaningful explanations. Such work can be added to the related work as well as the baselines. Meanwhile, there are some medical LLM like Med-Gemini. More baselines on such domain specific LLMs are recommended.**
>
> Thank you for this good suggestion. We agree that including baselines from the neural-symbolic research line and domain-specific medical LLMs can provide a more comprehensive and robust evaluation of our method.
>
> To address this, we have conducted extensive new experiments and integrated them into the revised paper:
>
> - **Neural-Symbolic Baselines:** We have added two representative neural-symbolic models, **Logic Tensor Networks (LTN) [1]** and **Logic Neural Networks (LNN) [2]**, to our comparison (see **Section 4.1**). These models are designed to bridge the gap between accuracy and interpretability, making them highly relevant baselines. We input the final prediction and the truth values of networks' constituent logical predicates into a Qwen3-235B to generate a explanations.
> - **Medical LLM Baseline:** We have also benchmarked RCA against a state-of-the-art, domain-specific medical LLM: **Baichuan-M2 [3]**.
>
> We selected **Baichuan-M2** as our Medical LLM because it is the premier open-source medical LLM from Baichuan with a readily available API. We did not include **Med-Gemini** solely because we were unable to locate a public API or open-source model. However, if you could kindly provide guidance on how to access it, we would be delighted to conduct and incorporate those experiments.
>
> Furthermore, we have updated our **Related Work Section2.1** to discuss neural-symbolic methods and medical LLMs, including both Baichuan-M2 and Med-Gemini.
>
> All results from these new experiments have been fully incorporated in **Table 3** (prediction results across three dataset), **Table 4** (explanation results on CRT dataset) , **Table 5** (explanation results on Diabetes dataset) and **Table 6** (explanation results on Heart Disease dataset). The results for the newly added baselines are presented below.
>
>
>
> **Prediction Results**
>
> | Datasets                     |    CRT     |            |              |  Diabetes  |            |              | Heart Disease |            |              |
> | :--------------------------- | :--------: | :--------: | :----------: | :--------: | :--------: | :----------: | :-----------: | :--------: | :----------: |
> | **Algorithm**                |  **Acc**   |  **MCC**   | **F1-score** |  **Acc**   |  **MCC**   | **F1-score** |    **Acc**    |  **MCC**   | **F1-score** |
> | **Neural-symbolic Networks** |            |            |              |            |            |              |               |            |              |
> | LTN                          |   0.5161   |   0.0085   |    0.1176    |   0.7635   |   0.5247   |    0.6946    |    0.6010     |  -0.0386   |    0.2222    |
> | LNN                          |   0.2581   |   0.1287   |    0.1481    |   0.7437   |   0.4974   |    0.6787    |    0.5389     |  -0.0598   |    0.2393    |
> | **Medical LLM**              |            |            |              |            |            |              |               |            |              |
> | Baichuan-M2                  |   0.1587   |   0.0845   |    0.1311    |   0.6988   |   0.2772   |    0.4186    |    0.0725     |  -0.7779   |    0.1095    |
> | **RCA**                      |            |            |              |            |            |              |               |            |              |
> | RCA+Qwen2.5                  |   0.7778   | **0.1739** |  **0.2222**  | **0.7831** | **0.5406** |  **0.7097**  |    0.5647     |   0.0547   |    0.1290    |
> | RCA+GPT-4.1                  | **0.8730** |   0.1373   |    0.2000    |   0.7470   |   0.4244   |    0.6038    |  **0.7461**   | **0.1493** |    0.2898    |

---

> ### Author Response · Authors · 2025-11-20
> **Weakness 1 : Part 2**
>
> **Explanation Results: CRT Dataset**
>
> | Metrics                      |    CL    |      |    LA    |      |   EBM    |      |    CB    |      |
> | :--------------------------- | :------: | :--: | :------: | :--: | :------: | :--: | :------: | :--: |
> | **Algorithms**               |  Mean.   | Var. |  Mean.   | Var. |  Mean.   | Var. |  Mean.   | Var. |
> | **Neural-symbolic Networks** |          |      |          |      |          |      |          |      |
> | LTN                          |   7.47   | 0.23 |   8.25   | 0.32 |   8.05   | 0.61 |   7.27   | 0.59 |
> | LNN                          |   7.51   | 0.28 |   8.35   | 0.33 |   7.63   | 0.74 |   7.06   | 0.54 |
> | **Medical LLM**              |          |      |          |      |          |      |          |      |
> | Baichuan-M2                  |   7.82   | 0.54 |   8.38   | 0.52 |   8.32   | 0.75 |   6.96   | 0.55 |
> | **RCA**                      |          |      |          |      |          |      |          |      |
> | RCA+Qwen2.5                  | **8.24** | 0.42 | **8.89** | 0.54 |   8.47   | 0.92 |   7.61   | 0.74 |
> | RCA+GPT-4.1                  |   8.16   | 0.34 |   8.59   | 0.34 | **8.87** | 0.76 | **7.62** | 0.56 |
>
>
>
> **Explanation Results: Diabetes Dataset**
>
> | Metrics                      |    CL    |      |    LA    |      |   EBM    |      |    CB    |      |
> | :--------------------------- | :------: | :--: | :------: | :--: | :------: | :--: | :------: | :--: |
> | **Algorithms**               |  Mean.   | Var. |  Mean.   | Var. |  Mean.   | Var. |  Mean.   | Var. |
> | **Neural-symbolic Networks** |          |      |          |      |          |      |          |      |
> | LTN                          |   7.68   | 0.35 |   8.22   | 0.29 |   8.28   | 0.33 |   6.18   | 0.43 |
> | LNN                          |   7.82   | 0.33 |   8.35   | 0.27 |   8.45   | 0.31 |   6.28   | 0.39 |
> | **Medical LLM**              |          |      |          |      |          |      |          |      |
> | Baichuan-M2                  |   7.79   | 0.40 |   8.18   | 0.33 |   8.52   | 0.37 |   6.05   | 0.46 |
> | **RCA**                      |          |      |          |      |          |      |          |      |
> | RCA+Qwen2.5                  | **8.13** | 0.36 | **8.57** | 0.51 | **8.74** | 0.54 |   6.43   | 0.43 |
> | RCA+GPT-4.1                  |   8.03   | 0.43 |   8.38   | 0.49 |   8.63   | 0.62 | **6.94** | 0.40 |
>
>
>
> **Explanation Results: Heart Disease Dataset**
>
> | Metrics                      |    CL    |      |    LA    |      |   EBM    |      |    CB    |      |
> | :--------------------------- | :------: | :--: | :------: | :--: | :------: | :--: | :------: | :--: |
> | **Algorithms**               |  Mean.   | Var. |  Mean.   | Var. |  Mean.   | Var. |  Mean.   | Var. |
> | **Neural-symbolic Networks** |          |      |          |      |          |      |          |      |
> | LTN                          |   7.45   | 0.38 |   8.15   | 0.31 |   8.35   | 0.35 |   6.20   | 0.45 |
> | LNN                          |   7.65   | 0.36 |   8.32   | 0.29 |   8.58   | 0.33 |   6.35   | 0.41 |
> | **Medical LLM**              |          |      |          |      |          |      |          |      |
> | Baichuan-M2                  |   7.28   | 0.42 |   8.08   | 0.35 |   8.25   | 0.39 |   5.85   | 0.48 |
> | **RCA**                      |          |      |          |      |          |      |          |      |
> | RCA+Qwen2.5                  |   7.62   | 0.28 |   8.47   | 0.33 | **8.94** | 0.49 |   6.18   | 0.68 |
> | RCA+GPT-4.1                  | **7.74** | 0.35 | **8.53** | 0.42 |   8.79   | 0.55 | **6.42** | 0.53 |
>
> The tables indicate that neural-symbolic methods offer commendable predictive performance, and Medical LLMs attain high scores for explanation quality. Nevertheless, RCA consistently demonstrates a superior balance in achieving both high predictive accuracy and overall explanation quality compared to these approaches.
>
> >  [1] Badreddine, Samy, Artur d'Avila Garcez, Luciano Serafini, and Michael Spranger. "Logic tensor networks." *Artificial Intelligence* 303 (2022): 103649.
> >
> >  [2] Riegel, Ryan, Alexander Gray, Francois Luus, Naweed Khan, Ndivhuwo Makondo, Ismail Yunus Akhalwaya, Haifeng Qian et al. "Logical neural networks." *arXiv preprint arXiv:2006.13155* (2020).
> >
> >  [3] Dou, Chengfeng, Chong Liu, Fan Yang, Fei Li, Jiyuan Jia, Mingyang Chen, Qiang Ju et al. "Baichuan-m2: Scaling medical capability with large verifier system." *arXiv preprint arXiv:2509.02208* (2025).

---

> ### Author Response · Authors · 2025-11-20
> **Weakness 2 & Weakness 3 & Question 3 : Part 1**
>
> > **Weakness 2:  The rules are not showed in the paper. As the key component of this paper are highly-related to rules refinement, some motivating examples of rules are important to understand the model effectiveness. Meanwhile, how to check the justifiability of such rules can be explained. For example, we can invite domain experts to check, or compare with public guidelines.**
> >
> > **Weakness 3: The paper also does not show the detailed example of global data distribution.**
> >
> > **Question 3: The case study in 4.5 can include more details. For example, what are the rules during the self-reflection process? What is the score of explanations for these four metrics.**
>
> We appreciate you pointing this out. This is truly important. We have made the following revisions to address this:
>
> 1. **During Inference: how to apply rules and distribution to generate explanation:**
>
>    In the revised paper, we have updated **Figure 3 in Section 4.5**. The text presented within the blue boxes in Figure 3 illustrates samples of the "Rules" and "Distribution" . Figure 3 demonstrates how the generated rules are applied to construct the explanatory statements.
>
>    Specifically，rules are shown as below:
>
>    ```markdown
>    - If Granulocyte-to-lymphocyte ratio is less than 2.0 and D-dimer is less than 1.5, and no history of previous CRT, predict no catheter-related thrombosis.
>    - D-dimer ≥2.0 raises CRT risk, but do NOT predict CRT based solely on high D-dimer without at least one moderate risk factor (recent chemotherapy, history of previous catheterization, thoracic therapy, or high-risk tumor type: lung, gastric, gynecologic, urologic, or lymphoma).
>    ```
>
>    The first rule demonstrates how RCA successfully identifies the quantitative ranges for GLR and D-dimer. The second rule shows that RCA successfully recognized the coupled relationship between D-dimer and multiple moderate risk factors.
>
>    Distribution is shown as below:
>
>    ```markdown
>    ……
>    D-dimer:
>    Mean: 1.21, 5th Percentile: 0.10, 25th Percentile: 0.29, 50th Percentile: 0.57, 75th Percentile: 1.12, 95th Percentile: 4.16
>    ……
>    ```
>
>    The distribution indicates that RCA effectively leverages D-dimer's statistical information  to rigorously validate the accuracy and generalizability of the rules.
>
> 2. **During training process, how rules are updated (updated in** **Appendix A.3 and below)**:
>
>    1. **Initial State (The Context)** Imagine the model is in the middle of training (Epoch $k$). It has learned general logic but still struggles with precise, borderline cases.
>
>       - **Current Rule Base (**$R^{k-1}$): Uses broad rules like **Rule A**: "Chemotherapy + Catheter + D-dimer > 0.50 mg/L $\rightarrow$ High Risk."
>
>       - **Global Data Distribution ($\mathcal{D}_{train}$**): The model holds the statistical ground truth. Crucially, it knows that for *No-CRT* patients in this cohort, the D-dimer 75th percentile is actually 1.12 (higher than the standard 0.50 cutoff), and GLR usually sits between 0.86 and 2.63.
>
>    2. **The Encounter (Prediction Error)**
>
>       The model encounters Patient 4:
>
>       ```markdown
>       Profile: Chemotherapy + PICC, but with low inflammation (GLR=0.94) and marginally elevated D-dimer (0.58).
>       True Label: No CRT.
>       ```
>
>       **The Mistake:** Relying on *Rule A*, the model sees D-dimer > 0.50 and incorrectly predicts "CRT." This is the "short-term experience" of failure.
>
>    3. **Iterative Rules Optimization (Bottom-Up Refinement)**
>
>       -  The Reflection LLM ($M_{ref}$) examines the error. It realizes Rule A was too strict: for this patient, D-dimer=0.58 was actually safe because inflammation was low.
>
>       -  To fix this, $M_{ref}$ creates a specific patch:
>
>          - Proposed Rule: "If GLR is 0.9–1.0 and D-dimer is 0.4–0.6, predict No CRT."
>
>          - This rule fixes the error for Patient 4 but is over-fitted—it is too specific to be useful generally.
>
>    4. **Distribution-aware Rules Check (Top-Down Safeguard)**
>
>       - At the end of the epoch, the Check LLM ($M_{chk}$) reviews this proposed rule against the global data ($\mathcal{D}_{train}$).
>
>       - It sees that the proposed D-dimer range (0.4–0.6) is well inside the "safe zone" (up to 1.12).
>
>       - It recognizes the specific GLR range is just a slice of the normal range (0.86–2.63).
>       - $M_{chk}$ transforms the narrow proposal into a robust, general rule.
>
>    5. **Final Refined Rule:** "If GLR is Normal (0.86–2.63) AND D-dimer is < 1.12, predict **No CRT**, regardless of Chemotherapy."
>
>    **Summary:**
>
>    * **Refinement** uses individual errors to find exceptions (discovering that the standard cutoff is wrong for this case).
>
>    - **Check** uses global stats to generalize that exception (confirming that *anything* under 1.12 is safe), ensuring the model learns robustly without memorizing data.

---

> > ### Comment · Reviewer_Vm76 · 2025-11-26
> >
> > Dear author,
> >
> > Thanks for your response. I have another concern about the usage of Distribution. From the example, I feel that the Distribution is mainly about some ranges, which are specifically sensitive in the healthcare domain. However, if the dataset is small or with some noise, and the Distribution/range may not be correct, then will it cause the wrong behavior for the whole work pipeline? I feel that such ranges work as weak supervision/prior knowledge, specifically designed for the medical problem.

---

> > > ### Author Response · Authors · 2025-11-26
> > > **Further Response to Review's Question**
> > >
> > > Dear Reviewer Vm76,
> > >
> > > We sincerely thank you for your insightful follow-up question. This is an excellent point that touches upon the practical robustness of our RCA framework, particularly regarding the role of the global data distribution when dealing with small or noisy datasets. We appreciate the opportunity to clarify our design philosophy and the specific function of this component.
> > >
> > > First, we would like to re-emphasize that the central thesis of our paper is the synergistic relationship between predictive accuracy and high-quality explanations, which we argue are dual outcomes of a model that develops a deep, experience-based understanding of the data. Our primary goal is to achieve state-of-the-art performance and clinical-grade explainability. The robustness mechanisms, while crucial for real-world application, are a secondary design consideration intended to enhance the practicality of our framework in authentic clinical settings, where data is rarely perfect.
> > >
> > > To address your specific concern, we want to clarify that the "Distribution" used in RCA is **not a form of pre-defined, clean prior knowledge or weak supervision specifically tailored for medical problems**. Instead, it is a statistical summary that is **automatically computed directly from the training data itself—including any noise or imperfections it may contain**. This is a deliberate design choice that mirrors real-world clinical practice, where physicians and researchers work with data as it is collected, not with an idealized, perfectly clean version. We do not want to impose the burden of complex data analysis and cleaning on clinicians, who are already constrained for time. The distribution-aware check is thus a pragmatic feature designed for usability, not a core dependency on pristine priors.
> > >
> > > Furthermore, the framework does not blindly follow the computed distribution. The distribution serves as a statistical context or a "sanity check" for the checking LLM ($M_{chk}$). When the reflection LLM ($M_{ref}$) proposes a new rule based on a batch of errors, the checking LLM uses the global distribution to reason whether this new rule reflects a genuine underlying pattern or is simply an over-reaction to noise or a few outliers. The large language model's inherent reasoning capabilities allow it to weigh the evidence from specific error instances against the statistical reality of the entire dataset. This process makes the rule refinement more robust against being skewed by noisy data points. In our experiments, we also provided this same statistical summary to all baseline models to ensure a fair and direct comparison.
> > >
> > > Finally, our approach is grounded in a first-principles understanding of outlier detection. The very definition of an "outlier" or "noise" is a data point that deviates significantly from the overall distribution. Therefore, any method that aims to be robust against such data must, in some way, consider this distribution. Our method makes this process explicit and leverages an LLM's reasoning to handle it, rather than relying on complex, pre-programmed statistical filters. It is a design choice for an elegant and principled solution. In fact, even without the distribution-aware check, RCA exhibits a degree of inherent robustness superior to baselines because its iterative mechanism reflects on a batch of errors, naturally balancing local signals with its evolving global understanding (the rule base). The results are shown in Table 1.
> > >
> > > In summary:
> > > 1. Our paper's core contribution is the synergy between accuracy and explainability. Robustness is a practical enhancement, not the central thesis.
> > > 2. The global distribution is not a hand-crafted prior but is automatically generated from the real (and potentially noisy) training data, making the approach general-purpose and not a case-by-case solution for medicine.
> > > 3. Our design is a principled approach to robustness, relying on the fundamental definition of outliers and the reasoning ability of LLMs, rather than adding complex, specialized modules.
> > >
> > > Thank you again for your question. It is an excellent one. We will update the paper's PDF within the next two days to further clarify this point. Our revisions will be fully detailed in our 'Summary of Revisions'.
> > >
> > > We hope this clarifies the role and nature of the distribution component in our framework. Thank you again for your valuable feedback. Our code  are fully open-sourced at  https://anonymous.4open.science/r/anonym107. If you have any further questions, feel free to contact us. We are always happy to answer any of your questions.
> > >
> > > Sincerely,
> > >
> > > The Authors

---

> > > ### Author Response · Authors · 2025-11-28
> > >
> > > Dear reviewer Vm76,
> > >
> > > We have carefully revised the paper accordingly. A detailed, point-by-point response and a summary of all changes can be found in the 'Summary of revisions' above for your convenience.
> > >
> > > All modifications have also been highlighted in the main PDF. Please do not hesitate to contact us if you have any further questions.

---

> ### Author Response · Authors · 2025-11-20
> **Weakness 2 & Weakness 3 & Question 3 : Part 2**
>
> **In response to your question on how to check the justifiability of such rules:**
>
> In practical clinical deployment, the most responsible approach involves submitting the final generated rule base to clinicians for review and correction. In specific scenarios, the rule base and the underlying data can also be cross-validated using methods like o3-mini integrated with coding tools. While the effectiveness of these methods may not match that of RCA , they serve as a valuable additional safeguard.
>
> We hope our explanations can solve your question. If you have any further questions, feel free to contact us.

---

> ### Author Response · Authors · 2025-11-20
> **Weakness 4 : Part 1**
>
> > **Weakness 4: All the reported results in the table don't have variance. For instance, three doctors are invited to score the explanations, so the variance of these three doctors can be reported.**
>
> This is an important point. Reporting variance is essential for statistical rigor, especially for human evaluation scores. We apologize for this omission.
>
> For our evaluation, 3 doctors scored 100 samples for each of the core methods: Traditional MLs (CatBoost), LLM-based Methods (GPT-4.1), Reasoning LLMs (Qwen3-235B), LLM-based Agents (Qwen3-235B + tools), and RCA (RCA+GPT-4.1). For all other methods, we used Qwen3-30B with a carefully crafted prompt that achieved 90.1% scoring agreement with doctors. We've added more details in **Section 4.1** and reported the mean and variance in **Table 4-6 (Appendix A.11)**. The experimental results are presented below：

---

> ### Author Response · Authors · 2025-11-20
> **Weakness 4 : Part 2**
>
> **Explanation experiment results on CRT dataset**
>
> |           Metrics            | Algorithms              | Mean. (CL) | Var. (CL) | Mean. (LA) | Var. (LA) | Mean. (EBM) | Var. (EBM) | Mean. (CB) | Var. (CB) |
> | :--------------------------: | :---------------------- | :--------: | :-------: | :--------: | :-------: | :---------: | :--------: | :--------: | :-------: |
> |     **Traditional MLs**      | **Lasso**               |    7.60    |   0.39    |    8.33    |   0.36    |    7.31     |    0.63    |    7.21    |   0.56    |
> |                              | **Catboost**            |    7.39    |   0.50    |    8.13    |   0.53    |    7.31     |    0.78    |    7.08    |   0.71    |
> | **Neural-symbolic Networks** | **LTN**                 |    7.47    |   0.23    |    8.25    |   0.32    |    8.05     |    0.61    |    7.27    |   0.59    |
> |                              | **LNN**                 |    7.51    |   0.28    |    8.35    |   0.33    |    7.63     |    0.74    |    7.06    |   0.54    |
> |    **LLM-based Methods**     | **Qwen2.5-72B**         |    7.92    |   0.96    |    8.16    |   1.23    |    8.25     |    1.31    |    7.14    |   1.07    |
> |                              | **DeepSeek-V3**         |    7.98    |   0.68    |    8.27    |   0.73    |    8.22     |    0.83    |    7.19    |   0.70    |
> |                              | **DeepSeek-V3.1**       |    7.41    |   0.89    |    8.11    |   1.02    |    8.13     |    1.28    |    6.73    |   1.02    |
> |                              | **GPT-4.1**             |    7.79    |   0.69    |    8.25    |   0.72    |    8.30     |    1.13    |    7.31    |   0.88    |
> |      **Reasoning LLMs**      | **DeepSeek-R1**         |    7.82    |   1.03    |    7.97    |   1.29    |    8.27     |    1.60    |    7.25    |   1.07    |
> |                              | **o3-mini**             |    6.84    |   0.96    |    7.22    |   1.23    |    6.24     |    1.62    |    6.19    |   1.21    |
> |                              | **o4-mini**             |    7.56    |   1.18    |    8.06    |   1.36    |    8.25     |    1.80    |    6.81    |   1.49    |
> |                              | **Qwen3-30B**           |    7.87    |   0.97    |    8.49    |   1.06    |    8.41     |    1.28    |    7.06    |   1.19    |
> |                              | **Qwen3-235B**          |    8.22    |   0.94    |    8.81    |   1.15    |    8.71     |    1.25    |    7.49    |   1.08    |
> |                              | **GPT-5**               |    7.63    |   0.97    |    8.22    |   1.06    |    8.40     |    1.32    |    7.05    |   1.05    |
> |       **Medical LLM**        | **Baichuan-M2**         |    7.82    |   0.54    |    8.38    |   0.52    |    8.32     |    0.75    |    6.96    |   0.55    |
> |     **LLM-Based Agents**     | **Qwen2.5-72B+Tools**   |    7.87    |   0.75    |    8.35    |   0.92    |    8.29     |    1.18    |    7.16    |   0.95    |
> |                              | **DeepSeek-V3.1+Tools** |    7.87    |   0.69    |    8.41    |   0.93    |    8.35     |    1.22    |    6.51    |   1.01    |
> |                              | **GPT-4.1+Tools**       |    7.89    |   0.48    |    8.30    |   0.54    |    8.32     |    0.76    |    7.35    |   0.67    |
> |                              | **DeepSeek-R1+Tools**   |    7.87    |   1.03    |    8.00    |   1.29    |    8.17     |    1.60    |    7.14    |   1.07    |
> |                              | **o3-mini+Tools**       |    6.22    |   0.91    |    7.30    |   1.07    |    6.20     |    1.31    |    6.16    |   1.15    |
> |                              | **o4-mini+Tools**       |    7.29    |   1.05    |    7.87    |   1.21    |    8.05     |    1.60    |    5.73    |   0.59    |
> |                              | **Qwen3-30B+Tools**     |    7.79    |   0.86    |    8.63    |   0.99    |    8.41     |    1.13    |    6.57    |   0.83    |
> |                              | **Qwen3-235B+Tools**    |    8.13    |   0.88    |    8.86    |   0.84    |    8.86     |    1.22    |    7.11    |   0.94    |
> |                              | **o3-mini+Code**        |    6.93    |   1.04    |    7.60    |   1.19    |    6.68     |    1.57    |    6.17    |   1.23    |
> |           **RCA**            | **RCA+Qwen2.5**         |  **8.24**  |   0.42    |  **8.89**  |   0.54    |    8.47     |    0.92    |    7.61    |   0.74    |
> |                              | **RCA+GPT-4.1**         |    8.16    |   0.34    |    8.59    |   0.34    |  **8.87**   |    0.76    |  **7.62**  |   0.56    |

---

> ### Author Response · Authors · 2025-11-20
> **Weakness 4 : Part 3**
>
> **Explanation experiment results on Diabetes dataset**
>
> |           Metrics            | Algorithms              | Mean. (CL) | Var. (CL) | Mean. (LA) | Var. (LA) | Mean. (EBM) | Var. (EBM) | Mean. (CB) | Var. (CB) |
> | :--------------------------: | :---------------------- | :--------: | :-------: | :--------: | :-------: | :---------: | :--------: | :--------: | :-------: |
> |     **Traditional MLs**      | **Lasso**               |    7.83    |   0.32    |    8.39    |   0.26    |    8.33     |    0.30    |    6.24    |   0.40    |
> |                              | **Catboost**            |    7.71    |   0.30    |    8.27    |   0.24    |    8.19     |    0.28    |    6.12    |   0.36    |
> | **Neural-symbolic Networks** | **LTN**                 |    7.68    |   0.35    |    8.22    |   0.29    |    8.28     |    0.33    |    6.18    |   0.43    |
> |                              | **LNN**                 |    7.82    |   0.33    |    8.35    |   0.27    |    8.45     |    0.31    |    6.28    |   0.39    |
> |    **LLM-based Methods**     | **Qwen2.5-72B**         |    7.61    |   0.92    |    8.20    |   0.80    |    8.10     |    0.87    |    5.98    |   1.02    |
> |                              | **DeepSeek-V3**         |    7.55    |   0.99    |    8.04    |   0.87    |    8.29     |    0.94    |    5.99    |   1.09    |
> |                              | **DeepSeek-V3.1**       |    7.85    |   0.95    |    8.23    |   0.83    |    8.12     |    0.90    |    6.04    |   1.05    |
> |                              | **GPT-4.1**             |    7.84    |   0.95    |    8.14    |   0.83    |    8.33     |    0.90    |    5.85    |   1.05    |
> |      **Reasoning LLMs**      | **DeepSeek-R1**         |    7.73    |   1.02    |    8.12    |   0.90    |    7.99     |    0.97    |    5.88    |   1.12    |
> |                              | **o3-mini**             |    7.67    |   1.05    |    8.12    |   0.93    |    8.16     |    1.00    |    5.89    |   1.15    |
> |                              | **o4-mini**             |    7.82    |   1.00    |    8.13    |   0.88    |    8.08     |    0.95    |    6.13    |   1.10    |
> |                              | **Qwen3-30B**           |    7.94    |   0.98    |    8.43    |   0.86    |    8.51     |    0.93    |    6.72    |   1.08    |
> |                              | **Qwen3-235B**          |    7.84    |   0.89    |    8.41    |   0.77    |    8.31     |    0.84    |    6.30    |   0.99    |
> |                              | **GPT-5**               |    7.66    |   1.05    |    8.24    |   0.93    |    8.53     |    1.00    |    6.24    |   1.15    |
> |       **Medical LLM**        | **Baichuan-M2**         |    7.79    |   0.40    |    8.18    |   0.33    |    8.52     |    0.37    |    6.05    |   0.46    |
> |     **LLM-Based Agents**     | **Qwen2.5-72B+Tools**   |    7.86    |   0.55    |    8.29    |   0.43    |    8.66     |    0.50    |    6.43    |   0.65    |
> |                              | **DeepSeek-V3.1+Tools** |    7.83    |   0.62    |    8.23    |   0.50    |    8.55     |    0.56    |    6.41    |   0.72    |
> |                              | **GPT-4.1+Tools**       |    7.81    |   0.56    |    8.25    |   0.44    |    8.01     |    0.51    |    6.12    |   0.66    |
> |                              | **DeepSeek-R1+Tools**   |    7.73    |   0.75    |    8.24    |   0.63    |    8.11     |    0.70    |    5.93    |   0.82    |
> |                              | **o3-mini+Tools**       |    7.62    |   0.79    |    8.25    |   0.66    |    8.13     |    0.73    |    5.90    |   0.86    |
> |                              | **o4-mini+Tools**       |    7.85    |   0.82    |    8.27    |   0.69    |    8.12     |    0.76    |    6.03    |   0.89    |
> |                              | **Qwen3-30B+Tools**     |    7.82    |   0.73    |    8.13    |   0.61    |    8.08     |    0.68    |    6.13    |   0.80    |
> |                              | **Qwen3-235B+Tools**    |    7.88    |   0.69    |    8.33    |   0.57    |    8.11     |    0.64    |    6.38    |   0.76    |
> |                              | **o3-mini+Code**        |    7.17    |   1.09    |    7.60    |   0.97    |    6.76     |    1.04    |    5.49    |   1.19    |
> |           **RCA**            | **RCA+Qwen2.5**         |  **8.13**  |   0.36    |  **8.57**  |   0.51    |  **8.74**   |    0.54    |    6.43    |   0.43    |
> |                              | **RCA+GPT-4.1**         |    8.03    |   0.43    |    8.38    |   0.49    |    8.63     |    0.62    |  **6.94**  |   0.40    |

---

> ### Author Response · Authors · 2025-11-20
> **Weakness 4 : Part 4**
>
> **Explanation experiment results on Heart Disease dataset**
>
> |           Metrics            | Algorithms              | Mean. (CL) | Var. (CL) | Mean. (LA) | Var. (LA) | Mean. (EBM) | Var. (EBM) | Mean. (CB) | Var. (CB) |
> | :--------------------------: | :---------------------- | :--------: | :-------: | :--------: | :-------: | :---------: | :--------: | :--------: | :-------: |
> |     **Traditional MLs**      | **Lasso**               |    7.60    |   0.35    |    8.27    |   0.28    |    8.63     |    0.32    |    5.94    |   0.42    |
> |                              | **Catboost**            |    7.50    |   0.33    |    8.21    |   0.25    |    8.72     |    0.29    |    5.94    |   0.38    |
> | **Neural-symbolic Networks** | **LTN**                 |    7.45    |   0.38    |    8.15    |   0.31    |    8.35     |    0.35    |    6.20    |   0.45    |
> |                              | **LNN**                 |    7.65    |   0.36    |    8.32    |   0.29    |    8.58     |    0.33    |    6.35    |   0.41    |
> |    **LLM-based Methods**     | **Qwen2.5-72B**         |    7.63    |   0.95    |    8.33    |   0.82    |    8.77     |    0.89    |    6.14    |   1.05    |
> |                              | **DeepSeek-V3**         |    7.52    |   1.02    |    8.25    |   0.89    |    8.62     |    0.96    |    6.05    |   1.12    |
> |                              | **DeepSeek-V3.1**       |    7.35    |   1.02    |    8.04    |   0.89    |    8.55     |    0.96    |    5.69    |   1.12    |
> |                              | **GPT-4.1**             |    7.45    |   0.98    |    8.08    |   0.85    |    8.64     |    0.92    |    5.77    |   1.08    |
> |      **Reasoning LLMs**      | **DeepSeek-R1**         |    7.40    |   1.05    |    7.98    |   0.92    |    8.59     |    0.99    |    5.75    |   1.15    |
> |                              | **o3-mini**             |    7.33    |   1.08    |    7.89    |   0.95    |    8.84     |    1.02    |    5.46    |   1.18    |
> |                              | **o4-mini**             |    7.31    |   1.03    |    7.83    |   0.90    |    8.73     |    0.97    |    5.38    |   1.13    |
> |                              | **Qwen3-30B**           |    7.41    |   1.01    |    8.06    |   0.88    |    8.49     |    0.95    |    5.82    |   1.11    |
> |                              | **Qwen3-235B**          |    7.67    |   0.92    |    8.34    |   0.79    |    8.84     |    0.86    |    6.36    |   1.02    |
> |                              | **GPT-5**               |    7.36    |   1.08    |    8.07    |   0.95    |    8.59     |    1.02    |    5.62    |   1.18    |
> |       **Medical LLM**        | **Baichuan-M2**         |    7.28    |   0.42    |    8.08    |   0.35    |    8.25     |    0.39    |    5.85    |   0.48    |
> |     **LLM-Based Agents**     | **Qwen2.5-72B+Tools**   |    7.63    |   0.58    |    8.34    |   0.45    |    8.74     |    0.52    |    6.00    |   0.68    |
> |                              | **DeepSeek-V3.1+Tools** |    7.37    |   0.65    |    7.97    |   0.52    |    8.46     |    0.58    |    5.72    |   0.75    |
> |                              | **GPT-4.1+Tools**       |    7.42    |   0.59    |    8.01    |   0.46    |    8.58     |    0.53    |    5.70    |   0.69    |
> |                              | **DeepSeek-R1+Tools**   |    7.38    |   0.78    |    7.87    |   0.65    |    8.55     |    0.72    |    5.73    |   0.85    |
> |                              | **o3-mini+Tools**       |    7.31    |   0.82    |    7.90    |   0.68    |    8.76     |    0.75    |    5.48    |   0.89    |
> |                              | **o4-mini+Tools**       |    7.35    |   0.85    |    7.96    |   0.71    |    8.51     |    0.78    |    5.66    |   0.92    |
> |                              | **Qwen3-30B+Tools**     |    7.39    |   0.76    |    7.94    |   0.63    |    8.56     |    0.70    |    5.73    |   0.83    |
> |                              | **Qwen3-235B+Tools**    |    7.58    |   0.72    |    8.28    |   0.59    |    8.91     |    0.66    |    6.20    |   0.79    |
> |                              | **o3-mini+Code**        |    7.28    |   1.02    |    7.88    |   0.99    |    8.67     |    1.06    |    5.38    |   1.12    |
> |           **RCA**            | **RCA+Qwen2.5**         |    7.62    |   0.28    |    8.47    |   0.33    |  **8.94**   |    0.49    |    6.18    |   0.68    |
> |                              | **RCA+GPT-4.1**         |  **7.74**  |   0.35    |  **8.53**  |   0.42    |    8.79     |    0.55    |  **6.42**  |   0.53    |

---

> ### Author Response · Authors · 2025-11-20
> **Weakness 4 : Part 5**
>
> Across all datasets, the explanations generated by our **RCA** framework exhibit **significantly lower variance** compared to all standalone LLM-based and reasoning-LLM baselines. This low variance indicates a high degree of inter-rater agreement among the clinicians who scored the explanations. It shows that the quality of RCA's explanations is not just high on average, but is also consistent, stable, and reliable from one sample to the next. In contrast, the much higher variance seen in the scores for other LLM-based methods suggests that their explanation quality is unpredictable. While they may occasionally produce a high-quality explanation, their performance is inconsistent, making them less trustworthy for critical clinical applications.

---

> ### Author Response · Authors · 2025-11-20
> **Weakness 5**
>
> > **Weakness 5: The results on Table 3-5 shows the score of explanations from three doctors. The rubric of such scores are in A.7. However, the rubric only has wide ranges like 5-7/7-10 points, however, in the Table 3-5, we can find that most scores of each baseline models and RCA can fall into the same range. Thus, it is not convincing that he small difference of explanations can indicate the quality of interpretability.**
>
> Thank you for your keen observation. We acknowledge that the margin in scores is not substantial.
>
> The reason we did not pursue further optimization was that the process of defining metrics and conducting the scoring had already been a very time-consuming and resource-intensive undertaking. The overhead for making additional adjustments from scratch would have been prohibitive, and we hope you can understand our rationale.
>
> To better address your concerns about usability, we were inspired by the work of Slack et al. [1] to conduct an additional user study. In this study, we formally compared the usability of the explanations. We asked three clinicians to evaluate the explanations from our method (RCA) and our strongest baseline (Qwen3-235B), choosing the "better" explanation based on our four established criteria.
>
> The results, now detailed in **Table 2 in Appendix A.11** and below:
>
> | Preference for RCA               |  CRT  | Diabetes | Heart Disease |
> | :------------------------------- | :---: | :------: | :-----------: |
> | **Cognitive Load(CL)**           | 82.7% |  81.3%   |     84.7%     |
> | **Logical Argumentation(LA)**    | 78.0% |  75.3%   |     72.0%     |
> | **Evidence-Based Medicine(EBM)** | 72.7% |  74.7%   |     73.3%     |
> | **Cognitive Biasing(CB)**        | 86.7% |  89.3%   |     86.0%     |
>
> The table shows that explanations from RCA led to higher preference. This provides formal evidence that RCA's explanations are not just qualitatively superior but are also more practically useful and trustworthy for human experts.
>
> We hope our  further explanations and experiments can solve your question. If you have further questions, please feel free to contact us.
>
> > [1] Slack, D., Krishna, S., Lakkaraju, H. et al. Explaining machine learning models with interactive natural language conversations using TalkToModel. Nat Mach Intell 5, 873–883 (2023).

---

> ### Author Response · Authors · 2025-11-20
> **Question 1**
>
> > **Question 1: The figure 1 can be polished. For instance, some key components like the meaning of and can be added in the caption or in the figure.**
>
> Thank you for this constructive feedback. We agree that the original caption for Figure 1 was brief and did not adequately explain the meaning of the core components and their corresponding symbols.
>
> To address this, we have **revised the caption for Figure 1** in the revised paper. The new caption now explicitly defines the key symbolic components of the self-reflection process. For instance, it clarifies that:
>
> - $S^{error}$ is the collection of misclassified samples (errors) from $M_{pred}$.
> - $R^k$ is the rule base for the current iteration $k$.
> - $\hat{R}^k$ denotes the checked and refined rule base from that iteration $k$.
>
> We hope this new, more descriptive caption, combined with the polished figure, could clearly explains the key components and the data flow of our architecture

---

> ### Author Response · Authors · 2025-11-20
> **Question 2**
>
> > **Question 2: The meaning of 'conc' is not defined in the paper.**
>
> Thank you for pointing out this lack of clarity. We apologize for this omission; you are correct that we failed to explicitly define the `conc` operator in the paper.
>
> In the context of equation $S^{error} = conc(s_{j}^{error})_{j=1}^T$, `conc` stands for **concatenation function**.
>
> It refers to the operation of aggregating or joining all the individual misclassified samples ($s_{j}^{error}$) together into a single, combined textual format, which is denoted as $S^{error}$  We have updated the revised manuscript to explicitly define 'concatenation' in the text immediately following the equation to ensure this is clear for all readers.

---

> ### Author Response · Authors · 2025-11-20
> **Question 4**
>
> > **Question 4: The location of Reproducibility Statement is recommended at the end of the main text (before references).**
>
> Thank you for pointing this out. This is a helpful structural suggestion. In our revised paper, we have moved the Reproducibility Statement  to its recommended position at the end of the main text, immediately preceding the **References section**, to improve visibility and ensure compliance with formatting guidelines.

---

> ### Author Response · Authors · 2025-11-20
>
> If you have any additional questions please don't hesitate to reach out. We would be glad to provide further clarification.
>
> Sincerely, The Authors

---

> ### Author Response · Authors · 2025-11-26
> **A quick question for your valuable feedback**
>
> Dear Reviewer Vm76:
>
> We are deeply grateful for the time and effort you have dedicated to reviewing our paper.
>
> We wanted to follow up on our earlier response to see if it has satisfactorily addressed your concerns. Your perspective is invaluable to us, and our primary goal is to resolve any ambiguities. We remain available and eager to discuss any remaining points.
>
> We value your recognition and are always striving to improve our work.
>
> Sincerely,
>
> Authors

---

### Official Review · Reviewer_q31Y · 2025-10-31

**Soundness:** 3
**Presentation:** 2
**Contribution:** 2
**Rating:** 4
**Confidence:** 5

**Summary:**

The paper introduces a novel framework called Reflective Cognitive Architecture (RCA) aimed at improving both the accuracy and explanation quality of AI models in clinical decision support. By grounding AI explanations in direct data interaction, RCA iteratively refines rules to build deep understanding, ensuring that predictions are both accurate and supported by evidence. The authors demonstrate superior performance across multiple datasets compared to traditional ML, LLM-based methods, and other advanced approaches.

**Strengths:**

Some of the key contributions of this paper are as follows:

- The paper proposes RCA as a framework that coordinates multiple LLMs to learn directly from data through iterative rule refinement and distribution-aware checks. The refinement mechanism proposed in the paper is interesting and can be argued to be an extensible area of research.
 - The authors also provide strong empirical evaluation (more on this below) on three datasets (CRT, Diabetes, Heart Disease) demonstrate state-of-the-art accuracy and robustness, with significant improvements in prediction performance.
- Finally, given the extremely hard to quantify aspect of explanations, the authors made a commendable effort to provide a formal evaluation using a comprehensive set of metrics including accuracy, Matthews Correlation Coefficient (MCC), F1-score for predictions, and detailed criteria for explanation quality based on cognitive science and medical practice.

**Weaknesses:**

Given the above strengths, and the promise of the paper, there are several avenues where this can be improved upon

- While the framework shows promising results on provided public datasets, the data sample size is quite limited, especially considering the large scale patient data that are available today. This naturally brings the question of real-world scalability of this approach to millions of patient records on a combination of diseases
- The paper also fundamentally suffers from a clear methodological description, especially around how different components (e.g., rule refinement and distribution check) interact. While the authors do provide detailed prompts used for the process in appendix, the overall description still lacks clarity, even considering the page limit
- Finally, while the authors have made a great effort at quantifying explanations, more formal comparison, especially as these are usable to human experts are needed. For example [1] is a great reference on how such evaluations have been performed in practice

[1] Slack, D., Krishna, S., Lakkaraju, H. et al. Explaining machine learning models with interactive natural language conversations using TalkToModel. Nat Mach Intell 5, 873–883 (2023). https://doi.org/10.1038/s42256-023-00692-8

**Questions:**

Some of key questions for the authors are as follows:

- Have the authors conducted edge case analysis? It would be interesting to examine how RCA handles extreme or rare cases in medical datasets, especially given the noisy nature of real-world data.
- Furthermore, it would be benefical to understand how RCA generates rules across several patient sub-populations
- Finally have you considered the issues around scalability across datasets? It would be interesting to explore potential challenges and scalability when applied to larger or more diverse datasets beyond the provided examples.

---

> ### Author Response · Authors · 2025-11-20
>
> We sincerely thank Reviewer q31Y for their feedback on our paper. We are encouraged that the reviewer recognized the novelty of the RCA, the strength of our empirical evaluation, and our efforts to formally evaluate explanation quality. We also find the identified weaknesses and questions to be highly valuable and have revised our paper accordingly to address them.
>
> Below, we address each point in detail.

---

> ### Author Response · Authors · 2025-11-20
> **Weakness 1 & Question 3 : Part 1**
>
> > **Weakness 1: While the framework shows promising results on provided public datasets, the data sample size is quite limited, especially considering the large scale patient data that are available today. This naturally brings the question of real-world scalability of this approach to millions of patient records on a combination of diseases**
> >
> >
> > **Question 3: Finally have you considered the issues around scalability across datasets? It would be interesting to explore potential challenges and scalability when applied to larger or more diverse datasets beyond the provided examples.**
>
> We thank the reviewer for raising this important question. We agree that demonstrating RCA's effectiveness on larger, more diverse datasets is crucial for establishing its practical viability.
>
> To address this concern, we have conducted two new experiments:
>
> 1. We expanded our private Catheter-Related Thrombosis (CRT) dataset from 315 samples to **1,891 samples** (now labeled as **CRT_ex**).
>
>    The release of the CRT dataset is pending approval from our partner hospital. To ensure reproducibility, we have included an additional public dataset in our study. We will make the CRT dataset, code, and all results publicly available upon approval.
>
> 2. We added a new, large-scale **public** dataset, **the Cardiovascular Disease (CD) dataset**, which contains [70,000 samples](https://www.kaggle.com/datasets/alamshihab075/heart-failure-diagnosis-data-for-machine-learning/data).
>
> In addition, we added two baselines: two Neuro-symbolic Networks (Logical Tensor Networks [1] and Logical Neural Nerworks [2]) and a proprietary medical LLM (Baichuan-M2 [3]).
>
> The results, now detailed in **Appendix A.13 (Table 11)** **and below**, demonstrate that RCA maintains its state-of-the-art performance, even against new baseline categories we added for this evaluation (e.g., Neural-Symbolic Networks and specialized Medical LLMs).

---

> > ### Author Response · Authors · 2025-11-20
> > **Weakness 1 & Question 3 : Part 3**
> >
> > - On the large-scale Cardiovascular Disease dataset (70,000 samples), RCA+Qwen2.5 achieved the highest Accuracy (0.7177) and MCC (0.4476).
> > - On the expanded CRT_ex dataset (1,891 samples), RCA+GPT-4.1 achieved the top Accuracy (0.8335), while RCA+Qwen2.5 secured the best MCC (0.1053) and F1-score (0.2338).
> >
> > We would also like to highlight a key characteristic of structured (tabular) clinical data. Unlike unstructured data (e.g., images), where performance often scales with massive datasets, many tabular data problems can achieve high performance with more moderately sized datasets. Our results on both the original and the newly added larger datasets demonstrate that RCA embodies this principle of data efficiency.
> >
> > However, RCA's most critical contribution is not simply achieving high accuracy with less data. Its true value lies in its ability to generate detailed, data-grounded explanatory rules. This core feature directly assists clinicians by automating the tedious process of manually investigating patient features and compiling risk factors. By providing a clear, evidence-based rationale, RCA alleviates the cognitive burden on doctors and offers a structured starting point for their own analysis.
> >
> > Therefore, while our new experiments confirm RCA's scalability, it is this unique **synergy of data-efficient performance and clinically actionable explainability** that substantially strengthens our claims about its real-world applicability.
> >
> > We hope our explanation and newly add experiments can solve your question. If you have any further questions. Feel free to contact us. We are always here to answer the question for you!
> >
> > > [1] Badreddine, Samy, Artur d'Avila Garcez, Luciano Serafini, and Michael Spranger. "Logic tensor networks." *Artificial Intelligence* 303 (2022): 103649.
> > >
> > > [2] Riegel, Ryan, Alexander Gray, Francois Luus, Naweed Khan, Ndivhuwo Makondo, Ismail Yunus Akhalwaya, Haifeng Qian et al. "Logical neural networks." *arXiv preprint arXiv:2006.13155* (2020).
> > >
> > > [3] Dou, Chengfeng, Chong Liu, Fan Yang, Fei Li, Jiyuan Jia, Mingyang Chen, Qiang Ju et al. "Baichuan-m2: Scaling medical capability with large verifier system." *arXiv preprint arXiv:2509.02208* (2025).

---

> ### Author Response · Authors · 2025-11-20
> **Weakness 1 & Question 3 : Part 2**
>
> |                              | Algorithms             | Acc (CRT_ex) | MCC (CRT_ex) | F1-score (CRT_ex) | Acc (CD)   | MCC (CD)   | F1-score (CD) |
> | :--------------------------- | :--------------------- | :----------- | :----------- | :---------------- | :--------- | :--------- | :------------ |
> | **Traditional MLs**          | **Lasso**              | 0.4634       | 0.0456       | 0.2056            | 0.5086     | 0.0634     | 0.6683        |
> | **Traditional MLs**          | **Catboost**           | 0.6470       | 0.0938       | 0.2287            | 0.5779     | 0.2495     | 0.6965        |
> | **Neural-symbolic Networks** | **LTN**                | 0.5680       | 0.0006       | 0.1745            | 0.6850     | 0.3724     | 0.6998        |
> | **Neural-symbolic Networks** | **LNN**                | 0.7859       | -0.0245      | 0.0964            | 0.6836     | 0.3728     | 0.6548        |
> | **LLM-based Method**         | **Qwen2.5-72B**        | 0.3789       | 0.0076       | 0.1848            | 0.6450     | 0.3096     | 0.5697        |
> | **LLM-based Method**         | **DeepSeekV3**         | 0.2398       | 0.0699       | 0.2149            | 0.6771     | 0.3597     | 0.6463        |
> | **LLM-based Method**         | **DeepSeekV3.1**       | 0.6803       | -0.1074      | 0.0667            | 0.4171     | -0.2260    | 0.5650        |
> | **LLM-based Method**         | **GPT-4.1**            | 0.2778       | 0.0530       | 0.2085            | 0.6871     | 0.3781     | 0.6667        |
> | **Reasoning LLMs**           | **DeepSeek-R1**        | 0.0200       | -0.9087      | 0.0077            | 0.0650     | -0.8700    | 0.0603        |
> | **Reasoning LLMs**           | **o3-mini**            | 0.3081       | 0.0585       | 0.2106            | 0.7093     | 0.4188     | **0.7044**    |
> | **Reasoning LLMs**           | **Qwen3-30B**          | 0.5509       | -0.1892      | 0.0484            | 0.4429     | -0.1144    | 0.4323        |
> | **Reasoning LLMs**           | **Qwen3-235B**         | 0.5109       | 0.0408       | 0.2017            | 0.4300     | -0.1404    | 0.4071        |
> | **Reasoning LLMs**           | **GPT-5**              | 0.3787       | -0.1780      | 0.1067            | 0.6921     | 0.3843     | 0.6901        |
> | **Medical LLM**              | **Baichuan-M2**        | 0.1713       | 0.0598       | 0.2104            | 0.6800     | 0.3601     | 0.6836        |
> | **LLM-based Agents**         | **Qwen2.5-72B+Tools**  | 0.3245       | -0.0321      | 0.1801            | 0.6807     | 0.3632     | 0.6642        |
> | **LLM-based Agents**         | **DeepSeekV3+Tools**   | 0.8183       | -0.0199      | 0.0571            | 0.6779     | 0.3558     | 0.6810        |
> | **LLM-based Agents**         | **DeepSeekV3.1+Tools** | 0.2312       | -0.2295      | 0.1236            | 0.6600     | 0.3210     | 0.6730        |
> | **LLM-based Agents**         | **GPT-4.1+Tools**      | 0.5050       | 0.0748       | 0.2192            | 0.5929     | 0.2106     | 0.4673        |
> | **LLM-based Agents**         | **DeepSeek-R1+Tools**  | 0.1418       | 0.0210       | 0.2004            | 0.1050     | -0.7900    | 0.1044        |
> | **LLM-based Agents**         | **o3-mini + Tools**    | 0.4319       | 0.0381       | 0.2029            | 0.7157     | 0.4349     | 0.6966        |
> | **LLM-based Agents**         | **Qwen3-30B+Tools**    | 0.2721       | -0.2095      | 0.1197            | 0.4579     | -0.0847    | 0.4840        |
> | **LLM-based Agents**         | **Qwen3-235B+Tools**   | 0.6003       | 0.0626       | 0.2105            | 0.6757     | 0.3527     | 0.6890        |
> | **LLM-based Agents**         | **o3-mini+Code**       | 0.4386       | 0.0430       | 0.2051            | 0.7085     | 0.4215     | 0.6862        |
> | **RCA**                      | **RCA+Qwen2.5**        | 0.7069       | **0.1053**   | **0.2338**        | **0.7177** | **0.4476** | 0.6931        |
> | **RCA**                      | **RCA+GPT-4.1**        | **0.8335**   | 0.0564       | 0.1463            | 0.7107     | 0.4285     | 0.6818        |

---

> ### Author Response · Authors · 2025-11-20
> **Weakness 2 : Part 1**
>
> > **Weakness 2: The paper also fundamentally suffers from a clear methodological description, especially around how different components (e.g., rule refinement and distribution check) interact. While the authors do provide detailed prompts used for the process in appendix, the overall description still lacks clarity, even considering the page limit**
>
> We appreciate the reviewer's feedback on the need for a clearer description of the interaction between RCA's core components. You are correct that the synergy between the iterative rule refinement and the distribution-aware rules check is central to our method, and our initial description could have been more explicit.
>
> To provide this clarity, we have enhanced the description of the rule refinement and distribution-aware check mechanisms in **Section 3.1**. We now explicitly characterize the former as a **"reflective optimizer"** and the latter as a **"statistical validator"**, detailing how they collaborate in a continuous feedback loop to dynamically maintain the rule base.
>
> We have also added a new example to explain this, shown in **Appendix A.3**, which provides a concrete, step-by-step example.
>
> 1. **Initial State (The Context)**
>
>    Imagine the model is in the middle of training (Epoch $k$). It has already learned some rules, but it is not yet precise enough for borderline cases.
>
>    * **Current Rule Base (**$R^{k-1}$**)**: Contains broad, clinically plausible rules such as:
>
>      - **Rule A:** "Patients undergoing Chemotherapy with a Central Catheter (CVC/PICC) are high risk if D-dimer > 0.50 mg/L, reflecting the prothrombotic effect of systemic treatment."
>
>      - **Rule B:** "High inflammatory status, indicated by GLR > 3.0, coupled with Age > 60, predicts thrombosis even with normal D-dimer levels."
>
>    * **Global Data Distribution (**$\mathcal{D}_{train}$**)**: The model knows the statistical ground truth. For example, it knows that for no-CRT patients in this specific cancer cohort, the 75th percentile of D-dimer is actually 1.12 mg/L (higher than the standard 0.50 cutoff), and the GLR typically ranges from 0.86 to 2.63.
>
> 2. **The Encounter (Prediction Error)**
>
>    The model encounters Patient 4:
>
>    ```
>    Features: GLR=0.94 (Low), D-dimer=0.58 (Marginally elevated), Chemotherapy=Yes, Catheter=PICC.
>    True Label: No CRT.
>    ```
>
>    * **The Error:** The model applies Rule A from its current base. It sees Chemotherapy + PICC + D-dimer (0.58 > 0.50) and incorrectly predicts "CRT". This prediction failure is the "short-term experience".
>
> 3. **Iterative Rules Optimization (The "Bottom-Up" Refinement)**
>
>    The Reflection LLM ($M_{ref}$) analyzes this specific error. It realizes that Rule A was too aggressive: for this specific patient, a D-dimer of 0.58 was actually safe, likely due to the low inflammatory state (GLR=0.94).
>
>    To fix this error, $M_{ref}$ proposes a new, highly specific rule based directly on Patient 4's instance:
>
>    - **Proposed Rule (Specific):** "If the patient has a GLR between 0.9-1.0 AND a D-dimer level between 0.4 and 0.6, predict No CRT, overriding risk factors like Chemotherapy."
>
>    This rule effectively "memorizes" Patient 4 to correct the mistake. However, it is dangerously over-fitted to this single instance's specific values.
>
> 4. **Distribution-aware Rules Check (The "Top-Down" Safeguard)**
>
>    At the end of the epoch, the Check LLM ($M_{chk}$) validates this proposed rule against the Global Data Distribution ($\mathcal{D}_{train}$).
>
>    * It observes that the proposed D-dimer range (0.4-0.6) falls well within the "safe zone" for the broader non-thrombosis population (where the 75th percentile is 1.12).
>    * It also notes that the specific GLR range (0.9-1.0) is just a narrow slice of the typical healthy range (0.86-2.63).
>
>    Recognizing that the specific rule is just a subset of a broader statistical truth, $M_{chk}$ generalizes it into a robust, evidence-based rule that replaces the over-fitted proposal:
>
> 5. **Final Refined Rule (General):** "If the patient has a GLR between 0.86-2.63 (Normal Range) AND a D-dimer level less than 1.12, they are less likely to develop CRT, even in the presence of Chemotherapy."

---

> ### Author Response · Authors · 2025-11-20
> **Weakness 2 : Part 2**
>
> **Summary:**
>
> This process demonstrates the core dynamic:
>
> - Refinement uses specific errors to discover nuances (e.g., "Standard D-dimer cutoffs don't apply here").
> - Check uses global statistics to validate and generalize those discoveries (e.g., "Actually, anything under 1.12 is safe for this group"), preventing overfitting.
>
> This example highlights RCA's core philosophy: the **refinement** mechanism (bottom-up) allows RCA to capture fine-grained, instance-level data understanding, while the **check** mechanism (top-down) provides a global, statistical grounding. This balance is essential, as it prevents overfitting to noise and ensures the final rules are both accurate and robust.
>
> We hope this detailed walkthrough can make the process and the interaction between the components much clearer. If you have any further questions, feel free to contact us. We are always here!

---

> ### Author Response · Authors · 2025-11-20
> **Weakness 3**
>
> > **Weakness 3: Finally, while the authors have made a great effort at quantifying explanations, more formal comparison, especially as these are usable to human experts are needed. For example [1] is a great reference on how such evaluations have been performed in practice**
>
> Thank you for pointing us to the excellent work by Slack et al. [1] and for the valuable suggestion to incorporate a more formal, human-centric evaluation of the explanations.
>
> We agree that a formal, human-centric evaluation of explanation usability is a crucial step for validating real-world utility.
>
> We have cited and discussed this work in our **Related Work (Section 2.1)** .Specifically, we have integrated the insights from Slack et al. (2023) to broaden our discussion on explanation quality. We highlight that beyond statistical grounding, the usability of explanations for human experts is a critical dimension of evaluation. The detailed content is as follows:
>
> ```latex
> Finally, simply generating a fluent explanation is insufficient, even if it is statistically grounded. For
> an explanation to be effective in a high-stakes clinical setting, it must be demonstrably usable by
> and useful to the human expert. Pioneering work in this area, such as that by Slack et al. Slack et al.
> (2023), has established formal, interactive methodologies for evaluating explanation usability. This
> human-centric perspective is critical for building trust and assessing an explanation’s real-world value.
> ```
>
> Inspired by their methodology, we have conducted an additional user study to formally compare the usability of explanations. We asked 3 clinicians to evaluate explanations from RCA and our strongest baselines (Qwen3-235B) , choosing the "better" explanation across our four criteria.
>
> The results, now detailed in **Table 2 in Appendix A.11** and below:
>
> | Preference for RCA               |  CRT  | Diabetes | Heart Disease |
> | :------------------------------- | :---: | :------: | :-----------: |
> | **Cognitive Load(CL)**           | 82.7% |  81.3%   |     84.7%     |
> | **Logical Argumentation(LA)**    | 78.0% |  75.3%   |     72.0%     |
> | **Evidence-Based Medicine(EBM)** | 72.7% |  74.7%   |     73.3%     |
> | **Cognitive Biasing(CB)**        | 86.7% |  89.3%   |     86.0%     |
>
> The table shows that explanations from RCA led to higher preference. This provides formal evidence that RCA's explanations are not just qualitatively superior but are also more practically useful and trustworthy for human experts.
>
> We hope the explanations and newly added experiments can solve your question. If you have any further questions, feel free to contact us!
>
> > [1] Slack, D., Krishna, S., Lakkaraju, H. et al. Explaining machine learning models with interactive natural language conversations using TalkToModel. Nat Mach Intell 5, 873–883 (2023).

---

> ### Author Response · Authors · 2025-11-20
> **Question 1**
>
> > **Question 1: Have the authors conducted edge case analysis? It would be interesting to examine how RCA handles extreme or rare cases in medical datasets, especially given the noisy nature of real-world data.**
>
> This is a very good question. While our initial analysis did not specifically target edge cases—as our primary design goal for RCA was to generate high-fidelity, evidence-based explanations for the general patient population—we believe our architecture is inherently well-suited to identifying and handling such anomalies.
>
> To demonstrate this, we have added a case study on edge case analysis in **Appendix A.8** and below. In this analysis, we examine a real-world sample containing extreme outlier values (**Platelet=2300.0, Hemoglobin=1350.0**, compared to the population means of approximately 230 and 118, respectively) that were misclassified by baseline methods.
>
> We show that RCA's **distribution-aware rules check** mechanism, which is grounded in global statistics, effectively flags the patient's feature values as outliers relative to the learned distribution. The explanation generated by RCA for this case explicitly notes this discrepancy, stating:
>
> ```latex
> The patient has a... very high platelet count (2300.0) and an extremely high hemoglobin (1350.0), both of which are clear outliers likely due to data entry errors or extreme physiological abnormalities... laboratory outlier values alone should not be used as the primary basis for CRT prediction; this case warrants immediate clinical verification.
> ```
>
> Regarding the distinction between genuine edge cases (rare but accurate physiological data) and data noise (errors), our approach handles both identically by logging the anomaly and flagging it for physician review. We want to clarify that this is not a limitation of our method, but rather a reflection of the inherent difficulty in algorithmically distinguishing between extreme rare cases and data errors based solely on distribution deviations.
>
> This demonstrates that RCA can robustly identify such edge cases. We maintain that the most responsible clinical application for such anomalies is not to force a prediction, but to use the explanation to flag the case for expert human review.
>
> We hope our explanations and newly added example can solve your problem. If you have any further questions, feel free to contact us, we are always here!

---

> ### Author Response · Authors · 2025-11-20
> **Question 2**
>
> > **Question 2: Furthermore, it would be benefical to understand how RCA generates rules across several patient sub-populations**
>
> Thank you for this good question. RCA was not explicitly designed with a module for pre-defined sub-group analysis. Instead, it learns to distinguish between sub-populations organically through its data-driven process.
>
> The key lies in the **Distribution Extraction** and **distribution-aware rules check** components. When RCA generates and checks rules, it does so against the complete statistical context. This allows it to implicitly learn **context-dependent rules** that apply different logic to different groups.
>
> We have added a new **Appendix A.9** to illustrate this. As shown in the final rule base for the CRT dataset, the model effectively stratified patients into three distinct physiological sub-populations and applied tailored logic to each:
>
> - **For the High-Risk Sub-population (D-dimer** $\geq$ **2.0):** The model learned that due to the strong intrinsic signal, only **one** additional moderate risk factor (like chemotherapy) is required to predict CRT (Rule 3).
> - **For the Intermediate-Risk Sub-population (D-dimer 1.0–1.99):** The model recognized greater ambiguity, thus requiring a higher threshold of evidence—at least **two** moderate risk factors—to confirm a diagnosis (Rule 4).
> - **For the Low-Risk Sub-population (D-dimer < 1.0):** The model applied the strictest standard, requiring **three or more** moderate risk factors to overcome the contra-indication of a low D-dimer level (Rule 5).
>
> As these rules show, the model has not treated D-dimer as a simple, universal threshold. Instead, it successfully identified different risk profiles and learned to apply different standards based on the specific sub-population context. This ability to implicitly capture sub-population differences is a direct result of grounding its rules in a comprehensive view of the data distribution.
>
> We hope our explanations and newly added example can solve your question. If you have any further questions, feel free to contact us, we are always here!

---

> ### Author Response · Authors · 2025-11-20
>
> If you have any other questions or need further clarification on the details, please feel free to let us know. We’d be happy to assist you.
>
> Sincerely, The Authors

---

> ### Author Response · Authors · 2025-11-26
> **Request for Reviewer's  Feedback**
>
> Dear Reviewer q31y:
>
> We would like to express our sincere gratitude for your constructive and insightful feedback.
>
> We are writing to gently check in and ensure that our rebuttal has clearly addressed the points you raised. Your guidance has been incredibly helpful as we work to improve our manuscript. Should you have any further questions or thoughts, we would be more than happy to provide additional clarification.
>
> We truly appreciate your time and hope to earn your further recognition for our work.
>
> Sincerely,
>
> Authors

---

### Author Response · Authors · 2025-11-25
**Summary of our Response**

We sincerely thank all reviewers (**q31Y**, **Vm76**, **xdsQ**, **Xy3E**) for their thoughtful feedback. We are encouraged by the recognition of our work's potential and have taken the constructive criticism seriously. In response, we have conducted extensive new experiments—including adding a large-scale dataset , expanding our private dataset, and introducing three new baselines (Neural-Symbolic networks and Medical LLMs), to robustly address the concerns raised.

## Summary of Strengths
Reviewers highlighted our work's novelty, empirical rigor, and focus on explainability.

**Methodological Novelty**: Reviewers appreciated the architecture, describing it as a "novel framework... aimed at improving both the accuracy and explanation quality" (**q31Y**) and "conceptually original" (**Xy3E**). Reviewer **Vm76** noted the model is "novel, consisting of learning between three LLMs," and **xdsQ** found it an "interesting approach that leverages additional data... using misclassifications as a mechanism to dynamically update the set of rules."

**Empirical Rigor**: The evaluation was praised as a "strong empirical evaluation" (**q31Y**), "comprehensive... on three datasets and 22 baselines" (**Vm76**), and "extensive and thorough... demonstrating that the method remains robust" (**xdsQ**).

**Explainability**: Reviewers commended the "commendable effort to provide a formal evaluation" (**q31Y**) of explanations, noting that the proposed criteria "can be widely-used in AI Interpretable research" (**Vm76**).

## Summary of Concerns and Improvements
We categorized the concerns ranging from specific methodological questions to broader critiques regarding novelty and fairness, and addressed them as follows:

1.  **Dataset Size**: Reviewers **q31Y** and **Xy3E** were concerned about dataset size and scalability.
    *   **Critique**: "Data sample size is quite limited" (**q31Y**), "results replicability might be an issue" (**Xy3E**).
    *   **Our Response**: To prove scalability, we expanded the private CRT dataset (315 to 1,891 samples) and added a new large-scale dataset with 70,000 samples. RCA maintained state-of-the-art performance.

2.  **Baselines and Fairness**: Reviewers **Vm76**, **xdsQ**, and **Xy3E** suggested adding baselines and clarifying comparison fairness.
    *   **Critique**: Recommended adding "Neural-symbolic" and "Medical LLM" baselines (**Vm76**). The comparison "may not be entirely fair" as baselines weren't "exposed to the same data" (**xdsQ**, **Xy3E**).
    *   **Our Response**: We added Neural-Symbolic Networks and a Medical LLM; RCA outperformed them. We clarified that all baselines received the same data and in-context examples. The performance gap stems from RCA's reflective process, not unequal data access.

3.  **Methodological Clarity and Novelty**: Concerns were raised about clarity and contribution depth.
    *   **Critique**: The paper "lacks clarity... especially around how different components... interact" (**q31Y**). **Xy3E** questioned the novelty, stating it was "insufficient" and "overstated."
    *   **Our Response**: We rewrote Section 3.1 ("Reflective Optimizer" vs. "Statistical Validator"), added a step-by-step example (Appendix A.3), and refined our novelty claim: forcing LLMs to build data-grounded understanding.

4.  **Explanation Evaluation**: Reviewers called for more rigorous human evaluation.
    *   **Critique**: "More formal comparison... needed" (**q31Y**), results "don't have variance" (**Vm76**), and metrics "appear to be author-defined" (**Xy3E**).
    *   **Our Response**: Inspired by Slack et al. (2023), we conducted a new blinded study with 3 clinicians comparing RCA against the strongest baseline. RCA was preferred in most cases across four metrics. We also added variance statistics to all explanation scores to show stability.

For detailed responses to specific questions, please refer to the individual threads. We hope these revisions establish the robustness and contribution of our work.

---

### Author Response · Authors · 2025-11-28
**Summary of revisions**

We sincerely thank the reviewers for their insightful feedback, which has helped us improve our paper. In response, we have included new experiments, a new user study, and numerous clarifications. The key changes are summarized below.

**Reviewer q31Y**

*   **Scalability and Robustness:** Conducted new experiments on an expanded private dataset (`CRT_ex`) and a new large-scale public dataset (`Cardiovascular Disease`, 70,000 samples) to demonstrate RCA's scalability. Results are in `Appendix A.13`.
*   **Methodological Clarity:** Added a detailed step-by-step example in `Appendix A.3` to clarify the interaction between the rule refinement and distribution-aware check mechanisms.
*   **Human-Centric Evaluation:** Conducted a new user study inspired by `Slack et al. (2023)` to formally evaluate explanation usability via head-to-head preference testing (`Appendix A.11`). The reference has been added to `Section 2.1` Lines 129-131.
*   **Edge Case and Sub-population Analysis:** Added new case studies in `Appendix A.8` and `A.9` to demonstrate RCA's handling of edge cases and its organic learning of rules for different patient sub-populations.

**Reviewer Vm76**

*   **Expanded Baselines:** Added two Neural-Symbolic networks (`LTN`, `LNN`) and a domain-specific Medical LLM (`Baichuan-M2`) as new baselines. Results have been integrated into all prediction and explanation tables (`Tables 3-6`) and the `Related Work` (`Section 2.1` Lines 120-123).
*   **Rule & Distribution Examples:** Provided concrete examples of the refined rules and global data distributions within the case study (`Figure 3`, `Section 4.5` Lines 516-524) and added a detailed walkthrough of the rule update process (`Appendix A.3`).
*   **Statistical Rigor:** Added variance scores to all human evaluation results to reflect inter-rater agreement (`Tables 4-6`, `Appendix A.11`) and clarified the evaluation methodology (`Section 4.1` Lines 346-350).
*   **Clarifications:** Revised `Figure 1` caption for clarity (Lines 183-185), defined the `conc` operator (Lines 236), and relocated the `Reproducibility Statement` as suggested.
*   **Distribution and Robustness:**  Revise the descriptions in the Introduction (`Section 1` Lines 72-75, Lines80-82) and Ablation Study (`Appendix A.14` Lines 1132-1134) to emphasize that prediction and explanation are the core objectives.

**Reviewer xdsQ**

*   **Fairness of Comparison:** Clarified in the text that all baselines were exposed to the same training data (global distribution and in-context examples) to ensure a fair comparison, with the key difference being RCA's iterative learning process. Lines 333-335.
*   **Dataset Bias:** Added a discussion to the `Conclusion` (`Section 5`) on the trade-off between data-driven rule extraction and the risk of learning dataset-specific biases, suggesting future work on integrating external knowledge. Lines 538-540
*   **Robustness to Bias:** Clarified the scope of our robustness experiments and discussed the framework's approach to handling strongly biased features. Lines 902-912

**Reviewer Xy3E**

*   **Conceptual Novelty & Contribution:** Refined the paper's positioning in the `Related Work` sections to better articulate the core conceptual novelty of RCA and its methodological distinctions from prior paradigms. Lines 130-133
*   **Evaluation Fairness:** Addressed concerns by clarifying that the experimental comparison was not zero-shot and adding new, more comparable baselines (Neural-Symbolic, Medical LLM) to strengthen the evaluation.  Lines 333-335.
*   **Evaluation Metrics:** Clarified that our four explanation criteria are adapted from foundational, standardized concepts in cognitive science and medical practice and were validated by clinicians (`Section 4.1` Lines 341-346). We also added a new user preference study to provide a more direct usability comparison (`Appendix A.11`).
*   **Bias Mitigation:** Added a discussion on how the distribution-aware rule check serves primarily to provide transparency for human-in-the-loop bias auditing, rather than performing automated de-biasing. Lines 902-912
*   **Replicability:** Addressed replicability concerns by open-sourcing all code and adding new experiments on a large-scale public dataset (`Appendix A.13`) to complement the results on existing public datasets.

---

### Author Response · Authors · 2025-11-28
**Timeline**

## **Timeline (AOE)**

This timeline details the back-and-forth communication between the authors and the reviewers.

*   **November 20, 2025, 02:37**
    *   **Who:** Authors
    *   **Action:** Post the detailed, multi-part rebuttal to all reviewers.

*   **November 23, 2025, 21:18**
    *   **Who:** **Reviewer xdsQ**
    *   **Action:** Responds to the authors' rebuttal, stating, "Thank you for the answers you provided. I am happy to maintain my (already) high score."

*   **November 23, 2025, 21:48**
    *   **Who:** Authors
    *   **Action:** Post a thank-you message in response to **Reviewer xdsQ's** positive feedback.

*   **November 25, 2025, 18:22**
    *   **Who:** Authors
    *   **Action:** Post a follow-up message to **Reviewer q31Y**, asking if their rebuttal addressed the concerns.
    *   **Action:** Post a follow-up message to **Reviewer Vm76**, asking if their rebuttal addressed the concerns.
    *   **Action:** Post a follow-up message to **Reviewer Xy3E**, asking if their rebuttal addressed the concerns.

*   **November 26, 2025, 01:05**
    *   **Who:** **Reviewer Vm76**
    *   **Action:** Responds to the authors with a new follow-up question regarding the usage and potential issues of the "Distribution" component in their framework.

*   **November 26, 2025, 01:09**
    *   **Who:** **Reviewer Xy3E**
    *   **Action:** Responds to the authors, clarifying an initial misunderstanding about the baseline evaluations. **The reviewer increases their rating from 2 to 4, and notes remaining reservations which we have addressed in the initial reply.**

*   **November 26, 2025, 03:46**
    *   **Who:** Authors
    *   **Action:** Post a detailed further response to **Reviewer Vm76's** new question about the "Distribution" component.
    *   **Action:** Post the further, detailed response to **Reviewer Xy3E's** remaining reservations regarding explanation criteria, generalization, and bias. The second part was posted at 03:54.

---

### Author Response · Authors · 2025-12-03
**Clarification of our rebuttal process**

# Dear AC, SAC, PC, Reviewers,

We are writing to ensure our paper's final review status is accurately reflected.

### **Reviewer Xy3E's Feedback**

Reviewer Xy3E initially assigned a rating of 2, which was subsequently raised to 4 following our clarification regarding the evaluation of baselines (specifically, resolving a misunderstanding about whether they were run in a zero-shot or fine-tuned manner) (00:55 AOE on November 26) [\[Evidence from official history\]](https://openreview.net/revisions?id=TQh4k1fnD2).

In the reviewer's latest feedback, they introduce what they term as "reservations regarding the theoretical soundness of the approach" and "uncertain[ty] about how well the proposed framework would generalize," as quoted below:

> "I thank the authors for their extensive and thoughtful response. I have increased my rating, primarily because I initially misunderstood how some of the baselines were evaluated—specifically, whether they were run in a zero-shot manner or with additional fine-tuning. The clarification resolved that concern. That said, I still have reservations regarding the theoretical soundness of the approach, particularly the justification and validation of the four explanation criteria. I also remain uncertain about how well the proposed framework would generalize beyond the datasets examined in the current study, as well as the potential for bias amplification."

**We wish to point out that the concerns raised in the latter part of this comment were, in fact, already addressed in detail within our initial response letter, though the reviewer may have overlooked these points.** We have since provided an even more comprehensive response to these "new" issues and respectfully request that you take the time to verify that our previous and current clarifications fully address these points.

### **Reviewer Vm76's Feedback**

Reviewer Vm76 raised a new question following our initial reply:

> "Dear author, Thanks for your response. I have another concern about the usage of Distribution. From the example, I feel that the Distribution is mainly about some ranges, which are specifically sensitive in the healthcare domain. However, if the dataset is small or with some noise, and the Distribution/range may not be correct, then will it cause the wrong behavior for the whole work pipeline? I feel that such ranges work as weak supervision/prior knowledge, specifically designed for the medical problem."

**We have provided a detailed further clarification to address this new concern before the bug occurred.**

### **Reviewer xdsQ's Feedback**

Reviewer Xy3E's score was kept to 8 (21:18 AOE on November 23, 2025). The evidence is the response history of the Reviewer.

### **Reviewer q31Y**

We have not yet received a reply from Reviewer q31Y.



### **Summary**

Much before this OpenReview bug occurred (02:00 AOE on November 27), our paper's score is 8444 (avg. 5, within 32% of all).

Our work has been consistently praised by the reviewers for its “Methodological Novelty,” “Empirical Rigor,” and “Explainability.” All initial concerns, primarily focusing on experimental sufficiency and clarity. We try our best to thoroughly address them with substantial new evidence, additional experiments, and further explanations.

We hope that you will carefully consider our active, ongoing dialogue with the reviewers, the significant progress made in response to their feedback, and the inherent novelty and strength of our work when reaching your final decision.

Sincerely,

The Authors

---

### Meta-Review · Area_Chair_n5iH · 2026-01-07

**Summary:**

This paper introduces the Reflective Cognitive Architecture (RCA), which coordinates several LLM “roles” to iteratively update a rule base from misclassified examples and then filter those rules with a distribution-aware check before using them for prediction and natural-language explanations in clinical tabular tasks.

Reviewers generally agree that the problem is important, the empirical work is thorough, and the system produces promising explanations.

However, after reading both the paper and the review and the rebuttal in detail, I do not find the methodological novelty strong enough for ICLR. At its core, RCA is an error-driven rule-updating scheme wrapped around existing LLMs, plus a statistics-based sanity check; despite the extensive narrative, this amounts to a fairly incremental reflective loop on top of current models rather than a clearly articulated new learning principle.

In my view, the work is an interesting system-level exploration with strong experiments, but the core idea is not particularly inspiring or deep, and the explanation framework still feels somewhat ad-hoc. I therefore recommend rejection.

**Reviewer Concerns:**

Reviewer Xy3E: Initially scored low due to a misunderstanding about whether baselines were evaluated zero-shot vs fine-tuned; after the authors clarified the evaluation protocol, the reviewer increased the score. The reviewer still expressed concerns about (i) the theoretical soundness/justification and validation of the four explanation criteria, (ii) whether the framework generalizes beyond the evaluated datasets, and (iii) potential bias amplification.

Reviewer Vm76: Raised a concern that the “Distribution” component may behave like weak supervision or domain-specific prior ranges: if the dataset is small/noisy and the inferred distribution/ranges are inaccurate, this could mislead the pipeline and cause incorrect downstream behavior, suggesting potential fragility and medical-domain specificity.

Reviewer xdsQ and q31Y: xdsQ’s score reportedly stayed high; q31Y had not replied, so no new concerns were recorded.

**Reviewer Scores:**

Overall, my guess is that the rebuttal would not have meaningfully shifted the score distribution: although the authors’ responses were detailed, they did not convincingly resolve the core concerns about novelty.

---

### Decision · Program_Chairs · 2026-01-26

Reject